# Highly efficient construction of monkey blastoid capsules from aged somatic cells

Junmo Wu [1,2,6], Tianao Shao[1,2,6], Zengli Tang[1,2,3,6], Gaojing Liu[4,5,6], Zhuoyao Li[1,2,6], Yuxi Shi[1,2], Yu Kang [1,2], Jiawei Zuo[1,2,3], Bo Zhao [4], Guangyu Hu[1,2], Jiaqi Liu[1,2], Weizhi Ji [1,2] ✉, Lei Zhang [1,2] ✉ & Yuyu Niu [1,2,3] ✉

Blastoids—blastocyst-like structures created in vitro—emerge as a valuable model for early embryonic development research. Non-human primates stem cell-derived blastoids are an ethically viable alternative to human counterparts, yet the low formation efficiency of monkey blastoid cavities, typically below 30%, has limited their utility. Prior research has predominantly utilized embryonic stem cells. In this work, we demonstrate the efficient generation of blastoids from induced pluripotent stem cells and somatic cell nuclear transfer embryonic stem cells derived from aged monkeys, achieving an 80% formation efficiency. We also introduce a hydrogel-based microfluidics platform for the scalable and reproducible production of size-adjustable, biodegradable blastoid capsules, providing a stable 3D structure and mechanical protection. This advancement in the high-efficiency, scalable production of monkey blastoid capsules from reprogrammed aged somatic cells significantly enhances the study of embryonic development and holds promise for regenerative medicine.

Blastoids—blastocyst-like structures generated in vitro—have recently emerged as a promising tool to systematically study early human development without requiring access to numerous donated human embryos[1–5]. Blastoids form from naïve pluripotent stem cells (PSCs) through successive lineage differentiation and self-organization. They comprise all three lineages of the blastocyst and have been shown to faithfully mimic human embryonic and extraembryonic development from pre-implantation to early gastrulation stages[6].

While studying human blastoids will undoubtedly provide important insights, longer-term culture and in vivo maternal transplantation of these are widely prohibited for ethical reasons. Blastoids derived from non-human primates (NHPs) present a powerful alternative for experimental investigations and should provide vital insights while circumventing ethical issues[7,8]. NHPs (such as cynomolgus monkeys and rhesus monkeys) are closely related to humans

and exhibit ageing-related physiological decline in various organs, including the ovaries[9], skin[10], placenta[11], and liver[12]. Hence, discoveries made in monkey blastoids would complement findings on early embryonic development made in other model systems[8] and likely generalize to human embryonic development. However, studies on monkey PSC-derived blastoids remain scarce. In fact, only one study by Liu and colleagues[8] recently reported the generation of blastoids using cynomolgus monkey embryonic stem cells (ESCs) with an efficiency of ~ 30%.

The generation of blastoids from aged autologous PSCs has so far not been reported in either human or monkey. Autologous—or patient-specific—PSCs are important cell sources in regenerative medicine, reflecting the individual status and reducing immune rejection[13]. An efficient method to develop blastoids from aged autologous PSCs is desirable given the greater practical significance of using somatic

[1]State Key Laboratory of Primate Biomedical Research; Institute of Primate Translational Medicine, Kunming University of Science and Technology, Kunming, Yunnan, China. [2]Yunnan Key Laboratory of Primate Biomedical Research, Kunming, Yunnan, China. [3]Southwest United Graduate School, Kunming, Yunnan, China. [4]Key Laboratory of Genetic Evolution & Animal Models, Kunming Institute of Zoology, Chinese Academy of Sciences, Kunming, Yunnan, China. [5]University of Chinese Academy of Sciences, Beijing, China. [6]These authors contributed equally: Junmo Wu, Tianao Shao, Zengli Tang, Gaojing Liu, Zhuoyao Li. ✉e-mail: wji@lpbr.cn; zhangl@lpbr.cn; niuyy@lpbr.cn

donor cells. Further, the differentiation capabilities of autologous PSCs from aged sources could be investigated in a blastoid system, providing a unique perspective for in-depth studies of tissue regeneration in aged individuals[14]. Autologous PSCs can be derived from somatic cells by either of two approaches: somatic cell nuclear transfer[15–18] (SCNT), resulting in nuclear transfer ESCs (nt-ESCs), and reprogramming by transcription factors, resulting in induced pluripotent stem cell (iPSCs)[19,20].

Another advantage of using blastoids as a model system is their potential for scalability. Utilizing microfluidic techniques together with appropriate biomaterials, recent progress has been made in developing platforms to create 3D organoids−mini-organ-like structures−with greater physiological relevance in a controllable, scalable manner[21–23]. An integrative strategy that combines blastoids derived from aged monkey PSCs, droplet microfluidics, and biomaterials would offer a robust and scalable platform for blastoid research and regenerative medicine.

In this work, we establish a highly efficient blastoid generation system using different cell sources as starting cells; these include monkey ESCs (M-ESCs), monkey-induced PSCs (M-iPSCs) and monkey nt-ESCs (M-nt-ESCs) from young and aged rhesus monkey fibroblasts (Fig. 1). We explore the appropriate initial cell number, lineage differentiation and self-organization capabilities of monkey PSCs obtained from old and young monkeys. In addition, we develop an engineering platform for hydrogel-based blastoid capsules in a monodisperse, consistent, and reproducible manner. We find that blastoid encapsulation in hydrogel using our platform improves mechanical preservation, enhances the blastoids' stability and offers an effective means for their delivery through fallopian tubes. In summary, we provide proof-of-concept that NHPs aged somatic cells possess the capability to efficiently form blastoids and present the generation of blastoid capsules in a monodisperse, scalable, and reproducible manner.

## Results

### Generation of monkey blastoids
To establish our monkey blastoid formation protocol, we first aimed to increase monkey PSCs viability, as this impacts the formation of embryoid bodies and organoids[24]. We found that the supplement CloneR in combination with either of the small molecules DDD00033325 (Pro-S) or Y27632[25] effectively promoted the survival of M-ESCs following cell passaging (Supplementary Fig. 1a–g) in our previously established culture medium (XF-PSC medium)[26]. To promote the aggregation of individual cells into blastoids, most studies have used the AggreWell™400 microwell plate along with small molecular compounds[27]. After inoculation of the digested single M-ESCs into AggreWell™400 microwell plates, we found that cells treated with the "CloneR + Pro-S" combination exhibited the lowest degree of cell death and the highest proliferation levels. Cell spheroids formed after two days, suggesting that these conditions are most favorable for constructing monkey blastoids (Supplementary Fig. 1h, i).

Various methods for generating human blastoids have been reported recently[1–3,5]. In our approach, we used hypoblast differentiation medium (HDM) followed by a modified version of trophoblast differentiation medium[3], which we termed TSG. TSG is a 1:1 mixture of our XF-PSC medium with trophoblast stem cells medium and resulted in more effective monkey blastoid formation. To examine the ability of M-ESCs to differentiate into cells resembling all cell types of a blastocyst in vitro, GATA3 + cells (trophoblast-like cells, TLCs) and GATA4 + cells (hypoblast-like cells, HLCs) stably expressed themselves on day 5 in HDM and TSG induction medium, respectively (Supplementary Fig. 2a, b). Cavitation of blastocysts and correct lineage formation can be observed after two days of TSG culture (Supplementary Fig. 2c−e).

We combined cultured M-ESCs with HDM and TSG, and the resulting cell aggregates contained compartments of epiblast-like cells (ELCs) and trophectoderm-like cells (TLCs) with a visible cavity; we term these structures "monkey blastoids" (Fig. 2a, b). The blastoids exhibited morphological similarities to cynomolgus monkey blastocysts at embryonic day 9 (Fig. 2c). To explore the contributions of HDM and TSG during monkey blastoid formation, we removed individual TSG components and found the subtraction of exogenous trophectoderm induced component (Supplementary Fig. 2f) reduced the efficiency of monkey blastoid formation, whereas at the induction stage with TSG, the efficiency was reduced by adding HDM (Fig. 2e). After the removal of A83-01 and PD0325901, blastoid cavitation still occurs (Supplementary Fig. 2g). We consider this may be similar to previously reported phenomenon, potentially due to the 3D structural reasons[28]. Adding lysophosphatidic acid (LPA) did not significantly improve the cavitation of monkey blastoids (Supplementary Fig. 2h), likely due to the current cavitation rate being around 80%. While the Hippo-YAP/TAZ signaling pathway plays an important role in trophectoderm (TE) specification in mouse[29,30] and human blastocysts[31], further studies are needed to validate its role in monkey TE lineage development.

Blastocyst formation commences with the delamination of epithelial TE cells from the surface of the undifferentiated morula, leading to the generation of a fluid-filled cavity[32,33]. Recent studies have reported the efficient generation of human blastoids[5,27]. However, the efficient generation of monkey blastoids has yet to be reported, with current rates around 30%[8], indicating that there is still potential for further optimization in non-human primate models. With our approach, we quantified blastoid cavity formation efficiency and achieved a blastoid formation efficiency of over ~ 80% (Fig. 2d).

### Generation of monkey blastoids from higher passage cell lines
The number of initial cells and passage number of cell lines are known factors affecting the efficiency of blastoid formation[1,27]. We found that increasing the starting number of cells (up to 200,000, 2E5) and extending the HDM culture time supported blastoid formation from higher passage M-ESCs (passage > 25) (Fig. 2f−i). We confirmed this result using immunofluorescence stainings, observing the markers representing the three lineages by Day 5 (Supplementary Fig. 2i, j). A starting number of around 100,000 (1E5) higher passage M-ESCs and culture in HDM for 3 days resulted in the highest blastoid formation efficiency with optimal characteristics (Fig. 2j, k). In summary, we developed a method to improve the efficiency of blastoid formation using higher passage M-ESCs.

### Lineage composition of monkey blastoids
We next analyzed the lineage composition of our monkey blastoids using immunofluorescence staining. POU5F1 (OCT4) was highly expressed in ELCs; HLCs expressed GATA4 and GATA6, and TLCs were stained positive for the TE marker GATA3 (Fig. 2l and Supplementary Fig. 2k). Intercellular tight junctions between the TLCs were further validated by demonstrating expression of ZO-1 (Fig. 2m). To explore cell lineage differentiation dynamics during blastoid formation, we performed an immunofluorescence time-course analysis. GATA6 + and GATA3 + cells first appeared on days 2 and 4, respectively. Around day 5, cells expressing GATA6, GATA3, and OCT4 began to separate, in a similar manner to our previous observation in cynomolgus monkey embryos[34] (Supplementary Fig. 2l). In addition, monkey blastocysts were included as a parallel comparison, Specifically, OCT4, NANOG and SOX2 for epiblast, SOX17 for hypoblast, CDX2, GATA3 and GATA2 for trophoblast, and CCR7 for polar trophoblast[35] (Fig. 2n−r).

To accurately evaluate the transcriptional states of monkey blastoids, we included high-quality single-cell transcriptomes from multiple publicly available datasets for comparison. These included cells from monkey embryos using the 10X platform[36], monkey embryos via Smart-seq2[37,38], human PXGL-blastoid[1], monkey 4CL-blastoids[8], and a comprehensive in vitro amnion dataset[39].

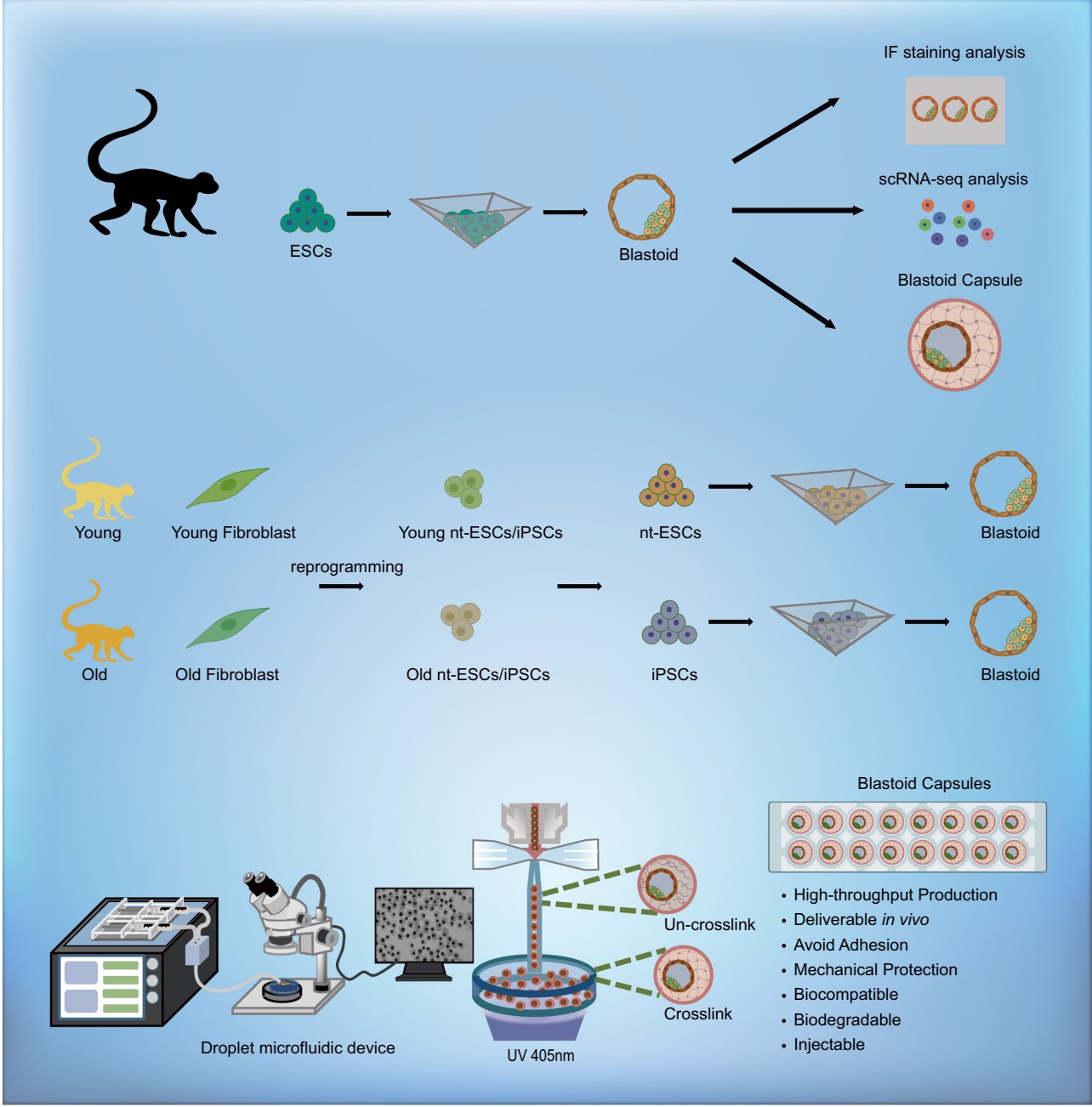

**Fig. 1 | A schematic diagram of the experimental design.** Generation of blastoids from M-ESCs, as well as from old and young M-nt-ESCs and M-iPSCs. High-throughput preparation of blastoid capsules using droplet microfluidic devices.

After unbiased clustering, we first observed a high degree of similarity between natural embryos and monkey blastoids (Supplementary Fig. 3a, b). We further identified cell clusters, in which we characterized clusters with ELC signatures (Clusters 4 and 7), HLC signatures (Cluster 2), and TLC signatures (Clusters 1, 3, and 6) according to the expression of corresponding marker genes and natural embryos cell ratio (Supplementary Fig. 3c–g). The ratio of ELC, HLC, and TLC was 2:2:6 in blastoids. (Supplementary Fig. 3h).

Single-cell transcriptomes of blastoids generated in this study clustered well with blastocysts and blastoids from other datasets (Fig. 3a–c and Supplementary Fig. 3i). Each blastoid lineage exhibited distinct gene expression profiles (Fig. 3d and Supplementary Data 1). The correlation analysis of various clusters in blastoids with each lineage in natural embryo showed high similarity (Fig. 3e). Specific

expression of marker genes of different lineages exhibited ELC, TLC, and HLC signatures in respective clusters on UMAP, which were further confirmed by specific expression of marker genes of different lineages (*POU5F1* and *NANOG* for ELC, *NR2F2* and *TEAD4* for TLC, and *GATA4* and *TTR* for HLC, Supplementary Fig. 3j–l). However, the amnion markers *ISL1* and *GABRP* were expressed at low levels in blastoid clusters (Fig. 3f). Meanwhile, in monkey blastoids, *ISL1* expression was detected in only a few cells at low levels (Fig. 3g). And we performed KEGG pathway and GO pathway analysis, for GO enrichment, ELCs genes were enriched in signal transduction, BMP signaling pathway and WNT signaling pathway which are related with primitive steak (Supplementary Fig. 3m–r and Supplementary Data 2).

Together, the transcriptomics results suggest similarity in gene expression patterns and lineage composition between our monkey

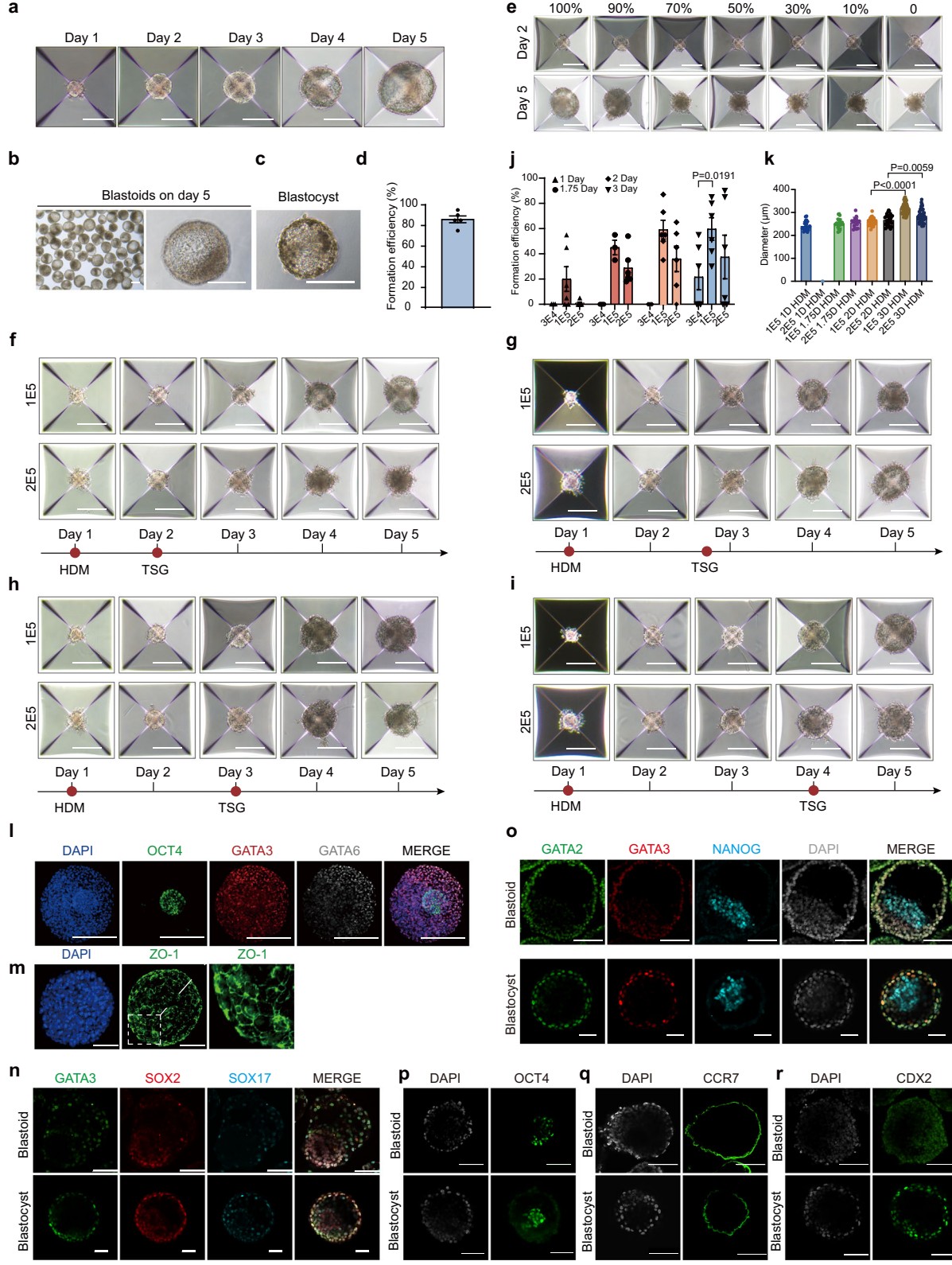

blastoids and published blastoids (human and monkey) and blastocysts (monkey) (Supplementary Fig. 3s). Moreover, the comparative analysis of the scRNA-seq dataset revealed transcriptional differences between monkey blastoids and blastocysts, providing a unique resource for further improving the monkey blastoid model in the future.

## Establishment of PSC lines from young and aged somatic cells

Having established a protocol to generate high-quality blastoids from M-ESCs, our next goal was to use reprogrammed somatic cells for blastoid formation. Cellular reprogramming techniques reset differentiated somatic cells to a state akin to PSCs, erasing their original differentiation memory. This process is considered a

**Fig. 2 | Generation and optimization of blastoid formation from higher passage ESC lines. a** Representative phase-contrast images of ESCs aggregates at indicated time points during blastoid formation. Scale bars, 200 µm. **b** and **c** Representative phase-contrast images of monkey blastoids (**b**) and monkey blastocyst-E9 (**c**). Scale bars, 200 m. **d** Blastoid formation efficiency from M-ESCs, mean ± S.E.M. of 5 biological replicates. **e** The effect of HDM components, indicated as a percentage of HDM concentration, on blastoid formation efficiency during the TSG stage. Scale bars, 200 µm. **f**–**i** Representative phase-contrast images of higher passage ESCs aggregates from different starting cell numbers (100,000, 1E5; 200,000, 2E5) and HDM culture time at indicated time points during blastoid formation. Scale bars, 200 µm. **j** Blastoid formation efficiency from higher passage M-ESCs at different M-ESC starting cell numbers and HDM culture time, $n = 6$ technical replicates. Error bars, mean ± S.E.M. Two-tail $t$ - test. **k** Diameter of blastoids at higher passage M-ESCs number aggregates and HDM culture time during blastoid formation,

$n = 47$ technical replicates. Error bars, mean ± S.E.M. Two-tail $t$ - test. **l** Representative immunofluorescence co-staining images of GATA3, OCT4 and GATA6 (3 independent replicates). Scale bars, 100 µm. **m** Representative immunofluorescence images of ZO-1 in monkey blastoids (3 independent replicates). Scale bars, 100 µm. **n** Representative immunofluorescent staining images of GATA3/SOX2/SOX17 in monkey blastoid (top) and in monkey blastocyst-E9 (bottom). Scale bars, 100 µm. **o** Representative immunofluorescent staining image of GATA2/GATA3/NANOG in monkey blastoid (top) and blastocyst (bottom). Scale bars, 100 µm. **p** Representative immunofluorescent staining image of OCT4 in monkey blastoid (top) and blastocyst (bottom). Scale bars, 100 µm. **q** Representative immunofluorescent staining image of CCR7 in monkey blastoid (top) and blastocyst (bottom). Scale bars, 100 µm. **r** Representative immunofluorescent staining image of CDX2 in monkey blastoid (top) and blastocyst (bottom). Scale bars, 100 µm. 3 independent replicates for (**n**–**r**).

strategy for delaying aging and mitigating its associated adverse effects[19,40–42]. However, the age of the donor can influence reprogramming efficiency[40].

SCNT (Fig. 4a) and Sendai virus-mediated non-integrative gene reprogramming (Fig. 4b) are currently the most widely used reprogramming techniques. We isolated fibroblasts from the skin of aged and young monkeys and reprogrammed these into M-iPSCs and M-nt-ESCs using the respective techniques (Fig. 4c, d). All autologous PSC lines expressed the expected pluripotency marker genes (Fig. 4e, f). They all exhibited the ability to differentiate into all three germ layers (Supplementary Fig. 4a–d) and maintained normal karyotypes (Fig. 4g, h). This confirmed the successful generation of M-iPSCs and M-nt-ESCs from monkey-derived fibroblasts of both young and aged donors.

### Generation of monkey blastoids from young and aged PSCs
In individuals, aging manifests itself as a progressive loss of physiological integrity, which leads to functional impairment and an increased risk of death[43]. By constructing blastoids from autologous PSCs derived from aged monkeys, the potential differentiation capabilities of autologous PSCs from aged sources can be investigated. This approach provides a unique perspective for in-depth studies of tissue regeneration in aged individuals[14].

To examine the ability of M-iPSCs and M-nt-ESCs to differentiate into cells resembling all cell types of a blastocyst in vitro, GATA3 + / TFAP2C + cells and GATA4 + /SOX17 + cells expressed themselves on day 5 in HDM and TSG induction medium, respectively (Supplementary Fig. 5a, b). We hence used M-nt-ESCs and M-iPSCs derived from both young and aged somatic cells as starting cells and found that the generated monkey blastoids exhibited similar morphologies to those constructed from M-ESCs (Fig. 5a–d). Immunofluorescence staining confirmed the expression of markers representative of three lineages in all blastoids (OCT4, GATA3, and GATA4) (Fig. 5e–h and Supplementary Figs. 5c–f). TLCs were validated by demonstrating the expression of ZO-1 (Fig. 5i–l). We statistically analyzed cavitation efficiency and found that monkey blastoids constructed from M-iPSCs and M-nt-ESCs derived from young donor cells had a similar cavitation efficiency to those constructed from M-ESCs. The monkey blastoids constructed from M-iPSCs and M-nt-ESCs derived from aged donor cells had a lower cavitation efficiency compared to those from young donor cells, but still achieved about 60% efficiency (Fig. 5m–p).

Hence, we have developed a highly efficient protocol to generate NHPs blastoids from various cell sources by adding bioactive compounds that boost the viability of the stem cells and by optimizing the methods for constructing blastoids.

### Derivation of stem cell lines
ESCs, extraembryonic endoderm (ExEnd) cells, and TSCs can be derived from natural blastocysts in vitro[44]. Similarly, following human

blastoid[3,45] and monkey blastoid[8] reported previously, we successfully derived ESCs, TSCs, and ExEnd cell lines (bELCs, bTLCs, and bHLCs) from individually plated blastoids (Supplementary Fig. 6a). The bELCs, bTLCs, and bHLCs derived from blastoids can be stably passaged for more than 10 generations in vitro. Using immunofluorescence, we detected robust expression of epiblast-specific markers, including NANOG, OCT4 and SOX2 (Supplementary Fig. 6b); specific expression of the trophoblast markers TFAP2C and GATA3 (Supplementary Fig. 6d); and specific expression of the hypoblast marker GATA4 and SOX17 (Supplementary Fig. 6c). We also subjected some of these stem cell lines to differentiate assess their developmental potentials. TSCs were successfully differentiated into syncytiotrophoblast (ST)-like cells, as observed by immunofluorescence of positive staining for SDC1 (Supplementary Fig. 6e). Besides, HLA-G and CGB, an extravillous cytotrophoblasts marker, were detected in blastoids-derived cells (Supplementary Fig. 6f–g). Altogether, these results indicate that monkey blastoids are capable of deriving stem cell lines with blastocyst lineage identities.

### Blastoids for implantation modeling in vitro
To evaluate whether further in vitro culture can drive the self-organization of these blastoids into peri-implantation or post-implantation embryo-like structures, we adopted an in vitro attachment assay reported previously[34] and performed 2D culture of blastoids to monitor their morphological changes in an additional IVC medium (Supplementary Fig. 6h). After attachment, the blastoids flattened, expanded, and progressed to form outgrowth structures (Supplementary Fig. 6i) resembling the changes observed in monkey blastocysts as well as blastoids derived by other protocols cultured in IVC[8,45]. We also detected an increased level of monkey Chorionic Gonadotropin (mCG) secretion in the IVC medium of attached blastoids since day 4 of the in vitro culture, while no mCG secretion could be observed in blank IVC medium (Supplementary Fig. 6j). In addition, we confirmed that blastoids derived from M-iPSCs and M-nt-ESCs exhibited similar states in the IVC culture system, comparable to the M-ESCs-derived, demonstrating consistent developmental potential (Supplementary Fig. 6k–m and 6p–s).

We transplanted the blastoids into surrogate mother monkeys and observed an increase in mCG levels (Supplementary Fig. 6n, o), but there was no substantial evidence of further development in vivo.

### Synthesis and performance of hydrogel to encapsulate monkey blastoids
To further expand the development of blastoid applications, we aimed to encapsulate the blastoids into hydrogel microspheres via a microfluidic emulsion technique. Droplet-based microfluidic systems generate, manipulate, and control sub-microlitre droplets enclosed within an immiscible carrier fluid[46]. The utilization of microfluidics together with synthetic hydrogel has allowed the development of 3D organoid models with higher physiological relevance in a controllable and

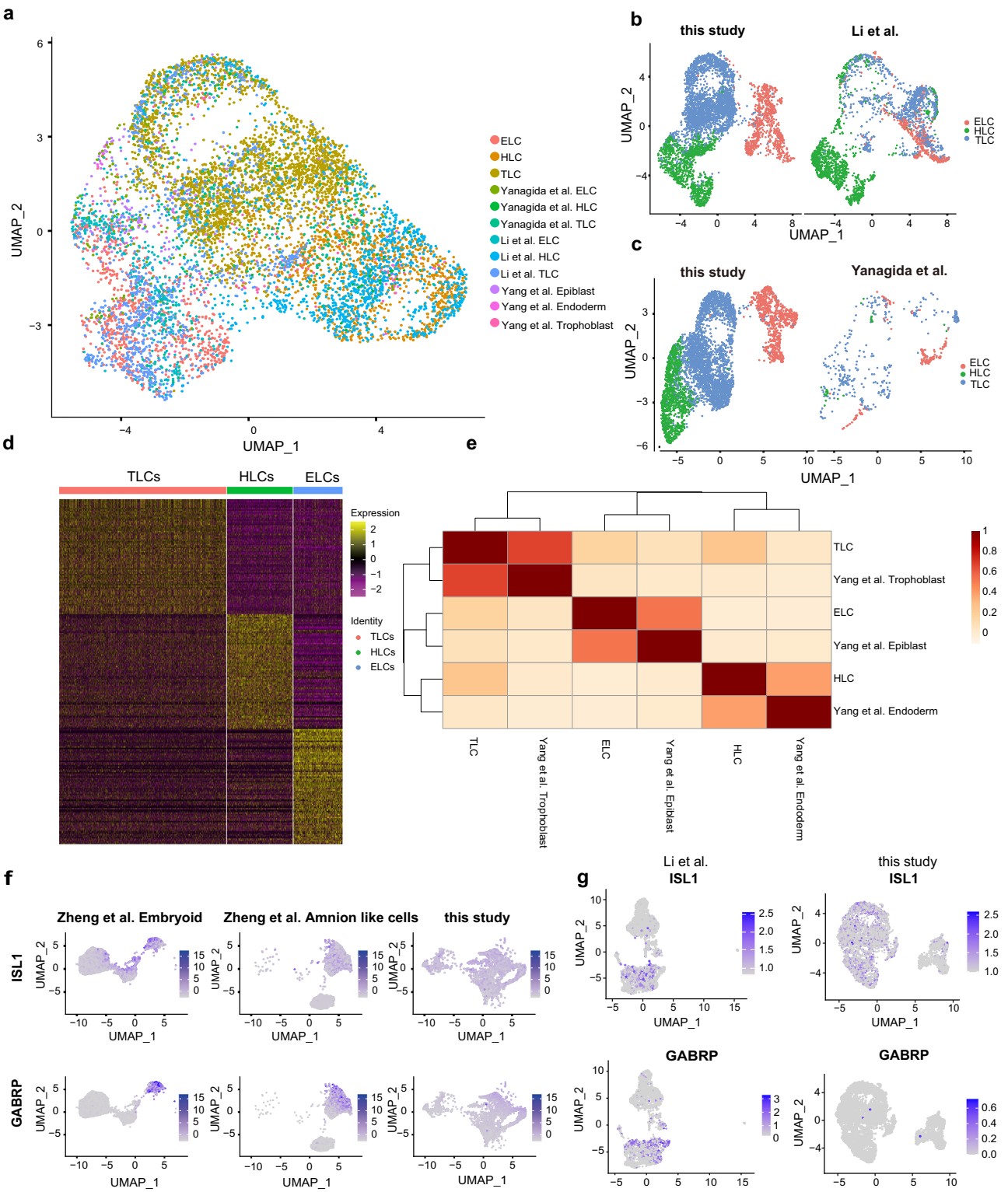

**Fig. 3 | Single-cell transcriptional profiling of monkey blastoids. a** Joint uniform manifold approximation and projection (UMAP) embedding of single-cell transcriptomes of monkey blastocysts, blastoids, and related datasets. **b** UMAP of single-cell transcriptomes of monkey blastoids and published monkey blastoids datasets. **c** UMAP of single-cell transcriptomes of monkey blastoids and monkey pre-implantation embryos. **d** Heatmap showing the ELC, TLC, and HLC signatures in the various clusters. **e** Correlation analysis of various clusters in blastoids with natural cynomolgus monkey embryos. **f** Amnion markers (*ISL1* and *GABRP*) expression patterns. **g** UMAP of clusters formed from blastoid in this study and published monkey blastoids datasets, showing the expression levels of *ISL1* and *GABRP*.

scalable manner[47]. Hence, hydrogel encapsulation should provide the constructed blastoids with a more physiological extracellular matrix (ECM) protective environment and attain high monodispersity and scalability.

We first synthesized light curable hydrogel (GelMA) based on gelatin (Fig. 6a). In the comparison of nuclear magnetic hydrogen spectroscopy (¹H-NMR) of GelMA and Gelatin: -C = CH2 group peaks of GelMA verified the successful conjugation of MA on Gelatin

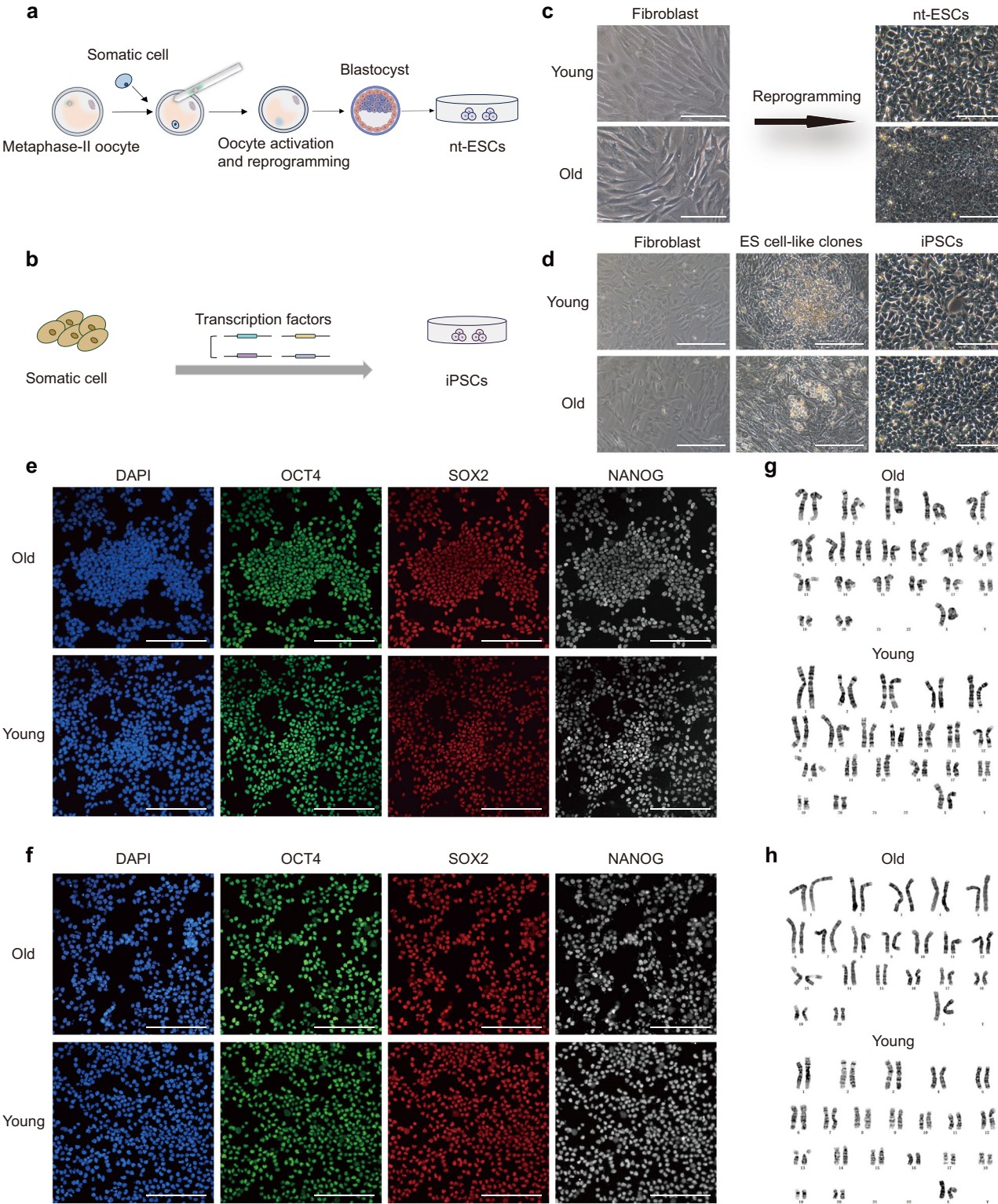

**Fig. 4 | Establishment of nt-ESC lines and iPSC lines from young and old monkeys. a** Experimental scheme for the establishment of young and old M-nt-ESCs from monkey day 6 in vivo fertilization embryos. **b** Experimental scheme for the establishment of young and old M-iPSCs from fibroblast. **c, d** Morphology of cells during establishment of young and old M-nt-ESCs (3 biological replicates for young and 3 biological replicates for old), M-iPSCs (2 biological replicates for young and 2 biological replicates for old). Scale bars, 100 μm. **e, f** Representative immunostaining images of pluripotency markers in old and young M-nt-ESCs and M-iPSCs. Scale bars, 100 μm. **g, h** Karyotype analyses of old and young M-nt-ESCs (3 biological replicates for young and 3 biological replicates for old) and M-iPSCs (2 biological replicates for young and 2 biological replicates for old).

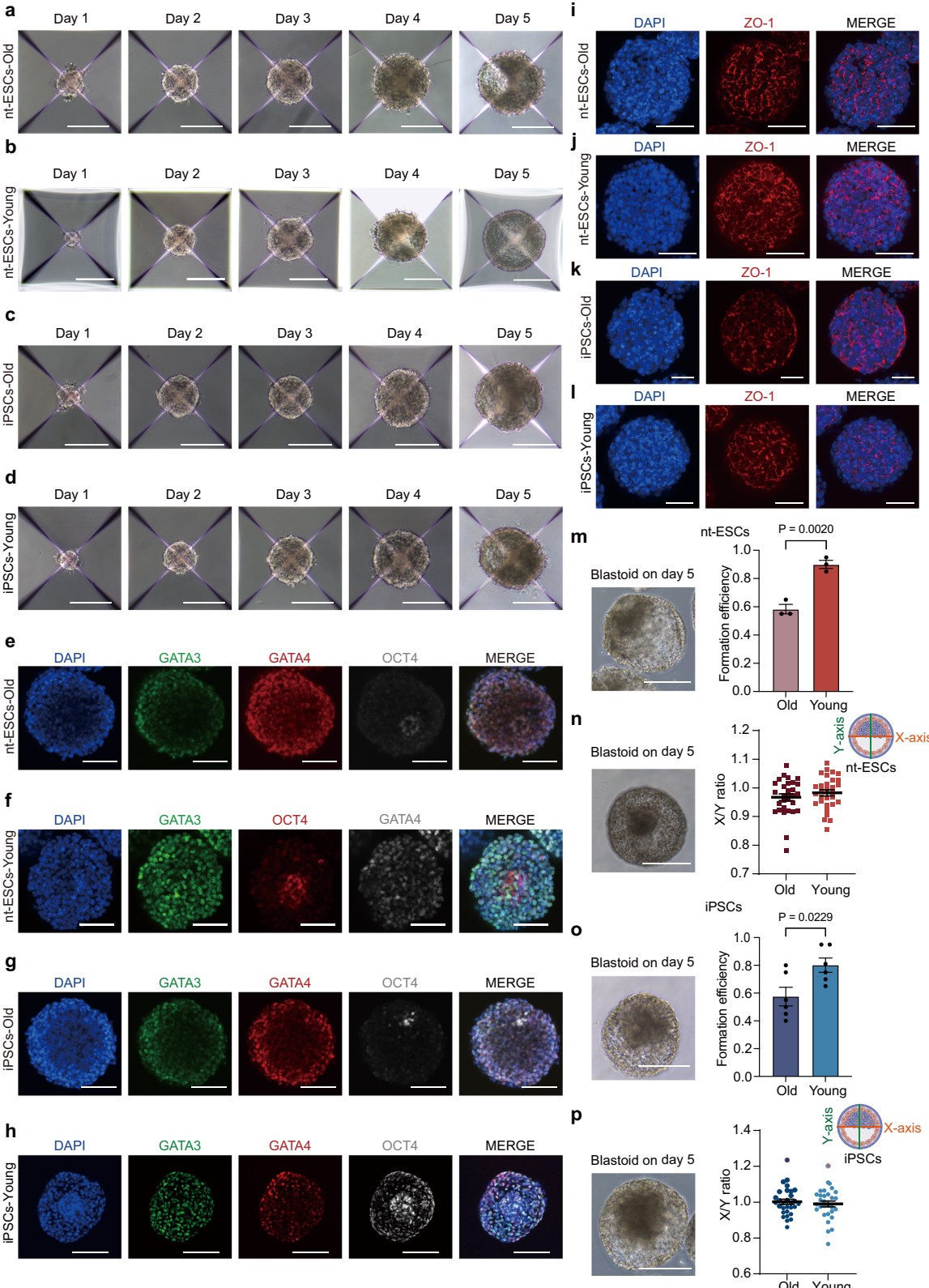

(Supplementary Fig. 7a). To verify the curability of the hydrogel, we tested the hydrogel using a rheometer. The hydrogel precursor solution was irradiated with a blue light at 405 nm wavelength, and it was found that the energy storage modulus of the hydrogel was higher than the loss modulus after light illumination (Supplementary Fig. 7b). The hydrogel precursor solution changed from liquid to solid, and further underwent an elastic deformation during shear loading. The

verification of light curing performance laid the foundation for the combination of hydrogel and droplet microfluidic emulsion techniques.

We further analyzed the degradation performance of hydrogel capsules in vitro at 37 °C, and finally, only 10% of the residue remained at 42 days (Supplementary Fig. 7c). Since hydrogel capsules had a sponge-like 3D network structure which consisted of hydrophilic

**Fig. 5 | Generation of monkey blastoids from M-nt-ESCs and M-iPSCs.**
**a–d** Representative phase-contrast images of old and young nt-ESCs and iPSCs aggregates at indicated time points during blastoid formation. Scale bars, 200 μm. **e–h** Representative immunofluorescence images of GATA3, OCT4, and GATA4 in monkey blastoids from old and young nt-ESCs, old and young iPSCs. Scale bars, 100 μm. **i–l** Representative immunofluorescence images of ZO-1 in monkey blastoids from old and young nt-ESCs, old and young iPSCs. Scale bars, 100 μm. **m** Representative phase-contrast image of nt-ESC blastoids on day 5 (left) and blastoid formation efficiency (right) of M-nt-ESCs (*n* = 3 biological replicates for old and *n* = 3 biological replicates for young). Scale bars, 200 μm. Error bars, mean ± S.E.M, unpaired two-tailed *t* test. **n** Representative phase-contrast image of

blastoids on day 5 (left) and measurement of the x/y ratio of blastoids from nt-ESCs (*n* = 30 technical replicates for old and *n* = 30 technical replicates for young). Scale bars, 200 μm. Error bars, mean ± S.E.M, unpaired two-tailed *t* test. **o** Representative phase-contrast image of blastoids on day 5 (left) and blastoid formation efficiency of monkey iPSCs (*n* = 2 biological replicates and 3 technical replicates for old and *n* = 2 biological replicates and 3 technical replicates for young). Scale bars, 200 μm. Error bars, mean ± S.E.M, unpaired two-tailed *t* test. **p** Representative phase-contrast image of blastoids on day 5 (left) and measurement of the x/y ratio of blastoids from monkey iPSCs (*n* = 30 technical replicates for old and *n* = 30 technical replicates for young). Scale bars, 200 μm. Error bars, mean ± S.E.M, unpaired two-tailed *t* test.

polymers groups, they were capable of imbibing large amounts of fluid. After incubation in deionized water for 10 h, the volume of hydrogel capsules expanded three-fold and then remained steady (Supplementary Fig. 7d).

To evaluate the hydrogel's biocompatibility, we co-cultured M-ESCs, M-iPSCs, and M-nt-ESCs respectively with the hydrogel. We established a co-culture system between hydrogel and cells by using a transwell apparatus and performed a live/dead assay to assess the impact of hydrogel on cell viability and proliferation. The vast majority of cells survived after 3 days of cultivation (Fig. 6b–e and Supplementary Fig. 7e–h), demonstrating good compatibility of the hydrogel with M-ESCs, M-iPSCs, and M-nt-ESCs. Furthermore, we observed that the monkey blastoids were able to proliferate in the hydrogel after encapsulation (Fig. 6f). This confirms that our synthetic hydrogel has good compatibility with monkey blastoids.

### High throughput preparation of monkey blastoid capsules

Via microfluidic emulsion technique, we generated blastoid-encapsulated hydrogel microspheres (blastoid capsules) in a monodisperse, consistent, and reproducible manner (Fig. 7a). We evaluated the morphology of empty hydrogel capsules and blastoid capsules (Fig. 7b–d and Supplementary movie 1). The droplet microfluidic device can prepare hydrogel capsules with different diameters by adjusting the oil phase rate. The maximum particle size is 550 μm, and the minimum particle size is 260 μm (Fig. 7e). By adjusting the oil phase velocity to 75 mL/hour in the microfluidic device, we successfully fabricated hydrogel capsules with an average particle size of 227 μm (Fig. 7f, g). We next calculated the blastoid encapsulation efficiency of the high-throughput microfluidic device, and we can produce up to 4000 blastoid capsules per hour, making it possible to produce blastoid capsules in a scalable manner (Fig. 7h). The microfluidic device was utilized to achieve high-flux encapsulation of monkey blastoids by adjusting the oil phase velocity to 75 mL/hour, resulting in blastoid capsules with an average particle size of 266 μm (Fig. 7i, j). Therefore, the construction of blastoid capsules by droplet microfluidic technique maintains a stable output of relatively uniform particle size.

To investigate the stability of the blastoid capsules, we assessed their delivery rates through NHPs fallopian tubes, and into the uterus of NHPs. First, we detected the degradation rate of the hydrogel in vitro, which simulated the degradation environment in vivo through added enzymes to the solution. The results demonstrate that the capsules almost completely degrade within 42 h (Supplementary Fig. 8a). The GelMA-FITC hydrogel, formed by grafting the fluorescent group FITC onto GelMA, emits green fluorescence when excited at a wavelength of 494 nm. We used GelMA-FITC hydrogel to visually observe the biodegradation of the hydrogel (Supplementary Fig. 8b). As the capsules degrade in vitro, the encapsulated blastoids are released and begin to exhibit outgrowth (Supplementary Fig. 8c). Given the protective effect the hydrogel provides on monkey blastoids, we explored whether the hydrogel could aid in the cryopreservation of blastoids[48]. Traditional embryo cryopreservation typically

involves storing embryos individually or in small batches using a straw[49]. However, this method severely limits their storage and transport possibilities. Here, we show that monkey blastoids that were not encapsulated in hydrogel and subjected to the same cryopreservation process exhibited extensive cell death in a short time (Supplementary Fig. 8d).

The blastoid capsules had high delivery rates in NHPs fallopian tubes (Fig. 8a). Our platform provides stable and scalable delivery of monkey blastoids, which can prevent the occurrence of adhesion between multiple blastoids and between blastoids and tissues, which may be generalizable to other organoids (Fig. 8b). After successful delivery of blastoid capsules to NHPs uteri, the blastoid capsules were collected and immunofluorescence stained. The hydrogel did not significantly impact the structure of the monkey blastoids, even 6 h after being delivered into the monkey uterus (Fig. 8c).

## Discussion

Although research on monkey blastoids is still limited, Liu et al.[8]. recently have reported the successful construction of monkey blastoids using ESCs with an efficiency of 30%. In this work, by optimizing the blastoid construction system, the efficiency of generating monkey blastoids is increased to 80%. In addition, the NHPs blastoids reported to date were generated from M-ESCs. Here we demonstrate that monkey blastoids can be effectively obtained using M-iPSCs and M-nt-ESCs from both young and aged donors, as well as M-ESCs. Similarly, we demonstrate that this method is not only efficient in constructing blastoids from M-ESC (~ 80%), but we can also adapt the method for the efficient generation of blastoids from aged M-nt-ESCs (~ 60%).

Rapid organoid modeling with high reproducibility and consistency is in high demand to advance the translational applications of organoids. For the first time, we introduce an integrative platform for the fabrication of NHPs blastoid capsules from reprogrammed aged somatic cells by using a home-made droplet microfluidic system that enables size-adjustable, monodisperse, and high-throughput generation (up to 4000 units per hour) of NHPs blastoid capsules. The monkey blastoid capsules also effectively delivered through the monkey fallopian tube when compared with the unencapsulated blastoids. Our high-throughput production of NHPs blastoid capsules may offer a robust and scalable platform for organoid research and regenerative medicine.

Extending blastoid culture in vitro to the gastrulation or organogenesis stages represents a critical focus for future research on modeling embryonic development using blastoids. This approach holds significant promise for comprehensively exploring the developmental potential of blastoids. However, achieving a long-term culture of monkey blastoids remains a substantial challenge. Advancements in extended blastoid culture have been reported, but current techniques still require further refinement to better mimic the developmental processes of natural embryos. Notably, recent studies have revealed the influence of different extracellular matrices on extended blastoid culture[6], offering directions for improving culture systems. In the future, optimizing culture conditions and developing original

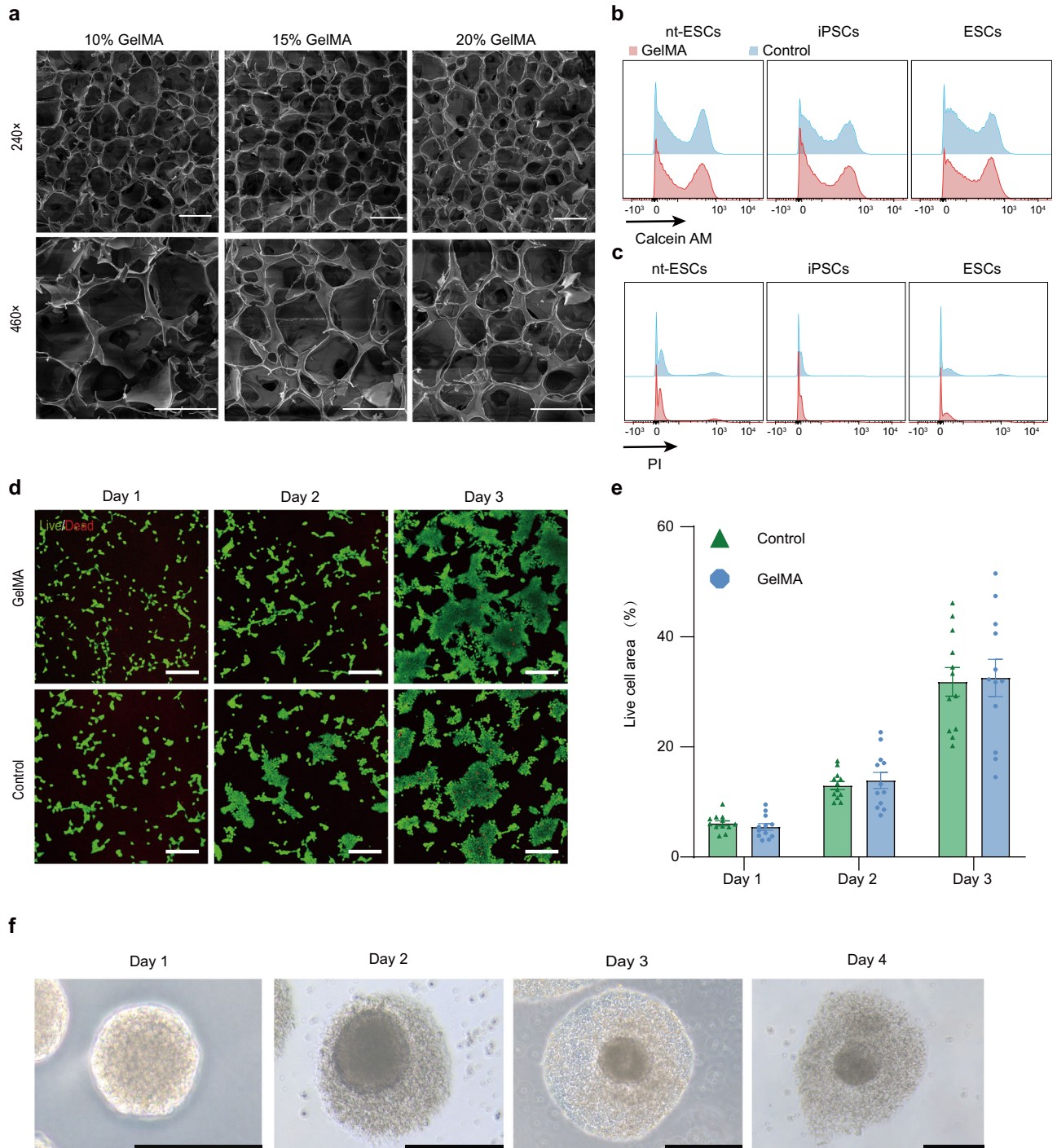

**Fig. 6 | Verification of hydrogel morphology and biological properties.**
**a** Scanning electron microscopy images of different concentrations of GelMA (3 independent replicates). Scale bars, 100 μm. **b**, **c** Flow cytometry analysis of live (top)/dead (bottom) ratio for M-nt-ESCs, M-iPSCs, and M-ESCs in control, GelMA groups on day 3 cultured. **d**, **e** Representative immunofluorescence staining images of live (green)/dead (red) cells and the live cell area for M-ESCs in control, GelMA groups on day 1, day 2, and day 3 cultured (n = 3 biological replicates and 4 technical replicates), statistical analysis of green area (**e**). Error bars, mean ± S.E.M. Scale bars, 200 μm. **f** Cultivating blastoids on the hydrogel. Scale bars, 200 μm.

materials may enable the successful in vitro culture of primate blastoids to the organogenesis stage. This progress would provide a valuable tool for elucidating primate embryonic development mechanisms and open avenues for disease research and drug screening.

Here, we present a method for the efficient formation of blastoids from monkey aged somatic cells, yet limitations remain that

necessitate further research. The process of creating blastoids involves inducing monkey PSCs into TLCs and HLCs under non-physiological conditions, using a variety of chemicals and growth factors[27,50,51]. Therefore, the primary limitation in the current work is that the quality and functionality of the trophoblast cells in the constructed blastoids still need to be further improved. This is crucial, as their performance directly impacts the success of forming a functional placental interface

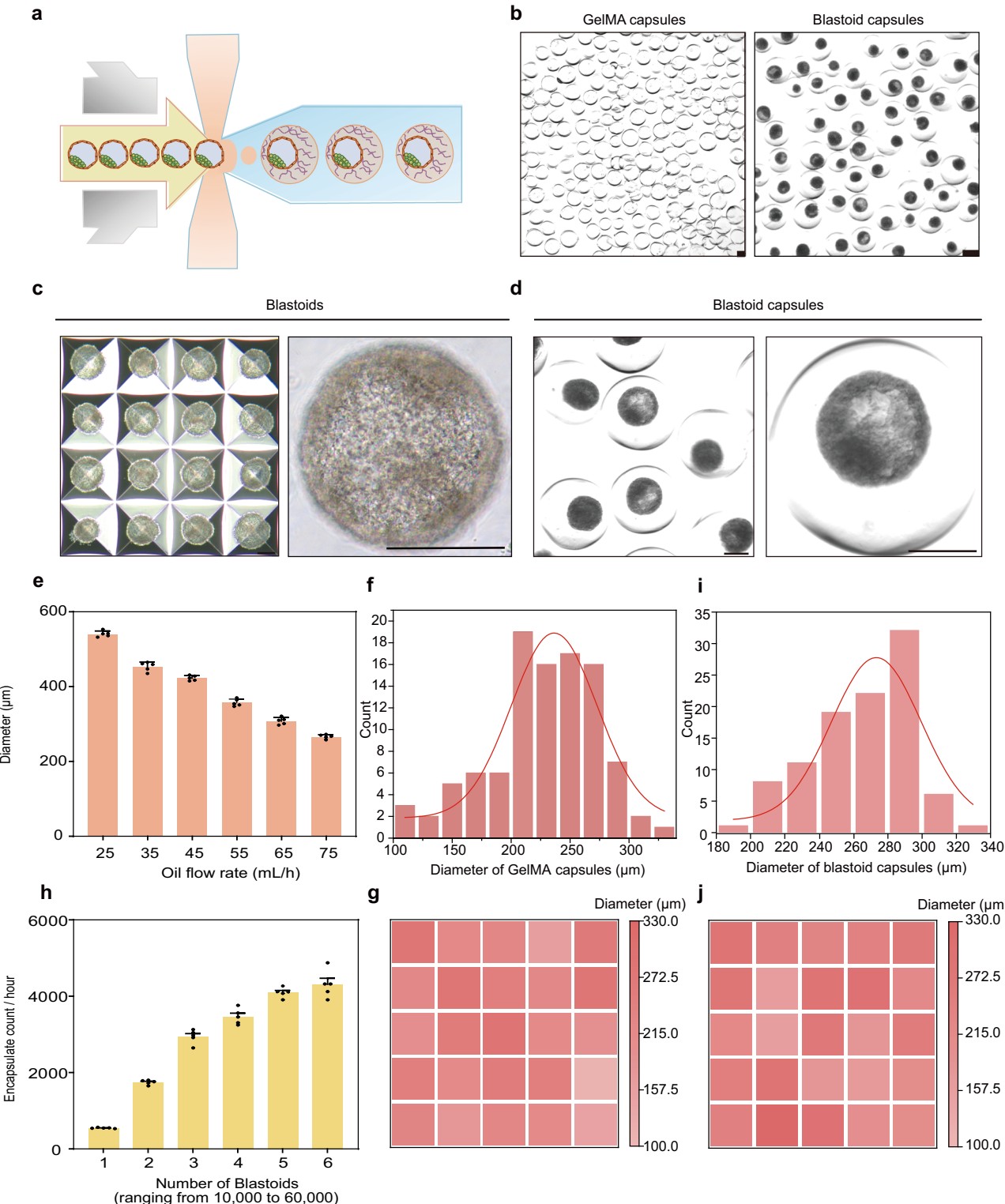

**Fig. 7 | High throughput preparation of blastoid capsules. a** Diagram of droplet generation and blastoid encapsulation. **b** Optical images show the capsules generated (left) after the use of the droplet microfluidic technique and the capsules containing blastoids (right) (20 independent replicates). Scale bars, 200 μm. **c**, **d** Representative phase-contrast images of blastoids and blastoid capsules (20 independent replicates). Scale bars, 200 μm. **e** Diagram of the relationship between oil velocity and particle size of capsules in the process of micro-flow control, *n* = 5 biological replicates. Error bars, mean ± S.E.M. **f** Size distribution of the resulting GelMA capsules (*n* = 100 capsules). **g** Heat map of size distribution of GelMA capsules obtained (*n* = 25 capsules). **h** The encapsulation efficiency of blastoid capsules prepared by microfluidic control, *n* = 5 biological replicates. Error bars, mean ± S.E.M. **i** Size distribution of the resulting blastoid capsules (*n* = 100 capsules). **j** Heat map of the size distribution of blastoid capsules obtained (*n* = 25 capsules).

**a**

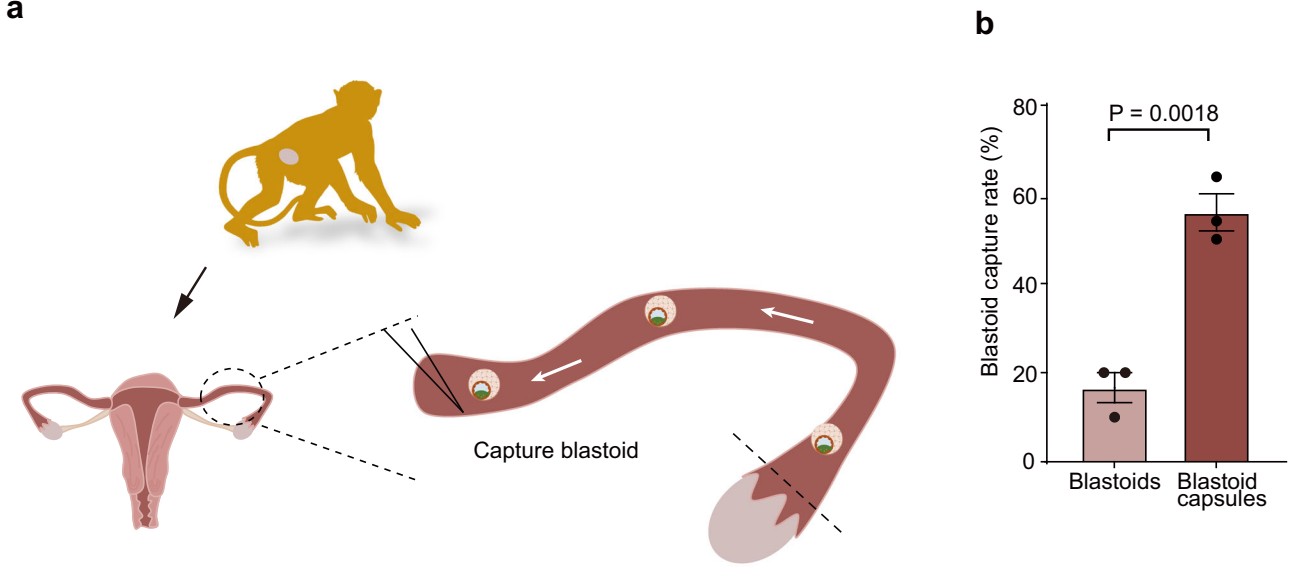

**b**

**c**

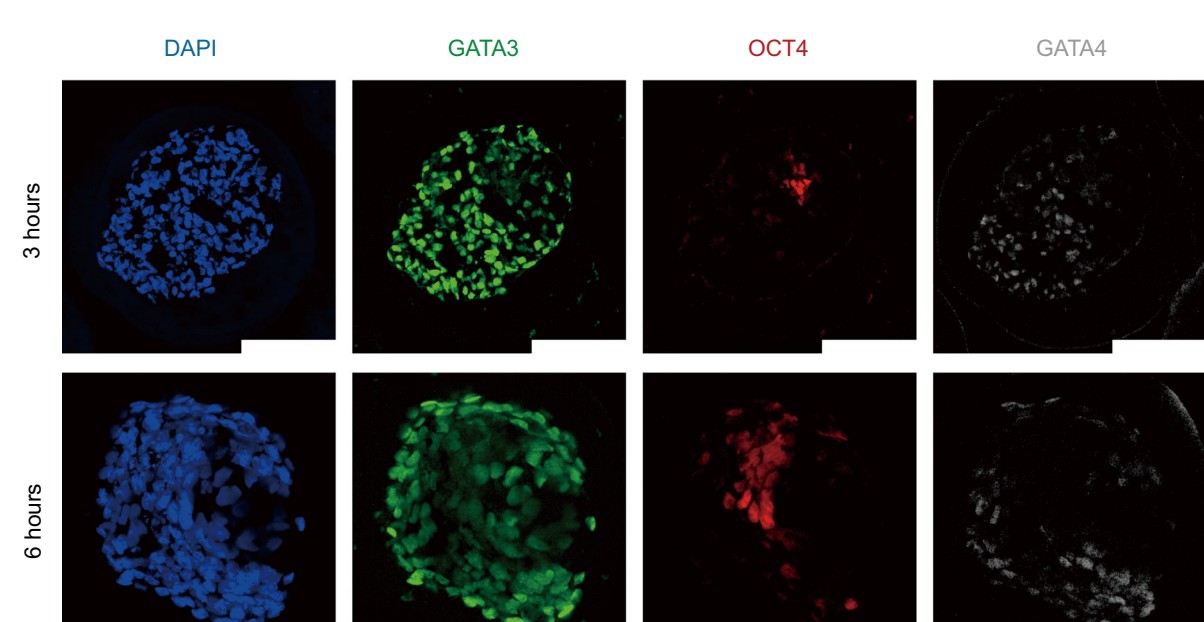

**Fig. 8 | Deliverability of blastoid capsules. a**, **b** The scheme and the capture rate of blastoids and blastoid capsules in the monkey fallopian tubes, $n = 3$ biological replicates. Error bars, mean ± S.E.M. unpaired two-tailed $t$ test. **c** Representative images of immunofluorescence staining at 3 and 6 h after the blastoid is loaded into a capsule. Scale bars, 200 μm.

in vivo[5,52,53]. Addressing these deficiencies will involve not only refining cell culture techniques but also potentially integrating advanced genetic and bioengineering methods to enhance cellular responses and integration[6]. As such, in vivo transplantation of monkey blastoids, despite inducing increased mCG levels, did not result in further development. Improving the outcome of in vivo transplantation of monkey blastoids from aged somatic cells is one of our goals for future studies, as this could potentially offer invaluable insights into the integration and function of blastoids within the organism, simulating more closely the natural processes of embryo development.

## Methods

### Ethics

All animals were housed at the State Key Laboratory of Primate Biomedical Research (LPBR). Every experiment involving animals have been carried out following a protocol approved by an ethical commission. All animal and experiment procedures were approved by the ethical committee of the LPBR and performed by following the guidelines of the Association for Assessment and Accreditation of Laboratory Animal Care International (AAALAC) for the ethical treatment of NHPs. All animal procedures were approved in advance by the

State Key Laboratory of Primate Biomedical Research (no. LPBR202104016 and PZWH (YN) K2023-0022). Healthy male and female cynomolgus monkeys and rhesus monkeys, ranging in age from 1 to 25 years, were selected for use in this study. The monkeys were housed with a 12 h light/dark cycle between 6:00 and 18:00 in a temperature-controlled room (22 ± 1 °C) with free access to water and food. The mouse were housed with a 12 h light/dark cycle between 6:00 and 18:00 in a temperature-controlled room (22 ± 1 °C) with free access to water and food.

### Cell cultures
All cells were cultured without antibiotics in humidified incubators at 37 °C in 5% $CO_2$ and maintained in atmospheric oxygen. Cell lines were tested negative for mycoplasma by periodic PCR screening.

### Primary monkey fibroblast isolation and culture
In brief, the rhesus monkey ear skin samples were sterilized with 75% ethyl alcohol and washed with PBS. After removing the hair and fat tissues, the samples were cut into pieces. These pieces were then adhered to the culture dish. Fibroblasts were cultured in DMEM (Thermofisher), supplemented with 10% fetal bovine serum (FBS, Thermofisher) and 1% penicillin/streptomycin (Thermofisher). The Fibroblast-312 and the Fibroblast-050 were derived from female cynomolgus monkeys aged 15 years old. The Fibroblast-418 and the Fibroblast-058 were derived from female rhesus monkeys aged 22 years old. The Fibroblast-372 was derived from female rhesus monkeys aged 25 years old. The Fibroblast-182004 and the Fibroblast-040 were derived from female cynomolgus monkeys aged 1 years old. The Fibroblast-004, the Fibroblast-006, and the Fibroblast-446 were derived from female rhesus monkeys aged 5 years old. In this study, M-iPSCs were derived from Fibroblast-312, Fibroblast-050, Fibroblast-182004 and Fibroblast-040; M-nt-ESCs were derived from Fibroblast-418, Fibroblast-058, Fibroblast-372, Fibroblast-004, Fibroblast-006, Fibroblast-446. In the experiments, none of the monkey underwent genetic modifications such as gene editing.

### Culture of monkey PSCs
All monkey ESCs (XF-ESC-1, XF-ESC-2, XF-ESC-3, CES1_1-XF PSC and CES_N from Wu et al.'s report), nt-ESCs (XF-nt-ESC-004, XF-nt-ESC-006, XF-nt-ESC-446, XF-nt-ESC-418, XF-nt-ESC-058 and XF-nt-ESC-372) and iPSCs (XF-iPSC-182004, XF-iPSC-040, XF-iPSC-312 and XF-iPSC-050) were cultured on Vitronectin XF (STEMCELL). The plates were incubated at 37 °C for at least 1.5 h or 4 °C overnight. Do not allow the culture surface to dry. PSCs were cultured in the chemically defined medium under atmospheric oxygen and 5% $CO_2$ at 37 °C. A total of 50 mL XF-PSC medium was prepared by including TeSR™-E8™ Medium (Thermofisher), 1× GlutaMAX (Thermofisher), 1× Chemically Defined Lipid Concentrate (Thermofisher), 10 ng/mL Activin-A (PeproTech), 100 ng/mL Recombinant Human Nodal Protein (biotechne), 2 μM IWR-1-endo (Selleck) and 1.94 mg/L L-Glutathione (Sigma). The cells were passaged with Accutase (Thermofisher) once they reached around 70–80% confluency (usually every 3-4 days).

### Monkey SCNT
The MII oocytes were treated with TH3 containing 5 μg/mL cytochalasin B at 37 °C for 5 min, followed by the removal of the spindle using the spindle viewer system. After a brief incubation of donor fibroblasts with Sendai virus (GenomONE), the donor cells were inserted into the perivitelline space of the enucleated oocytes using a laser perforation system. The fusion was completed after 1.5 to 2 h. For chemical activation, the reconstructed oocytes were treated with HECM-9 containing 5 μM calcium ionophore for 5 min, followed by incubation in HECM-9 containing 2 mM 6-dimethylaminopurine (Sigma) and 10 nM trichostatin A (Sigma) for 5 h. After being activated for 6 h, the mRNA of H3K9me3 demethylase KDM4D (1000 ng/μL, 10 pl) was injected.

The SCNT embryos were cultured to the blastocyst stage and utilized for establishing ESC lines.[15,16]

### Generation and culture of iPSCs and nt-ESCs from primary fibroblasts
The iPSC lines were generated using the Sendai Reprogramming Kit (Thermofisher) according to the manufacturer's instructions. Subsequently, the iPSCs were cultured using an established feeder layers culture system[26]. The SCNT blastocysts at d.p.f.7 were transiently treated with 0.5% protease to remove the zona pellucida, and cultured on feeders in XF-PSC medium supplemented with 10 μM Y27632 (Selleck). The nt-ESCs were cultured as same as iPSCs[26].

### Karyotype analysis
Cells were collected at a density of 60–80% of confluence on the day of sampling. After 2 h of incubation with fresh medium, Colcemid (Sigma) solution was added to the culture at a final concentration of 0.02 mg/mL. Then the cells were incubated for 1 h. After incubation, cells were washed, digested, and centrifuged. To obtain a single-cell suspension, the pellet was re-suspended in hypotonic solution (0.56% KCL, Sigma), and incubated at room temperature for 6 min. After centrifuging and removing the hypotonic solution, 5 mL of ice-cold fixative (3:1 (v/v) mixture of methanol and acetic acid) was added to the suspension in a dropwise manner. Then the cells were incubated at room temperature for 5 min before spinning down. The fixing procedure was repeated for additional three times. Afterward, the pellet was resuspended in a final volume of 1 mL fixative. Then, the cells were dropped onto 5% acetic acid ± ethanol (ice-cold) washed slides and stained with Giemsa. For each experiment, 30–40 metaphases were analyzed. The number of chromosomes and the presence of structural chromosomal abnormalities were examined.

### Teratoma assay
Approximately $5 \times 10^6$ PSCs were suspended in 50 μL XF-PSC medium, and mixed with the same volume of Matrigel (Corning) (thawed before the experiment on ice). The cell mixture was subcutaneously injected into immunodeficient NOD/SCID mice. Teratomas developed within 4–8 weeks. The teratomas were isolated and embedded in paraffin, which were processed for hematoxylin and eosin staining.

### Generation of monkey blastoids
For the generation of monkey blastoids, 60–70% confluent monkey PSCs were dissociated into single cells by incubation with Accutase at 37 °C for 3 min. Cells were collected and resuspended in XF-PSC medium after centrifugation at 200 × g for 5 min, and cells were counted. Meanwhile, the AggreWell™400 (STEMCELL) was prepared according to the manufacturer's instructions. In brief, wells were rinsed with anti-adhesion rinsing solution (STEMCELL), centrifuged at 2000 × g for 5 min and incubated at room temperature for 10 min. After incubation, wells were washed with XF-PSC medium once.

Approximately 100,000 cells were resuspended in 1 mL XF-PSC medium with 5 μM DDD00033325 (Sigma), and then 1× CloneR (STEMCELL) was added, and the mixture was seeded into one well of a prepared AggreWell™400 24-well plate. Aggregates formed after around 12–18 h of culture, and the medium was replaced with HDM. The day of switching from XF-PSC medium to differentiation medium (HDM) was designated as day 1. On day 3 (High-passage cells require the fourth day), we carefully removed as much HDM as possible and then added 1 mL of TSG. This step was repeated once to help completely remove the remaining HDM. TSG was replaced with fresh TSG every two days. After three to five days of culture in TSG (the time may vary between different cell lines, different starting monkey PSC culture conditions, different starting cell numbers, and different batches), monkey blastoids can be observed in some microwells.

HDM was prepared using the following: 1:1 (v/v) mixture of DMEM/F12 and neurobasal medium, 1 × N2 supplement (Thermofisher), 1 × B27 supplement (Thermofisher), 1 × GlutaMAX, 1 × nonessential amino acids (Thermofisher), 0.1 mM β-mercaptoethanol (Sigma), 20 ng/mL bFGF (Proteintech), 20 ng/mL Activin-A and 3 μM CHIR99021 (Selleck).

TSG was prepared using the following: 1:1 (v/v) mixture of DMEM/F12 and TeSR™-E8™ Medium, 0.5 × ITS-X (Thermofisher), 0.05 mM β-mercaptoethanol, 0.5% knockout serum replacement (KSR, Thermofisher), 0.1% FBS, 50 mg/mL bovine serum albumin (BSA, Sigma), 1 μM PD0325901(Selleck), 0.5 μM A83-01 (STEMCELL), 1 μM CHIR99021, 0.5 μM SB431542(Selleck), 25 ng/mL EGF (Peprotech), 0.75 μg/mL L-ascorbic acid (Wako), 0.4 mM VPA (STEMCELL), 0.5 × GlutaMAX, 0.5 × Chemically Defined Lipid Concentrate, 5 ng/mL Activin-A, 50 ng/mL Recombinant Human Nodal Protein, 1 μM IWR-1-endo and 0.97 mg/L L-Glutathione.

## Quantification of blastoid formation efficiency

Cell aggregates were collected from microwells by gently pipetting up and down 2-3 times with a 1 mL pipette. To minimize the shearing force, around 3–5 mm from the end of the 1 mL pipette tips were cut off before use. All of the cell aggregates were transferred into a well of a six-well plate and counted under a stereomicroscope. The aggregates with the presence of an ICM-like compartment, a round ball shape, and a visible cavity were counted as blastoids. The blastoid formation efficiency was calculated as the number of blastoids per number of total aggregates.

## Immunofluorescence analysis

For immunostaining, cells and monkey blastoids were fixed on plates in 4% PFA (Biosharp) for 120 min at room temperature, permeabilized in PBS containing 1% Triton X-100 for 120 min at room temperature, and then incubated with blocking buffer (PBS containing 4% BSA) for 60 min. After incubation, primary antibodies were incubated overnight at 4 °C followed by secondary antibodies at room temperature for 2 h or overnight at 4 °C. The antibodies used were as follows: OCT4 antibody (Santa Cruz Biotechnology), GATA3 antibody (Santa Cruz Biotechnology), GATA4 antibody (R&D Systems), TUJ1 antibody (Abcam), AFP, α-SMA antibody (Abcam), CDX2 antibody (Abcam), GATA6 antibody (R&D Systems), ZO-1 antibody (Thermofisher), Calcein-AM/PI Double Stain Kit (YESEN), Donkey anti-rabbit IgG (H + L) 647 (Invitrogen), Donkey anti-rabbit IgG (H + L) 488 (Invitrogen), Donkey anti-mouse IgG (H + L) 555 (Invitrogen), Donkey anti-goat IgG (H + L) 647 (Invitrogen), Donkey anti-rabbit IgG (H + L) 555 (Invitrogen).

## Derivation of cell lines from monkey blastoids

Derivation experiments were performed with blastoids following the reported protocols with some modifications[26,54,55]. Individual monkey blastoids were plated onto 24-well plate coated with feeder layers and cultured in XF-PSC (for blastoid ESCs), NACL medium (for blastoid nEnd) or TS medium (for blastoid TSCs). Within one week, blastoids attached and outgrowths could be observed. Outgrowths were dissociated with Tryple and passaged onto newly prepared feeder layers plates. After 24 h of culture on feeders, colonies were formed.

The NACL medium contained 1:1 (v/v) DMEM/F12 and Neurobasal with 1x N2 supplement, 1x B27 supplement, 1 x GlutaMAX, 1 x nonessential amino acids, 0.5% penicillin/streptomycin, 0.1 mM 2-mercaptoethanol, 100 ng/mL Activin A, 3 μM CHIR99021 and 10 ng/mL LIF.

The TS medium was prepared following: DMEM/F12 supplemented with 0.1 mM 2-mercaptoethanol, 0.2% FBS, 0.5% Penicillin-Streptomycin, 0.3% BSA, 1% ITS-X supplement, 1.5 μg/mL L-ascorbic acid, 50 ng/mL EGF, 2 μM CHIR99021, 0.5 μM A83-01, 1 μM SB431542, 0.8 mM VPA, 5 μM Y27632.

## In vitro differentiation blastoid-derived TSCs to EVT and SCT like cells

Differentiation of blastoid TSCs followed reported protocols[55]. For EVT and SCT differentiation, TSCs were dissociated with TrypLE for 5 min at 37 °C and seeded at a density of 4,000 cells per well onto a 48-well plate pre-coated with collagen IV. For differentiation of EVT-like cells, cells were cultured in 0.2 mL of EVT medium supplemented with 100 ng/mL NRG1 and 2% Matrigel. After three days of differentiation, the medium was replaced without NRG1 and reduced Matrigel (0.5%) for an additional three days of cell culture. Cells were cultivated for another 2 days before analysis. To generate SCT-like cells, cells were cultured in 0.2 mL of SCT medium until day 6. The medium was refreshed every other day.

The EVT medium comprised DMEM/F12 supplemented with 0.1 mM β-mercaptoethanol, 0.5% penicillin–streptomycin, 0.3% BSA, 1% ITS-X, 7.5 μM A83-01, 2.5 μM Y-27632 and 4% KSR.

The SCT medium contained DMEM/F12 supplemented with 0.1 mM β-mercaptoethanol, 0.5% penicillin–streptomycin, 0.3% BSA, 1% ITS-X, 2.5 μM Y-27632, 2.5 μM forskolin and 4% KSR.

## Flow Cytometry

Blastoids were collected under a stereoscope and dissociated into single cells by Accutase. Cells were collected by centrifugation at $200 \times g$ for 5 min. After centrifugation, DPBS was used to perform wash steps in 1.5 mL Eppendorf tubes and the cells were filtered using a 40 μm cell strainer. Centrifuged again, the supernatant was discarded and 4% PFA was add for fixation as sample preparation for immunofluorescence staining. Flow cytometry was performed using the appropriate unstained control and stained groups. Data analyzed by Flow Jo.

## Superovulation and oocyte collection

The procedures for superovulation and oocyte collection were conducted according to the reported method[26]. In brief, Injecting human folli tropin and cholionic gonadotropin into oocyte donor monkeys. Cumulus-oocyte complexes were collected after 32–35 h following rhCG administration by laparoscopic follicular aspiration. Preheating HEPES buffered Tyrode's albumin lactate pyruvate (TALP) medium supplemented with 0.3% BSA place the follicular contents in it. Oocytes were stripped from cumulus cells by exposure to hyaluronidase (0.5 mg/mL) in TALP-HEPES (< 1 min), visually selected metaphase II (MII; existence of the first polar body) oocytes.

## oocyte fertilization in vitro

Intracytoplasmic sperm injection (ICSI) was performed immediately on mature oocytes, using CMRL-1066 medium containing 10% FBS to culture the oocytes until oocytes achievement in vitro maturation. The existence of the second polar body and two pronucleuses confirmed fertilization. Zygotes were cultivated in chemically defined hamster embryo culture medium-9 (HECM-9) containing 10% FBS to allow embryo development.

## Monkey blastoids extended culture

Monkey blastoids were cultured beyond the implantation stages according to the reported method of monkey blastocysts culture[34,56]. In brief, monkey blastoids were manually isolated using capillary, washed twice with PBS and transferred into 8-well ibidi-Treat dishes containing IVC-1 medium. After 24 h, the medium was switched to IVC-2 medium. Almost all blastoids attached to the plate within two days.

IVC-1 medium contained DMEM/F12 supplemented with 20% (v/v) heat-inactivated fetal bovine serum, 2 mM L-glutamine, 1 x ITS-X, 8 nM β-oestradiol, 200 ng/mL progesterone, 25 mM N-acetyl-L-cysteine. The medium of IVC-2 used 30% knockout serum replacement to replace 20% FBS in IVC1, and other components unchanged.

## Quantification of monkey Chorionic Gonadotropin in monkey blastoids extended culture

200 µL of spent medium was collected and stored at −80 °C. The sample monkey Chorionic Gonadotropin (mCG) level in the medium was analyzed by electrochemiluminescence on a Roche Cobas e411. mCG level is reported in mIU/mL.

## Monolayer differentiation of monkey PSCs into TLC and HLC

Monkey PSCs were dissociated into single cells by Accutase at 37 °C for 5 min. Cells were collected by centrifugation at 200 × g for 5 min. After centrifugation, resuspend the cell pellet with XF-PSC medium and replated into a 24-well plate (50,000 cells per well) pre-coated with Matrigel. Y27632 (STEMCELL) or Clone R (STEMCELL) was needed for passage. After cells adhered, TSG or HDM medium was used to replace XF-PSC, and cells were cultured for an additional five days.

## Preparation of hydrogel

Briefly, 20 g gelatin (Sigma) was dissolved in PBS solution at 55 °C. An alkaline solution was used to adjust the pH to 8.5. 60 mL of methacrylic anhydride (MA, Sigma) was slowly added into the solution and allowed to react overnight. The unreacted MA was removed by centrifugation, and the supernatant was dialyzed with a 12 Kda MWCO dialysis membrane for 1 week. We then filtered GelMA through a 0.22 µM syringe filter and froze-dried after filtration. The treated sample was kept at −20 °C to avoid the influence of light and water[57]. The functional degree of GelMA was determined by proton nuclear magnetic resonance ($^1$H-NMR) spectroscopy. We used a rheometer (TA, DiscoveryHR10, USA) to detect the modulus of GelMA during the curing process.

## Scanning electron microscopy

Prepare a photo initiator (LAP) solution with a mass to volume ratio of 0.25% (w/v). Add GelMA to it to prepare a solution of 10% (w/v), 15% (w/v), and 20% (w/v). Photocrosslink the solution under 405 nm blue light irradiation for 15 s. After freeze-drying, the morphology and pore size of GelMA were characterized by scanning electron microscopy (VEGA3-SBH) after sputter coating with gold.

## Verification of biocompatibility of hydrogel

Blastoids cultured for 4 days after reconstruction were planted in GelMA and continuously cultured for 4 days to observe their sustained proliferation. In addition, M-ESCs, M-iPSCs, and M-nt-ESCs were respectively co-cultured with hydrogel for three days, using a Calcein-AM/PI dual staining kit to observe their growth, and the coverage area was calculated.

## Degradation and swelling tests

The GelMA capsules were immersed in PBS containing Type I Collagenase (Macklin) to evaluate degradability. In brief, GelMA capsules (20 mg) were suspended in the Collagenase solution (0.1 mg/mL in PBS, 1 mL, pH = 7.4) and placed in a shaking incubator (20 × g, 37 °C). Every 2 days, the supernatant was replaced with fresh Collagenase solution. At the specified time point, the residual weight of GelMA capsules was measured and was compared to the initial weight. For the swelling test, 2 mg of GelMA capsules were added into the 1.5 mL tube, and the weight of the GelMA capsules and the tube was measured before 1 mL deionized water was added. The pH of the suspension was adjusted to 7.4, and the tube was placed in a shaking incubator at 37 °C and 20 × g. At defined time points, the tubes were centrifuged (600 × g, 3 min) before removing the supernatant. Excess water was blotted using filter paper before weighting the GelMA capsules. The whole procedure was repeated until a constant weight was obtained.

## Fabrication of blastoid capsules

Firstly, we maintained the core flow rate of the inner needle in the microfluidic device and changed the external flow rate of the outer needle to prepare capsules of different sizes. The number of blastoids in the GelMA hydrogel precursor solution was changed to increase the number of blastoid capsules prepared per hour.

In order to prepare microcapsules, a coaxial microfluidic device was assembled. In short, it consists of two micro syringe pumps and an integrated operation panel (Changsha nanoapparatus JDF05). In addition, a 23 G needle (inner needle: outer diameter, 630 µm; Inner diameter, 330 µm) was added and a 17 G needle (outer needle: outer diameter, 1470 µm; Inner diameter, 960 µm) was inserted to assemble into two concentric coaxial nozzles.

Through a coaxial microfluidic system, 15% (w/v) GelMA and 0.25% (w/v) photoinitiator were mixed with blastoids (core flow) and injected from the inner nozzle, and mineral oil was mixed with 5% (v/v) Span 80 (external flow) and injected from the outer nozzle. The flow rates of core flow (2 mL/hour) and shell flow (75 mL/hour) were independently regulated by independent micro-injection pumps. The nozzle tip is connected to a hose to allow GelMA microdroplets and mineral oil to drip from the hose into the collection plate. GelMA microdroplets were cured with a light source (EFL-LS-1601-405 China) for 15 s to form blastoid capsules. Afterward, the blastoid capsules were rinsed 3 times with 37 °C physiological saline, and the blastoid capsules were briefly stored in physiological saline (Hubei Kangsheng Pharmaceutical SUNGSHIM) for subsequent experiments. All steps are carried out in a biosafety cabinet at room temperature, and the preparation parameters remain unchanged unless otherwise specified.

## Optical microscopic morphology of blastoid capsules

The morphology of blastoid capsule samples was analyzed using an inverted microscope (Leica DMIL). For size distribution analysis, a total of 200 microcapsules with at least 3 images were analyzed using size analysis software (ImageJ 1.53 k).

## Delivery efficiency of blastoids and blastoid capsules

To assess the delivery efficiency of blastoid capsules compared to blastoids alone, we conducted experiments in monkeys. Utilizing the same quantity of 10 for each type, blastoids and blastoid capsules were separately introduced into one end of the fallopian tubes using a micropipette. The outputs were then collected from the opposite end. The effluent was carefully analyzed to count the number of intact blastoids and blastoid capsules.

## Single-cell RNA-seq library preparation, sequencing

Select aggregates with a blastocyst-like structure, spherical shape, and visible cavities. The blastoids were treated with Accutase to maintain a single-cell state. Subsequently, the cells were loaded onto the 10x Genomics Chromium system and prepared for library construction following the manufacturer's instructions. Library sequencing was performed by Annoroad Gene Technology (Beijing China).

## Cryopreservation and resuscitation of blastoids and blastoid capsules

To achieve cryopreservation and resuscitation, VITRIFICATION KIT (101) (Cryotech) and WARMING KIT (102) (Cryotech) were used. In brief, blastoids or blastoid capsules were immersed in the equilibration solution for 15 min, and then the equilibration solution was removed, and cryosolution was added twice and immersed for 60 s, the first for 40 s and the second for 20 s. Then, removed cryofluid, placed the blastoids or blastoid capsules in a cryopreservation tube, and rapidly transferred to liquid nitrogen. For resuscitation, it was removed from liquid nitrogen, a thawed solution was added and incubated at 37 °C for 1 min, removed thawed solution, added diluent solution and

incubated at room temperature for 3 min, then cleaning solution was added twice at room temperature for five min each, removed cleaning solution and blastoids or blastoid capsules were retained for later use. Blastoids and blastoid capsules were stained with the Calcein-AM/PI double staining kit (YEASEN, HB210602) at 37 °C and incubated for 30 min at room temperature.

### Single-cell RNA-seq data analysis

Clean FASTQ files with 10 × single-cell barcode and unique molecular identifiers were processed using 10 × Genomics CellRanger (v.7.0.0) with default parameters to generate a gene expression matrix. The Macaca fascicularis genome (release 102) was downloaded from Ensembl and used for mapping. The filtered matrixes from CellRanger output were imported into Seurat (v4.1.1) in R (v4.1.2) for quality control and further analysis. Cells with low quality (< 2000 genes/cell, > 7500 genes/cell, < 3000 UMI,> 40000 UMI and >10% mitochondrial genes) were excluded. The doublet or multiplet cells were figured out by the DoubletFinder (v2.0.3) R package. 8941 eligible cells were identified. Cell cycle phase scores were calculated using the CellcycleScoring function in Seurat and regressed out during data scaling. The RunPCA function was then employed for PCA dimensionality reduction, retaining the top principal components for computing Uniform Manifold Approximation and Projection (UMAP) or t-Distributed Stochastic Neighbor Embedding (t-SNE) dimensional reduction. The FindClusters function with a parameter resolution of 0.6 was utilized to identify clusters of cells resulting in the classification of these 8941 cells into 10 clusters. Subsequently, we utilize the data from GSE148683 to aid in the identification of cell types. Finally, we identified a total of 2787 TLCs, 1096 HLCs, and 820 ELCs. After classifying all single cells, we analyzed the differentially expressed genes (DEGs) between these three cell types using the FindAllMarkers function in Seurat, with the parameters test.use set to 'wilcox' and only.pos set to TRUE. A heatmap was plotted based on the top 80 highly variable genes of each cell type (Supplementary Data 1). The KEGG and GO enrichment analyses are conducted by KOBAS-i[58]. KOBAS-i provides a web interface for KEGG and GO enrichment analysis (Supplementary Data 2). In the Gene-list Enrichment function, the statistical method "hypergeometric test/Fisher's exact test" should be selected, and the FDR correction method "Benjamini and Hochberg" should be applied, which are also the default methods on the KOBAS-i web platform.

### Single-cell RNA-seq data collection and integration

The primate dataset included GSE148683, GSE74767 and GSE75764. The monkey blastoid dataset included GSE218375. The human dataset included GSE171820, and the amnion dataset included GSE134571.

We used the scTransform Integration pipeline to integrate different datasets. In brief, the single object/dataset was normalized individually using the SCTransform function in Seurat[59]. Then, 3000 integration features and anchors among the different datasets were identified. Finally, the different datasets were integrated into one object.

### Reporting summary

Further information on research design is available in the Nature Portfolio Reporting Summary linked to this article.

## Data availability

The main data supporting the results of this study are available within the paper and its Supplementary Information. The sequence data reported in this paper have been deposited in the Genome Sequence Archive in the National Genomics Data Center, Chinese Academy of Sciences, under accession number PRJCA030207 [https://ngdc.cncb.ac.cn/gsa/browse/CRA019019/CRX1191842] and are publicly accessible at https://ngdc.cncb.ac.cn/gsa. Other data used in this manuscript: Natural

monkey embryo data (GSE148683 [https://www.ncbi.nlm.nih.gov/geo/query.cgi?acc=GSE148683], GSE74767 [https://www.ncbi.nlm.nih.gov/geo/query.cgi?acc=GSE74767] and GSE75764 [https://www.ncbi.nlm.nih.gov/geo/query.cgi?acc=GSE75764]); monkey blastoid data (GSE218375 [https://www.ncbi.nlm.nih.gov/geo/query.cgi?acc=GSE218375]); amniotic ectoderm-like cells data (GSE134571 [https://www.ncbi.nlm.nih.gov/geo/query.cgi?acc=GSE134571]); human blastoid data (GSE171820 [https://www.ncbi.nlm.nih.gov/geo/query.cgi?acc=GSE171820]). Every request about data availability can be directed to and will be fulfilled by, the corresponding author. Source data are provided in this paper.

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

## Acknowledgements

This work was supported by grants from the National Key R&D Program of China (2021YFA0805700 to Y.N. and 2018YFA0801400 to Y.N.), the National Natural Science Foundation of China (U2102204 to Y.N. and 52202350 to L.Z.) and the Natural Science Foundation of Yunnan Province (202001BC070001 to W.J., 202102AA100053 to W.J., 202101BE070001-057 to L.Z. and 202201AU070080 to L.Z.). We are grateful to X. Qi, and L. Gong for their helpful and to T. Tan and Z. Ai for their discussions at the Kunming University of Science and Technology; We acknowledge the members of the Animal Facility of the Yunnan Key Laboratory of Primate Biomedical Research for excellent animal welfare and husbandry; the instrumental from Advanced Imaging Platform of Institute of Primate Translational Medicine, Kunming University of Science and Technology.

## Author contributions

Y.N., L.Z., and W.J. designed the study and supervised all experiments. J.W., Z.T., Y.S., J.Z., and J.L. performed experiments on stem cells. J.W., Z.L., and J.Z performed IF stainings. G.L., B.Z., and J.W. performed the original bioinformatics analysis and management. T.S. and G.H. constructed microfluidic devices to encapsulate blastoids with high flux to form blastoid capsules. Y.K. performed embryo micromanipulation. T.S. prepared GelMA and performed functional verification of GelMA. Y.N., L.Z., W.J., J.W., T.S., and Z.T. wrote the manuscript.

## Competing interests

The authors declare no competing interests.

## Additional information

**Supplementary information** The online version contains
supplementary material available at

Weizhi Ji, Lei Zhang or Yuyu Niu.

**Peer review information** *Nature Communications* thanks Carlos Pinzón-
Arteaga, and the other anonymous reviewer(s) for their contribution to
the peer review of this work. A peer review file is available.

