## [Transparent Peer Review file · Nature Communications]

Highly efficient construction of monkey blastoid capsules from aged somatic cells

Corresponding Author: Professor Yuyu Niu

Version 0:

Reviewer comments:

Reviewer #1

(Remarks to the Author)

In this manuscript, the authors claimed a high efficiency monkey blastoid generation system using different cell sources as starting cells include monkey ESCs, monkey iPSCs and monkey nt-ESCs from young and aged rhesus monkey fibroblasts. They also developed an engineering platform for hydrogel-based blastoid capsules in a monodisperse, consistent and reproducible manner, which improves mechanical preservation, enhances the blastoids' stability and offers an effective means for their delivery through fallopian tubes. The authors claimed this integrative strategy that combines blastoids derived from aged monkey PSCs, droplet microfluidics and biomaterials would offer a platform for blastoid research and regenerative medicine.

Overall the authors tried to establish a high efficient monkey blastoids system. However, there are several major concerns on the data quality of the morphology and lineage compositions analysis of the monkey blastoids and the functional assays in evaluating the in vitro and in vivo developmental potential of the monkey blastoids. These data are quite important for supporting the conclusions the author claimed.

As monkey blastoid has been reported before, this reviewer believes that the novel point of this manuscript mainly relies on the hydrogel-based blastoid capsules system. However, the advantage of this system has not been well demonstrated and investigated.

In this reviewer's opinion, the authors should largely improve the data quality and address the concerns in a revision version.

The followings are specific comments:

Line 106, "effectively promoted the survival of M-ESCs following cell passaging", but in the corresponding figure, no significant difference was found.

The author should demonstrate the iPSC and nt-PSC they used can differentiate to PrE and TE in vitro, before monkey blastoid induction.

The author should demonstrate that ESC, PrE and TSC can be derived from the induced monkey blastoids.

Removing A83 and PD031542, blastoid formation efficiency was still about ~50%. This result needs to be further examined.

In line 134-135 "in various species the blastocyst cavity formation rate was low". In many reported papers, the blastoid formation rate has been largely improved. The author should update the citation of the reference. (e.g. DOI <https://doi.org/10.1038/s41586-021-04267-8> and DOI: <https://doi.org/10.1016/j.stem.2023.08.002>)

The author should also detect the expression of additional marker genes for ELCs and TLCs (beyond the ones used in the article, OCT4, GATA3, GATA4, GATA6) to validate the corresponding differentiation features of these cells.

How about the in vitro developmental potential of the blastoids? This is important and necessary.

Line 164, progesterone levels rise after in vivo transplantation. The reviewer did not see the data in the manuscript.

Line 170, The cell ratio of EPI, PE and TE lineages in normal blastocysts was about 2:2:6. The ratio of ELCs in this study is very low. The author should discuss about this point.

Line 174, in figure 3c, the expression levels of ELCs marker genes including SOX2 and POU5F1 is relatively low. The authors should provide more evidence when annotating the cell clusters.

The KEGG was not conclusively associated with cell type annotation. GO term analysis of these three lineages is much more prefer.

Line 174-175, in figure 3d-e, the two clusters HLC and TLC are not distinguished. Their identification needs to be further performed in-depth.

Line 179, in figure 3f-h, the ELC of blastoids is not much like the EPI of normal blastocysts in monkeys or humans, but rather the TLCs is much closer to the EPI of normal blastocysts in monkeys or humans? Cluster trees should be provided in figure 3f-h to show the similarity among cell clusters from this study and other monkey/human embryos.

Similarly, in the immunofluorescence staining of all iPSC and nt-ESC induced blastoids, no cavity was seen. Can you provide the staining image of mono layer from z-stack?

In vitro and in vivo developmental potential of iPSC and nt-ESC induced blastoids also should be evaluated by prolonged in vitro culture and in vivo transplantation.

The reviewer did not find the advantage for the application of the hydrogel capsule system in blastoids generation. Most of the morphology in the capsule looks not good enough.

Reviewer #2

(Remarks to the Author)

This work demonstrates the establishment of monkey blastoids from monkey iPSCs and nuclear transfer ESCs, and attempts to use a hydrogel-based microfluidics platform for blastoid generation. However, in its current form, I believe the article lacks focus on its theme, and each section lacks systematic demonstration. More experiments need to be added for it to be suitable for publication in Nature Communications. Here are my comments:

1. From the bright field observation, the outer layer cells are dense, with a high nucleus-to-cytoplasm ratio, and co-express GATA4 and GATA6. This suggests to me that these might be endoderm cells (endoderm cells also form cavity structures), rather than trophoblast cells. More validation is needed to confirm that these cells are indeed TE cells.
2. The characterization of blastoid in the article is severely lacking. At the very least, immunofluorescence staining should be used to identify different lineage cells. Specifically, OCT4, SOX2, NANOG, etc. for Epiblast, GATA6, SOX17, PDGFR, etc. for Hypoblast, CDX2, GATA3, TFAP2C, TEAD4, KRT18, etc. for trophoblast, and CCR7 for polar trophoblast. Additionally, monkey blastocysts should be included as a parallel comparison.
3. Quantitative analysis of the proportions of different cell types within this structure is needed.
4. The single-cell analysis is overly simplistic. It needs to include clustering information and the expression profiles of key genes in each lineage (EPI, TE, HYPO). Ideally, this should be compared with the blastocyst (from monkeys and/or humans) dataset.
5. Whether these blastoids can continue to develop in vitro is crucial. It is necessary to conduct in vitro culture and characterization experiments to confirm their developmental capacity.
6. What is the key message the authors want to send that to generate the blastoids from young and old monkeys? It should be clarified whether a comparative analysis is needed or if their potential applications should be demonstrated.
7. The work on hydrogels seems unrelated to the previous sections. The advantages and applications of adding a hydrogel shell have not been demonstrated, which makes the article's theme appear somewhat confusing. Therefore, I suggest removing this part from the article, and this part of the work is more suitable for systematic presentation in a specialized journal.
8. The degradability of these hydrogels and whether blastoids can hatch from the hydrogel need to be systematically demonstrated. If the blastoids cannot hatch from the gel, then this approach lacks significant value.
9. The in vivo transplantation experiments are too simplistic; merely reporting the capture rate is insufficient. From my understanding, the hydrogel shell actually hinders the blastoids from implanting in the uterus, leaving them in a free-floating state where they can be captured.
10. The differences and advantages of this work compared to previously published study (Jie Li et al, 2023, Cell Stem Cell) need to be further discussed.

Reviewer #3

(Remarks to the Author)

In this study by Lei Zhang, Yuyu Niu and colleges, the authors report on the generation of cavities from from induced pluripotent stem cells (iPSCs) and somatic cell nuclear transfer ESCs (nt-ESCs) obtained from young and aged rhesus monkeys, that they have termed blastocyst like structures or "blastoids". These cells were cultured in their previously established naïve-like XF-PSC medium, these cells survived aggregation in the presence of a high-affinity rock inhibitor (Pro-S) and a commercial lipid-enriched albumin supplement (CloneR). Aggregates after exposure to hypoblast differentiation medium (HDM) and then followed by trophoblast differentiation medium (TSG) form cavities with up to 80% efficiency. In addition, the authors have developed a hydrogel microfluidic platform for the encapsulation of the blastoids in a scalable manner that protects the cells upon transfer to the fallopian tubes.

While I found their results interesting and data is promising, I believe the data does not fully support their claims. Therefore, I do not support the publication of this manuscript in Nature Communications unless my following concerns are addressed:

Major points:

1. Lack of evidence of bona fide trophectoderm formation:

My main concern of their findings is that the cavities the authors observe are not trophectoderm but amnion. This concern arises from the nature of the XF-PSC culture conditions, high levels of FGF (high ERK signaling) a key characteristic of primed culture conditions. Naïve culture conditions (5i/LA, PXGL, PXGGY, etc) are normally cultured with a combination of one or more inhibitors of the FGF-RAS-RAF-MEK-ERK pathway and this key signaling difference allows for the

differentiation to extra-embryonic tissues (i.e Trophoblast), which the authors have included in the TSG medium.

Both naïve and primed cells have the capacity to form cavities upon exposure to MEK1/2 inhibitor PD0325901 and the ALK5 TGF- β receptor inhibitor A-83, with naïve cells generating bona fide trophoblast, but cavities from partially primed or primed PSCs correspond to early and late amnion-like stages¹. Other groups have succeeded in the production of well-characterized cynomolgus monkey blastoids from 4C1 culture conditions².

These concerns arise from the following observations:

- -Visually the trophoblast has a high level of cellularity that does not correspond with a normal blastocyst trophoblast. Side-by-side images should be shown.
- -The majority of the cells that show GATA3 expression co-express SOX17, a strong endoderm marker and not present in trophoblast².
- -The data shows a large proportion of the putative trophoblast shows expression of ISL1, normally considered an amnion marker.
- -scRNA seq data shows very few cells with expression of ICM pluripotency markers (SOX2, OCT4)
- -The data on hCG production capacity is not provided.

The authors should provide more detailed characterization to convincingly show these cavities are indeed trophoblast and not amnion.

2. Side-by-Side Comparisons with Monkey Blastocysts:

A side-by-side comparison with monkey blastocysts will be a good control not only for pattern of expression but also the amount of expression of key markers. If possible the authors should consider performing 10x genomics sequencing to diminish the cross-platform comparison of rhesus monkeys.

3. Discrepancies Between Immunostaining and scRNA-seq Data:

The immunostaining results don't match the scRNA seq data, what percent of blastoids are expressing, epiblast, hypoblast and trophoblast markers, and what is the average ratio per blastoid?

4. Comparison with Published Datasets:

The authors should compare their data with other published datasets by unbiased clustering and show how the transcriptome of their cells overlap other previously annotated cell types.

Minor points:

1. In Vitro Culture and Gastrulation:

Can these blastoids be cultured in vitro and undergo gastrulation?

2. Pluripotency Proof:

To prove the blastoids retain pluripotency, they should be able to derive ESCs from the blastoids.

3. Post-Encapsulation Expansion and Hatching:

Can the putative blastoids continue to expand after the encapsulation?, and Can the blastoids "hatch" from the hydrogel capsules?

4. Naïve or Totipotent-Like Culture Conditions:

Have the authors tried culture in naïve conditions, or totipotent-like culture conditions (e.g 4CL).

5. Lack of rationale or mechanistic explanation for TSG Medium:

The rationale or scientific explanation for the TSG medium is lacking, why the authors activate WNT (1 μ M CHIR99021) and inhibit WNT (1 μ M IWR-1-endo), at the same time? Hippo pathway is one of the most important pathways in the trophoblast differentiation, the authors should evaluate if the addition of LPA has any benefit in their TSG differentiation medium. Have the authors tried a simpler condition (PD03, A83 and LPA)?. If their medium can support all lineages, can a normal blastocyst form under the TSG medium?

Bibliography

1. Rostovskaya, M., Andrews, S., Reik, W., and Rugg-Gunn, P.J. (2022). Amniogenesis occurs in two independent waves in primates. *Cell Stem Cell* 29, 744-759.e746. 10.1016/j.stem.2022.03.014.
2. Li, J., Zhu, Q., Cao, J., Liu, Y., Lu, Y., Sun, Y., Li, Q., Huang, Y., Shang, S., Bian, X., et al. (2023). Cynomolgus monkey embryo model captures gastrulation and early pregnancy. *Cell Stem Cell* 30, 362-377.e367. 10.1016/j.stem.2023.03.009.

Version 1:

Reviewer comments:

Reviewer #1

(Remarks to the Author)

In the rebuttal and revised manuscript, the authors have made an effort to address the issues. There have some questions remain confused and unanswered, the followings are comments:

(1-1): The authors have answered my questions properly.

(1-2): The authors have answered my questions properly.

(1-3): The stem cells in in response Figure 1-3 seems to be stained immediately when the blastoids attached. Can the blastoid-derived three lineage stem cells be passaged stably? If so, please provide evidence. And whether the signal for HLA-G seems to be non-specific noise. This result needs to be further examined.

(1-4): In Reviewers Figure 1-4c, I did not see obvious presence of inner cell mass-like cells inside of the blastoids in several groups. DOUBT on the structure and morphology of the blastoid you induced. The signal of ZO1 in the panel c is incorrect (hypoblast lineage) in response Figure 1-6. It is recommended to replace it with higher quality images.

(1-5): The authors have answered my questions properly.

(1-6): In Response Figure 1-6: the panel of b shows no signal for GATA4 (hypoblast lineage). The result needs more check.

(1-7): The in vitro developmental potential of iPSC and nt-ESC induced blastoids should be identified more detailed except the simple staining. More identification information including specific morphological description, marker genes expression and lineage specification in in vitro-cultured blastoids should be provided. Develop to what stage? Have the in vitro-cultured blastoids really reached the post-implantation stage? Was it compared to post-implantation natural embryo of blastoids data? How to judge the in vitro developmental potential of your blastoids strictly and precisely? Why the IVC of blastoids only kept for 4-5 days? Is the developmental potential is not enough or the blastoids did not continue to develop further? What are the criteria for stopping in vitro culture? Please provide a detailed analysis and identification process. The significance of the establishment of monkey blastoids is to break through the ethics that human blastoids cannot be cultured in vitro beyond E14. If it is only raised before E14, it can be directly study in a human blastoid, there is no need to do in monkey.

(1-8): The authors have answered my questions properly.

(1-9): The ratio of GATA4+ HLCs seems too high, similar to TLCs, which does not fit the ratio of 2:2:6 described previously, and the location of the signal seems not right. Can you explain the reason or provide new image or another set of markers?

(1-10): The authors have answered my questions properly.

(1-11): The authors have answered my questions properly.

(1-12): The clusters are still not distinguished from each other in the three lineages of blastoids in Supplementary Figs. 3j-l according to the expression of specific marker genes.

(1-13): The HLC data shows is similar to the endoderm data of yanagida in the in response Figure panel b. Please provide a detailed analysis.

(1-14): The authors have answered my questions properly.

(1-15): The in vitro developmental potential of iPSC and nt-ESC induced blastoids should be identified more detailed except the simple staining.

(1-16) : The morphology, gene expression and lineage composition of blastoids in the hydrogel after encapsulation at different time points require more in-depth and detailed identification to real time monitor the state of blastoids in the hydrogel after encapsulation. And in figure 6f, we didn't see proliferation of blastoid in the hydrogel after encapsulation, but rather it seems to degrade with time going, especially on day2 and day4.

Reviewer 1 was asked by the editorial team to comment on how the authors addressed the concerns raised by Reviewer 2 in the first round because Reviewer 2 was unable to provide their report.

The following is Rev. 1's assessment on Rev. 2's points.

(2-1): No significant improvement in blastoid morphology was observed using either white light or immunofluorescence staining. The ratio of induced TLC seems relatively low based on the response to Figure 2-1, and the signal for HLA-G may be non-specific noise. Additionally, there are too few markers for TE, and the GATA3 appears to be expressed across all three cell types, while the expression level of TEAD4 is not particularly high in TLC according to the RNA-seq data.

(2-2): It may be appropriate to present the data showing co-expression of the markers for each of the three "lineage cells", instead of a single expression pattern for blastoid and blastocysts.

(2-3): The ratio of GATA4+ HLCs seems too high, similar to TLCs, which does not fit the ratio of 2:2:6 described previously, and the location of the signal seems not right.

(2-4): The HLC cells express higher levels of GATA3 compared to the TLC cells, which raises my suspicion that your TLC in Supplementary Fig. 3f may not be true TLC.

(2-5): The in vitro developmental potential of blastoids should be characterized in more detail beyond simple staining. Typical structures such as the epiblast, amniotic cavity, and yolk sac cavity should be included.

(2-6): I am satisfied with your explanation. Exploring cells from different ages will be beneficial for us in establishing models in the future!

(2-7) : I'm convinced by what you've presented.

(2-8): The authors have answered my questions properly for the degradability of these hydrogels.

(2-9): After the embryo transfer, is there any further observation or monitoring of the transplanted blastoid development, or are the levels of chorionic gonadotropin being checked for changes? More explanations and experiments are needed

(2-10): The authors only briefly compare the differences to Jie Li et al., but the discussion section of the article requires a deeper analysis. This part should be integrated into the main text.

Reviewer #3

(Remarks to the Author)

I believe the revision process has been very helpful to improve the overall quality of the manuscript and to support the claims. I believe the authors have done a good job addressing my concerns as well as the concerns of the other reviewers. I fell the inclusion of the new data supports the conclusions and support the publication given minor edits are addressed:

1. Some references are misplaced or are missing context, example Line 620 the reference 54.

2. The authors disregard very important historical references regarding the discovery of the hippo signaling pathway in trophectoderm differentiation^{1,2} although they cite a reference showing its evolutionary conservation³. A simple immunostaining for YAP in monkey blastocysts and blastoids should answer Hippo signaling role in monkey TE specification⁴.

References

1 Nishioka, N. et al. Tead4 is required for specification of trophectoderm in pre-implantation mouse embryos. 125, 270-283 (2008). PMID <https://doi.org/10.1016/j.mod.2007.11.002>

2 Nishioka, N. et al. The Hippo signaling pathway components Lats and Yap pattern Tead4 activity to distinguish mouse trophectoderm from inner cell mass. 16, 398-410 (2009). PMID <https://doi.org/10.1016/j.devcel.2009.02.003>

3 Gerri, C. et al. Initiation of a conserved trophectoderm program in human, cow and mouse embryos. 587, 443-447 (2020). PMID <https://doi.org/10.1038/s41586-020-2759-x>

4 Liu, L. et al. Modeling post-implantation stages of human development into early organogenesis with stem-cell-derived peri-gastruloids. 186, 3776-3792.e3716 (2023). PMID <https://doi.org/10.1016/j.cell.2023.07.018>

Version 2:

Reviewer comments:

Reviewer #1

(Remarks to the Author)

In the rebuttal and revised manuscript, the authors have made a commendable effort to address the issues raised in the review. The inclusion of new experiments and higher-quality images are particularly praiseworthy. These revisions have significantly enhanced the quality of the manuscript. I am satisfied with the revisions and support publication.

Peer Review File

Reviewer Comments & Author Point-by-point response Reviewer Reports on the Initial Version:

REVIEWER COMMENTS

Reviewer #1 (Remarks to the Author):

In this manuscript, the authors claimed a high efficiency monkey blastoid generation system using different cell sources as starting cells include monkey ESCs, monkey iPSCs and monkey nt-ESCs from young and aged rhesus monkey fibroblasts. They also developed an engineering platform for hydrogel-based blastoid capsules in a monodisperse, consistent and reproducible manner, which improves mechanical preservation, enhances the blastoids' stability and offers an effective means for their delivery through fallopian tubes. The authors claimed this integrative strategy that combines blastoids derived from aged monkey PSCs, droplet microfluidics and biomaterials would offer a platform for blastoid research and regenerative medicine.

Overall the authors tried to establish a high efficient monkey blastoids system. However, there are several major concerns on the data quality of the morphology and lineage compositions analysis of the monkey blastoids and the functional assays in evaluating the in vitro and in vivo developmental potential of the monkey blastoids. These data are quite important for supporting the conclusions the author claimed.

As monkey blastoid has been reported before, this reviewer believes that the novel point of this manuscript mainly relies on the hydrogel-based blastoid capsules system. However, the advantage of this system has not been well demonstrated and investigated.

In this reviewer's opinion, the authors should largely improve the data quality and address the concerns in a revision version.

The followings are specific comments:

Line 106, "effectively promoted the survival of M-ESCs following cell passaging", but in the corresponding figure, no significant difference was found.

The author should demonstrate the iPSC and nt-PSC they used can differentiate to PrE and TE in vitro, before monkey blastoid induction.

The author should demonstrate that ESC, PrE and TSC can be derived from the induced monkey blastoids.

Removing A83 and PD031542, blastoid formation efficiency was still about ~50%. This result needs to be further examined.

In line 134-135 "in various species the blastocyst cavity formation rate was low". In many reported papers, the blastoid formation rate has been largely improved. The author should update the citation of the reference. (e.g.

DOI <https://doi.org/10.1038/s41586-021-04267-8> and

DOI: <https://doi.org/10.1016/j.stem.2023.08.002>)

The author should also detect the expression of additional marker genes for ELCs and TLCs (beyond the ones used in the article, OCT4, GATA3, GATA4, GATA6) to validate the corresponding differentiation features of these cells.

How about the the in vitro developmental potential of the blastoids? This is important and necessary.

Line 164, progesterone levels rise after in vivo transplantation. The reviewer did not see the data in the manuscript.

Line 170, The cell ratio of EPI, PE and TE lineages in normal blastocysts was about 2:2:6. The ratio of ELCs in this study is very low. The author should discuss about this point.

Line 174, in figure 3c, the expression levels of ELCs marker genes including SOX2 and POU5F1 is relatively low. The authors should provide more evidence when annotating the cell clusters.

The KEGG was not conclusively associated with cell type annotation. GO term analysis of these three lineages is much more prefer.

Line 174-175, in figure 3d-e, the two clusters HLC and TLC are not distinguished. Their identification needs to be further performed in-depth.

Line 179, in figure 3f-h, the ELC of blastoids is not much like the EPI of normal blastocysts in monkeys or humans, but rather the TLCs is much closer to the EPI of normal blastocysts in monkeys or humans? Cluster trees should be provided in figure 3f-h to show the similarity among cell clusters from this study and other monkey/human embryos.

Similarly, in the immunofluorescence staining of all iPSC and nt-ESC induced blastoids, no cavity was seen. Can you provide the staining image of mono layer from z-stack?

In vitro and in vivo developmental potential of iPSC and nt-ESC induced blastoids also should be evaluated by prolonged in vitro culture and in vivo transplantation.

The reviewer did not find the advantage for the application of the hydrogel capsule system in blastoids generation. Most of the morphology in the capsule looks not good enough.

Reviewer #2 (Remarks to the Author):

This work demonstrates the establishment of monkey blastoids from monkey iPSCs and nuclear transfer ESCs, and attempts to use a hydrogel-based microfluidics platform for blastoid generation. However, in its current form, I believe the article lacks focus on its theme, and each section lacks systematic demonstration. More experiments need to be added for it to be suitable for publication in Nature Communications. Here are my comments:

1. From the bright field observation, the outer layer cells are dense, with a high nucleus-to-cytoplasm ratio, and co-express GATA4 and GATA6. This suggests

to me that these might be endoderm cells (endoderm cells also form cavity structures), rather than trophoblast cells. More validation is needed to confirm that these cells are indeed TE cells.

2.The characterization of blastoid in the article is severely lacking. At the very least, immunofluorescence staining should be used to identify different lineage cells. Specifically, OCT4, SOX2, NANOG, etc. for Epiblast, GATA6, SOX17, PDGFR, etc. for Hypoblast, CDX2, GATA3, TFAP2C, TEAD4, KRT18, etc. for trophoblast, and CCR7 for polar trophoblast. Additionally, monkey blastocysts should be included as a parallel comparison.

3.Quantitative analysis of the proportions of different cell types within this structure is needed.

4.The single-cell analysis is overly simplistic. It needs to include clustering information and the expression profiles of key genes in each lineage (EPI, TE, HYPO). Ideally, this should be compared with the blastocyst (from monkeys and/or humans) dataset.

5.Whether these blastoids can continue to develop in vitro is crucial. It is necessary to conduct in vitro culture and characterization experiments to confirm their developmental capacity.

6.What is the key message the authors want to send that to generate the blastoids from young and old monkeys? It should be clarified whether a comparative analysis is needed or if their potential applications should be demonstrated.

7.The work on hydrogels seems unrelated to the previous sections. The advantages and applications of adding a hydrogel shell have not been demonstrated, which makes the article's theme appear somewhat confusing. Therefore, I suggest removing this part from the article, and this part of the work is more suitable for systematic presentation in a specialized journal.

8.The degradability of these hydrogels and whether blastoids can hatch from the hydrogel need to be systematically demonstrated. If the blastoids cannot hatch from the gel, then this approach lacks significant value.

9.The in vivo transplantation experiments are too simplistic; merely reporting the capture rate is insufficient. From my understanding, the hydrogel shell actually hinders the blastoids from implanting in the uterus, leaving them in a free-floating state where they can be captured.

10.The differences and advantages of this work compared to previously published study (Jie Li et al, 2023, Cell Stem Cell) need to be further discussed.

Reviewer #3 (Remarks to the Author):

In this study by Lei Zhang, Yuyu Niu and colleagues, the authors report on the generation of cavities from induced pluripotent stem cells (iPSCs) and somatic cell nuclear transfer ESCs (nt-ESCs) obtained from young and aged rhesus monkeys, that they have termed blastocyst like structures or “blastoids”. These cells were cultured in their previously established naïve-like XF-PSC

medium, these cells survived aggregation in the presence of a high-affinity rock inhibitor (Pro-S) and a commercial lipid-enriched albumin supplement (CloneR). Aggregates after exposure to hypoblast differentiation medium (HDM) and then followed by trophoblast differentiation medium (TSG) form cavities with up to 80% efficiency. In addition, the authors have developed a hydrogel microfluidic platform for the encapsulation of the blastoids in a scalable manner that protects the cells upon transfer to the fallopian tubes.

While I found their results interesting and data is promising, I believe the data does not fully support their claims. Therefore, I do not support the publication of this manuscript in Nature Communications unless my following concerns are addressed:

Major points:

1. Lack of evidence of bona fide trophoblast formation:

My main concern of their findings is that the cavities the authors observe are not trophoblast but amnion. This concern arises from the nature of the XF-PSC culture conditions, high levels of FGF (high ERK signaling) a key characteristic of primed culture conditions. Naïve culture conditions (5i/LA, PXGL, PXGGY, etc) are normally cultured with a combination of one or more inhibitors of the FGF-RAS-RAF-MEK-ERK pathway and this key signaling difference allows for the differentiation to extra-embryonic tissues (i.e Trophoblast), which the authors have included in the TSG medium.

Both naïve and primed cells have the capacity to form cavities upon exposure to MEK1/2 inhibitor PD0325901 and the ALK5 TGF- β receptor inhibitor A-83, with naïve cells generating bona fide trophoblast, but cavities from partially primed or primed PSCs correspond to early and late amnion-like stages¹. Other groups have succeeded in the production of well-characterized cynomolgus monkey blastoids from 4C1 culture conditions².

These concerns arise from the following observations:

- -Visually the trophoblast has a high level of cellularity that does not correspond with a normal blastocyst trophoblast. Side-by-side images should be shown.
- -The majority of the cells that show GATA3 expression co-express SOX17, a strong endoderm marker and not present in trophoblast².
- -The data shows a large proportion of the putative trophoblast shows expression of ISL1, normally considered an amnion marker.
- -scRNA seq data shows very few cells with expression of ICM pluripotency markers (SOX2, OCT4)
- -The data on hCG production capacity is not provided.

The authors should provide more detailed characterization to convincingly show these cavities are indeed trophectoderm and not amnion.

2. Side-by-Side Comparisons with Monkey Blastocysts:

A side-by-side comparisons with monkey blastocysts will be a good control not only for pattern of expression but also the amount of expression of key markers. If possible the authors should consider performing 10x genomics sequencing to diminish the cross-platform comparison of rhesus monkeys.

3. Discrepancies Between Immunostaining and scRNA-seq Data:

The immunostaining results don't match the scRNA seq data, what percent of blastoids are expressing, epiblast, hypoblast and throphectoderm markers, and what is the average ratio per blastoid?

4. Comparison with Published Datasets:

The authors should compare their data with other published datasets by unbiased clustering and show how the transcriptome of their cells overlap other previously annotated cell types.

Minor points:

1. In Vitro Culture and Gastrulation:

Can these blastoids be cultured in vitro and undergo gastrulation?

2. Pluripotency Proof:

To prof the blastoids retain pluripotency, they should be able to derive ESCs from the blastoids.

3. Post-Encapsulation Expansion and Hatching:

Can the putative blastoids continue to expand after the encapsulation?, and Can the blastoids "hatch" from the hydrogel capsules?

4. Naïve or Totipotent-Like Culture Conditions:

Have the authors tried culture in naïve conditions, or totipotent-like culture conditions (e.g 4CL).

5. lack of rationale or mechanistic explanation for TSG Medium:

The rational or scientific explanation for the TSG medium is lacking, why the authors activate WNT (1 μ M CHIR99021) and inhibit WNT (1 μ M IWR-1-endo), at the same time? Hippo pathway is one of the most important pathways in the

throphectoderm differentiation, the authors should evaluate if the addition of LPA has any benefit in their TSG differentiation medium. Have the authors tried a simpler condition (PD03, A83 and LPA)?. If their medium can support all lineages, can a normal blastocyst form under the TSG medium?

Bibliography

1. Rostovskaya, M., Andrews, S., Reik, W., and Rugg-Gunn, P.J. (2022). Amniogenesis occurs in two independent waves in primates. *Cell Stem Cell* 29, 744-759.e746. [10.1016/j.stem.2022.03.014](https://doi.org/10.1016/j.stem.2022.03.014).
2. Li, J., Zhu, Q., Cao, J., Liu, Y., Lu, Y., Sun, Y., Li, Q., Huang, Y., Shang, S., Bian, X., et al. (2023). Cynomolgus monkey embryo model captures gastrulation and early pregnancy. *Cell Stem Cell* 30, 362-377.e367. [10.1016/j.stem.2023.03.009](https://doi.org/10.1016/j.stem.2023.03.009).

Point-by-point response to referees' comments

Encouraged by the positive assessment and constructive comments made by the reviewers we have now performed an in-depth revision of our manuscript. In this we have performed significant additional experiments, conducted further computational analyses and re-wrote the manuscript. Together, we feel to have significantly improved our manuscript. We have also addressed each specific point raised by the reviewers, as detailed below. The reviewers' comments are in black and our replies are indicated in blue.

Reviewer #1 (Remarks to the Author):

In this manuscript, the authors claimed a high efficiency monkey blastoid generation system using different cell sources as starting cells include monkey ESCs, monkey iPSCs and monkey nt-ESCs from young and aged rhesus monkey fibroblasts. They also developed an engineering platform for hydrogel-based blastoid capsules in a monodisperse, consistent and reproducible manner, which improves mechanical preservation, enhances the blastoids' stability and offers an effective means for their delivery through fallopian tubes. The authors claimed this integrative strategy that combines blastoids derived from aged monkey PSCs, droplet microfluidics and biomaterials would offer a platform for blastoid research and regenerative medicine.

Overall the authors tried to establish a high efficient monkey blastoids system. However, there are several major concerns on the data quality of the morphology and lineage compositions analysis of the monkey blastoids and the functional assays in evaluating the in vitro and in vivo developmental potential of the monkey blastoids. These data are quite important for supporting the conclusions the author claimed. As monkey blastoid has been reported before, this reviewer believes that the novel point of this manuscript mainly relies on the hydrogel-based blastoid capsules system. However, the advantage of this system has not been well demonstrated and investigated.

In this reviewer's opinion, the authors should largely improve the data quality and address the concerns in a revision version.

We thank the reviewer for their thoughtful comments on our manuscript and address their comments below.

The followings are specific comments:

(1-1) Line 106, "effectively promoted the survival of M-ESCs following cell passaging", but in the corresponding figure, no significant difference was found.

Our reply: Thank you for your valuable comment, which has indeed helped us improve our manuscript. We increased the number of replicates and performed additional statistical analyses to better evaluate the survival of monkey ESC (M-ESCs) following cell passaging. Cells treated with the "CloneR+Pro-S"

combination have promoted self-assembly levels. (NEW Supplementary Figs. 1b, d and i).

Response to Reviewers Figure 1-1: The cells treated with the "CloneR + Pro-S" combination enhanced the level of self-assembly.

(a) Ratio of percentage of viable cells in M-ESCs with Clone R, Y27632 and Pro-S. (b) Ratio of percentage of viable cells in M-ESCs with CloneR + Y27632 + Pro-S, Clone R + Y27632, and Clone R + Pro-S. (c) Representative bright field images of cell aggregates at indicated time points during blastoid formation with CloneR+ Pro-S. Scale bar, 200 μ m.

(1-2) The author should demonstrate the iPSC and nt-PSC they used can differentiate to PrE and TE *in vitro*, before monkey blastoid induction.

Our reply: We agree that the ability of M-iPSCs and M-nt-ESCs to differentiate into hypoblast-like cells (HLCs) and trophoblast-like cells (TLCs) *in vitro* is very important. As you pointed out, except M-ESCs, we also should demonstrate the potential of differentiation into hypoblast and trophoblast in M-iPSCs and M-nt-ESCs. We have therefore carried out new experiments in which we generated inducible into HLCs and TLCs *in vitro*. M-nt-ESCs and M-iPSCs were differentiated into HLCs using previous protocol¹, and GATA4 and SOX17, markers of hypoblast, were detected in the cells after differentiation. These indicated that HLCs were differentiated from M-nt-ESCs and M-iPSCs. Besides, GATA3 and TFAP2C, markers of trophoblast, were expressed in M-nt-ESCs and M-iPSCs cultured in medium of TE-differentiation. Accordingly, these

suggest that there was developmental potential of trophoblast and hypoblast in M-nt-ESCs and M-iPSCs. (NEW Supplementary Figs. 5a, b)

In previous study, we confirmed that XF-PSC cultured monkey PSCs labeled with GFP-Akaluc were chimeric in homologous embryos. After long-term culture, chimeric cells were detected to differentiate into hypoblast, trophoblast, primordial germ cells, and primitive streak cells lineages in d.p.f.15-19 chimeric embryos; On the other hand, after transplantation the chimeric embryos into surrogate monkeys, chimeric cells can be detected in both the placenta and the body of aborted fetuses. Besides, chimeric cells can be detected in the umbilical cord of living chimeric monkey, and in particular, signals from chimeric cells were detected for at least 2 years in the body of living chimeric monkey². These results suggest that monkey PSCs cultured in XF-PSC have the potential to develop into both embryonal and extraembryonic lineages, indicate that these monkey pluripotent stem cells can be used to construct primate embryo models.

Response to Reviewers Figure 1-2: iPSCs and nt-PSCs are capable of differentiating into hypoblast and trophoblast cell lineages *in vitro*.

(a) Representative immunofluorescent staining of TFAP2C/GATA3 showing differentiation of M-nt-ESCs and M-iPSCs into trophoblast lineage. Scale bars, 100 μm. (b) Representative immunofluorescent staining of GATA4/SOX17 showing differentiation of M-nt-ESCs and M-iPSCs into hypoblast lineage. Scale bars, 100 μm.

(1-3) The author should demonstrate that ESC, PrE and TSC can be derived from the induced monkey blastoids.

Our reply: We agree with you that it is important to demonstrate the derivation of epiblast, hypoblast, and trophoblast from the induced monkey blastoids. And we also realize that this is an important characterization of blastoid. Using previous protocol^{2, 3}, blastoid-derived epiblast-like cells (b-ELCs) were derived from blastoids, and core pluripotency gene, such as *POU5F1* and *SOX2*, were expressed in b-ELCs. Besides, blastoid-derived trophoblast-like cells (b-TLCs) were derived from blastoids under previous reports, b-TLCs express *GATA3* and *TFAP2C*. Similarly, b-hypoblast-like cells (b-HLCs) which express *GATA4* and *SOX17* were derived from blastoids. Furthermore, the cells derived from blastoids were able to differentiate into SDC1-positive cells and HLA-G-positive cells which are characteristics of syncytiotrophoblasts and extravillous

cytotrophoblasts⁴. These results suggest that b-ELCs, b-HLCs and b-TLCs can be derived from the induced monkey blastoids. (NEW Supplementary Fig. 6a-f)

Response to Reviewers Figure 1-3: ESCs, PrE and TSCs can be derived from the induced monkey blastoids.

(a) Schematic diagram for the derivation of b-ELCs, b-TLCs and b-HLCs cells from blastoids. (b) Representative immunofluorescent staining of markers of epiblast (OCT4/SOX2) in b-ELCs. Scale bars, 100 μ m. (c) Representative immunofluorescent staining of markers of hypoblast (GATA4/SOX17) in b-HLCs. Scale bars, 100 μ m. (d) Representative immunofluorescent staining of markers of trophectoderm (TFAP2C/GATA3) in b-TLCs. Scale bars, 100 μ m. (e) Representative immunofluorescent staining images of marker genes detection in STB-like cells (SDC1). Scale bars, 100 μ m. (f) Representative immunofluorescent staining images of marker genes detection in EVT-like cells (HLA-G). Scale bars, 100 μ m.

(1-4) Removing A83 and PD, blastoid formation efficiency was still about ~50%. This result needs to be further examined.

Our reply: Your suggestions are greatly appreciated. By withdrawing A83-01 and PD0325901 during the construction of blastoids, cavities still appeared in the blastoids in multiple experiments, and there was not significant change in efficiency compared to the normal group. Immunofluorescence staining showed that the removal of A83-01 and PD0325901 did not affect the appearance of GATA3-positive cells. In recent reports, Guo et al. reported that human pluripotent stem cells can generate blastoids without the addition of A83-01/PD0325901⁵. (NEW Supplementary Fig. 2g)

Response to Reviewers Figure 1-4: The removing of A83-01 and PD0325901 during blastoids construction did not have much effect.

(a) Representative bright field images of blastoids treated without A83-01, PD0325901, A83-01 and PD0325901 compared with control. Scale bars, 200 μm . (b) Cavity formation efficiency of monkey blastoids cultured without A83-01, PD0325901, A83-01 and PD0325901 compared with complete TSG. Ordinary one-way ANOVA test; Error bars, mean \pm S.E.M. (c) Representative immunofluorescence staining images of GATA3/ZO-1/PHALLOIDIN in monkey blastoids without A83-01, PD0325901, A83-01 and PD0325901 compared with control. Scale bars, 100 μm .

(1-5) In line 134-135 “in various species the blastocyst cavity formation rate was low”. In many reported papers, the blastoid formation rate has been largely improved. The author should update the citation of the reference. (e.g. DOI <https://doi.org/10.1038/s41586-021-04267-8> and DOI: <https://doi.org/10.1016/j.stem.2023.08.002>)

Our reply: Thank you for your valuable feedback. We have revised the relevant sentence to better reflect the recent advancements in blastoid formation efficiency. Specifically, we have updated the text to mention the efficient generation of human blastoids as reported in the suggested references. However, we also emphasized that despite these advancements, efficient generation of monkey blastoids has yet to be reported. This indicates that there remains room for optimization in this area, especially for non-human primate models. The revised text and updated citations can be found in lines (L143-146).

<< Recent studies have reported the efficient generation of human blastoids^{6, 7}. However, the efficient generation of monkey blastoids has yet to be reported, with current rates around 30%⁸, indicating that there is still potential for further optimization in non-human primate models. >>

(1-6) The author should also detect the expression of additional marker genes for ELCs and TLCs (beyond the ones used in the article, OCT4, GATA3, GATA4, GATA6) to validate the corresponding differentiation features of these cells.

Our reply: We agree with you that expression of additional marker genes for ELCs and TLCs should be detected. Using immunofluorescence staining, we found that inner layer cells of the induced monkey blastoid expressed core pluripotent markers (OCT4, SOX2, NANOG). For TLCs, CDX2, GATA3, GATA2 and CCR7, markers of trophoblast were expressed in outer layer cells of the blastoids. TFAP2C were not detected in outer layer cells of the blastoids, but were expressed in the blastoid-derived b-TLCs. That may be caused short induction of trophoblast, because in the previous screening, in order to generate aged blastoids, we screened the processing time of HDM and TSG, and finally found that HDM for 4 days and TSG for 1 day were the best. (NEW Supplementary Fig. 2n-r and Supplementary Fig. 6d)

Response to Reviewers Figure 1-6: The expression of additional marker genes for ELCs and TLCs.

(a) Representative immunofluorescent staining images of GATA3/SOX2/SOX17 in monkey blastoids. Scale bars, 100 μm . (b) Representative immunofluorescent staining images of GATA3/NANOG/GATA4 in monkey blastoids. Scale bars, 100 μm . (c) Representative immunofluorescent staining images of CDX2/OCT4/GATA2 in monkey blastoids. Scale bars, 100 μm . (d) Representative immunofluorescent staining image of CCR7 in monkey blastoids. Scale bars, 100 μm . (e) Representative immunofluorescent staining of markers of trophectoderm (TFAP2C/GATA3) in ESC derived from monkey blastoids. Scale bars, 100 μm .

(1-7) How about the the in vitro developmental potential of the blastoids? This is important and necessary.

Our reply: Thank you for highlighting the importance of assessing the *in vitro* developmental potential of the blastoids. To address this, we performed an *in vitro* attachment assay based on previously established protocols⁹. The blastoids were cultured in a 2D environment using IVC medium, and their morphological changes were closely monitored. Upon attachment, the blastoids flattened and expanded, forming outgrowth structures that resemble those observed in human blastocysts and blastoids derived through other methodologies. At 2 and 4 days for *in vitro* culture (IVC), OCT4 or SOX2 can be detected in the blastoids. Interestingly, the separation of GATA3 and GATA6 expression was more pronounced in blastoids after *in vitro* culture, because the separation of GATA3 and GATA6 expression was not significant before IVC. At 5 days for IVC, T (also known as TBXT or Brachyury, a feature of primitive streak) were expressed in the blastoids. Similar result was observed on our previous study about long-term natural monkey embryo culture *in vitro*⁹. Additionally, we measured the levels of mCG secretion in the culture medium. Attached blastoids exhibited a significant increase in mCG secretion starting from day 5 of *in vitro* culture, whereas no mCG secretion was detected in the blank IVC medium or blastoid induction medium after 5 days of culture. These findings demonstrate the *in vitro* developmental potential of the induced monkey blastoids. (NEW Supplementary Fig. 6g-l)

Response to Reviewers Figure 1-7: The developmental potential of the induced monkey blastoids *in vitro*.

(a) Schematic showing the process of blastoid formation from monkey PSC. (b) Monkey mCG in Blastoid IVC day 5 compared with IVC medium as control. (c) Representative bright field images of monkey blastoid IVC on day 1-4. Scale bars, 200 μ m. (d) Representative immunofluorescent staining images of GATA3/OCT4/GATA6 in monkey blastoid IVC on day 2. Scale bars, 100 μ m. (e) Representative immunofluorescent staining images of GATA3/SOX2/GATA6 in monkey blastoid IVC on day 4. Scale bars, 100 μ m. (f) Representative immunofluorescent staining images of T/GATA6 in monkey blastoid IVC on day 5. Scale bars, 100 μ m.

(1-8) Line 164, progesterone levels rise after *in vivo* transplantation. The reviewer did not see the data in the manuscript.

Our reply: Thank you for pointing this out. We did not measure the level of progesterone in monkeys after transplantation, but according to previous

reports, mCG can be used to reflect the transfer of blastoids. We have now included the mCG data for the transplantation of both blastoids and natural embryos into the monkey uterus. A certain level of mCG was detected after natural blastocysts and blastoids were transplanted into monkeys. (NEW Supplementary Figs. 6m, n)

Response to Reviewers Figure 1-8: mCG levels rise after *in vivo* transplantation.

(a) The level of mCG after natural blastocyst transfer from day 5 to day 11 in monkeys. (b) The level of mCG after blastoids transfer from day 5 to day 11 in monkeys.

(1-9) Line 170, The cell ratio of EPI, PE and TE lineages in normal blastocysts was about 2:2:6. The ratio of ELCs in this study is very low. The author should discuss about this point.

Our reply: Thank you for highlighting this important point. We have conducted additional scRNA-seq analysis of the blastoids. The ratio of ELCs, HLCs, and TLCs in blastoids is 2:2:6. An increase in the proportion of ELCs were observed. However, the ratio remains different from the typical 2:2:6 ratio observed in natural blastocysts.

Besides, we used flow cytometry to quantify the percentage of various cell types in blastoids: the percentage of high-expressing OCT4 cells was 19.3%, GATA6+ cells was 75.9%, and GATA3+ cells was 90.9%.

Similar to other blastoids derived using various methods, the cell lineage proportions in our blastoids deviate from those in natural blastocysts. In particular, scRNA-seq analysis revealed an increased HLCs population, which is distinct from natural monkey blastocysts. We believe this discrepancy may be due to continuous signaling during the induction process, which promotes the hypoblast lineage. This suggests that further optimization of the blastoid derivation protocol is necessary to better mimic the natural cell lineage distribution. (NEW Supplementary Figs. 3g, h)

Response to Reviewers Figure 1-9: Projection and frequency of the three pre-implantation lineages on the UMAP.

(a) UMAP shows monkey blastoids scRNA-seq included three subclusters: TLC, HLC and ELC. (b) Pie chart showing the frequencies of ELC, TLC, and HLC in monkey blastoids indicated by lineage markers. (c) FACS shows the percentage of OCT4-, GATA6- and GATA3-positive cells in monkey blastoids.

(1-10) Line 174, in figure 3c, the expression levels of ELCs marker genes including SOX2 and POU5F1 is relatively low. The authors should provide more evidence when annotating the cell clusters.

Our reply: Thanks for making this project better with your review. We have conducted additional scRNA-seq analysis of the blastoids and observed an increase in the proportion of ELCs. We have updated the scRNA-seq data to classify different cell populations in blastoids. We conducted unbiased clustering analysis on the data, which showed high similarity compared to natural embryos. In the ELCs cell cluster, *POU5F1*, and *NANOG* were expressed, with *POU5F1* having a higher expression level. (NEW Figs. 3d and Supplementary Figs. 3g, j)

Response to Reviewers Figure 1-10: Genes expression of ELC in blastoid. (a) UMAP show the cell clusters of ELC, HLC and TLC in monkey blastoid. (b) Projection of EPI markers (OCT4 and NANOG) on the UMAP. (c) Heatmap showing the ELC, TLC, and HLC signatures in the various clusters.

(1-11) The KEGG was not conclusively associated with cell type annotation. GO term analysis of these three lineages is much more prefer.

Our reply: Thank you for pointing out the issue. In KEGG, the ELCs gene is enriched in axon guidance and WNT signaling pathway, the HLCs gene is enriched in tight junction and ECM-report interaction, and the TLCs gene is enriched in cell cycle for the TOP80 genes of the three cell clusters; For GO enrichment, ELCs genes are enriched in signal transduction, BMP signaling pathway and WNT signaling pathway which were related with primitive steak, HLCs genes are enriched in cadherin binding and extracellular matrix organization, and TLCs genes are enriched in cell cycle. (NEW Supplementary Figs. 3m-r)

Response to Reviewers Figure 1-11: Genes enrichments in ELC, HLC and TLC.

(a) KEGG enrichments and representative genes in top 80 genes (clusters ELC, HLC and TLC). (b) GO enrichments and representative genes in top 80 genes (clusters ELC, HLC and TLC).

(1-12) Line 174-175, in figure 3d-e, the two clusters HLC and TLC are not distinguished. Their identification needs to be further performed in-depth.

Our reply: We thank the reviewer for this suggestion and have now added further analysis. We performed unbiased clustering by integrating 10X data from natural embryos¹⁰ with our data. This analysis revealed a high similarity between the three lineages of natural embryos (epiblast, trophoblast, and hypoblast) and our own data. Using key genes identified from the natural embryo data, we were able to successfully distinguish between HLC and TLC in our dataset. These results confirm the presence of distinct cell populations and address the previous lack of clear separation between these two clusters. Meanwhile, we followed the HDM+TDM blastoid induction method reported by Wu's group. We found that using 4 days of HLCs induction and 1 day of TLCs induction was sufficient to generate blastoids from aged and high-passage PSCs. However, compared to previous studies, our protocol involved a shorter TLCs induction time, which may have resulted in the incomplete maturation of the TE lineage and contributed to the lack of clear distinction between HLCs and TLCs. To address this issue, we updated our scRNA-seq data and performed more in-depth analysis. Although the distinction between HLCs and

TLCs remains somewhat subtle, we observed improved separation between these clusters in UMAP. (NEW Supplementary Figs. 3a-d)

Response to Reviewers Figure 1-12: Single-cell transcriptional profiling of human blastoids.

(a) and (b) UMAP of single-cell transcriptomes of cells from monkey blastoids, and monkey preimplantation embryos. (c) Define each cell cluster in monkey blastoids. (d) UMAP plot of monkey blastoid cells integrated with published monkey embryo cells. Unclassified cells were in grey.

(1-13) Line 179, in figure 3f-h, the ELC of blastoids is not much like the EPI of normal blastocysts in monkeys or humans, but rather the TLCs is much closer to the EPI of normal blastocysts in monkeys or humans? Cluster trees should be provided in figure 3f-h to show the similarity among cell clusters from this study and other monkey/human embryos.

Our reply: We thank the reviewer for this thoughtful comment. We conducted in-depth analysis of scRNA-seq and compared it with reported monkey blastocyst data, monkey natural embryos, and human blastoids data for unbiased clustering analysis. It was found that ELC, HLC, and TLC have high similarity with their corresponding lineages on UMAP. (NEW Figs. 3a-c, e and Supplementary Fig. 3s)

Response to Reviewers Figure 1-13: Single-cell transcriptomes of monkey blastoids, human blastoids and monkey blastocysts.

(a) UMAP of single-cell transcriptomes of cells from monkey blastoids, published monkey blastoids datasets. (b) UMAP of single-cell transcriptomes of cells from monkey blastoids, and monkey preimplantation embryos. (c) UMAP of single-cell transcriptomes of cells from monkey blastoids compared with published datasets from human early embryos. (d) and (e) Correlation

analysis of ELC, HLC and TLC in monkey blastoid with natural preimplantation monkey and human embryo and published monkey blastoid.

(1-14) Similarly, in the immunofluorescence staining of all iPSC and nt-ESC induced blastoids, no cavity was seen. Can you provide the staining image of mono layer from z-stack?

Our reply: Thanks for your suggestions. In the maximum projection images provided previously, the cavity was not clearly visible. To address this, we have now included single-layer images from the z-stack immunofluorescence staining, which provide a clearer view of the blastoids' structure. These images will help to better visualize and confirm the presence of the cavity in the M-iPSC and M-nt-ESC induced blastoids.

Response to Reviewers Figure 1-14: The presence of the cavity in the M-iPSC and M-nt-ESC induced blastoids.

(a) Representative immunofluorescent staining images of OCT4/GATA3/GATA4 in M-iPSCs-derived blastoid. Scale bars, 100 μm . (b) Representative immunofluorescent staining images of OCT4/GATA3/GATA4 in M-nt-ESCs-derived blastoid. Scale bars, 100 μm .

(1-15) In vitro and in vivo developmental potential of iPSC and nt-ESC induced blastoids also should be evaluated by prolonged in vitro culture and in vivo transplantation.

Our reply: Thank you for your valuable suggestion. We have indeed evaluated the developmental potential of M-iPSCs-derived and M-nt-ESCs-derived blastoids through both extended *in vitro* culture. After prolonged *in vitro* culture, we observed outgrowth in these blastoids, and further analysis revealed the expression of lineage-specific markers, including SOX2 (ELCs), GATA3 (TLCs), and GATA6 (HLCs). These findings demonstrate that the blastoids have the capacity for further development, which we have now detailed in the revised manuscript. (NEW Supplementary Figs. 6o-r)

Response to Reviewers Figure 1-15: The blastoids have the capacity for further development *in vitro* culture and *in vivo* transplantation.

(a) Representative bright-field images of M-nt-ESCs-derived blastoid developing *in vitro* on day 3. Scale bars, 200 μm . (b) Representative bright-field images of M-iPSCs-derived blastoid developing *in vitro* on day 3. Scale bars, 200 μm . (c) Representative immunofluorescent staining images of GATA3/GATA6/SOX2 in M-nt-ESCs-derived blastoid IVC on day 3. Scale bars, 100 μm . (d) Representative immunofluorescent staining images of GATA3/GATA6/SOX2 in M-iPSCs-derived blastoid IVC on D3. Scale bars, 100 μm .

(1-16) The reviewer did not find the advantage for the application of the hydrogel capsule system in blastoids generation. Most of the morphology in the capsule looks not good enough.

Our reply: Thank you for your valuable feedback. Biological hydrogels are a type of three-dimensional network structure formed by the physical or chemical crosslinking of natural or synthetic polymer materials. Hydrogels possess certain mechanical properties that can resist the pressure, shear forces, and other physical factors that blastoids may encounter during delivery. Therefore, encapsulating blastoids in hydrogels to create blastoid capsules can improve the survival rate and stability of the blastoids, facilitating manipulation and processing, and allowing for easy transplantation, transportation, and storage. This is particularly important in applications such as regenerative medicine and

drug screening. Additionally, in our work, we have established a coaxial microfluidic system that allows us to efficiently utilize hydrogels for high-throughput and high-uniformity encapsulation of blastoids, wrapping them in micrometer-scale microsphere capsules. The smooth surface of the capsules can reduce collisions and adhesion during the delivery of blastoids, thereby improving delivery efficiency. We simulated the delivery channel environment of the fallopian tube in vitro, and the efficiency of delivering blastoids through capsules increased by 300% compared to direct delivery of blastoids. Cultivation and staining of the delivered blastoid capsules showed that the encapsulation did not negatively affect the growth of the blastoids. (NEW Figs. 8b, c)

On the other hand, hydrogel materials are biocompatible, so blastoid capsules don't affect the activity and growth of blastoids. Furthermore, we supplemented the results of blastoids growing and being released within hydrogel capsules. During embryo implantation and pregnancy, the activity of collagenase increases during the remodeling of the endometrium and the formation of the placenta. Here, we conducted in vitro degradation experiments on biomaterials by simulating the concentration of collagenase within organs during this period. To confirm whether blastoids can grow and be released within the blastoid capsules after delivery, we tested the degradation performance of the hydrogel in vitro. After 48 hours of degradation, we could clearly observe that the hydrogel material was essentially completely degraded. We also continuously cultured blastoid capsules in vitro and observed that blastoids gradually grew and were released as the hydrogel degraded. (NEW Supplementary Figs. 8a, b)

In the cryopreservation of blastoids, extreme low temperatures can cause the formation of intracellular ice crystals, and the osmotic pressure changes during the freezing process can impact cell viability, thereby causing severe damage to the structure and cells of the blastoids. However, by encapsulating blastoids within hydrogel capsules, we have provided a certain level of protection for the blastoids. After low-temperature storage for 12 hours under the same conditions, the viability and morphology of the blastoids were observed through live-dead staining. The results showed that the blastoids within the hydrogel capsules were superior in morphology and survival to those that were not encapsulated. (NEW Supplementary Fig. 8d)

In summary, the blastoid capsules prepared by our hydrogel encapsulation possess good biocompatibility, allowing blastoids to complete culture and growth within them and to hatch out smoothly after the degradation of the hydrogel shell. This is beneficial for the storage, protection, and delivery of blastoids within tissues and under extreme conditions.

Finally, We understand that the previous image may have misled readers regarding the morphology of blastoids in hydrogel capsules. Thank you for pointing it out, and we have promptly replaced the image of the blastoid

capsules. We appreciate your insights and hope this clarifies the advantages of our hydrogel capsule system.

Response to Reviewers Figure 1-16: Degradation of hydrogel, growth, delivery and cryopreservation of blastoid capsules.

(a) Biodegradation of round GelMA hydrogel with diameter of 5mm for 48 h. (b) Biodegradation of microspheres with a diameter of 300 microns prepared using high-throughput microfluidic devices after 42 hours. (c) GelMA-FITC hydrogel

grafted on GelMA using fluorescent group FITC was excited to emit green fluorescence under 494nm wavelength irradiation. GelMA-FITC hydrogel is used to observe the biodegradability of hydrogel more intuitively. (d) and (e) The scheme and the capture rate of blastoids and blastoid capsules in the monkey fallopian tubes, n = 3 biological replicates. Error bars, mean \pm S.E.M. unpaired two-tailed *t*-test. (f) Representative images of immunofluorescence staining at 3 and 6 hours after the blastoid is loaded into a capsule. Scale bars, 200 μ m. (g) Biodegradation of blastoid capsules formed by GelMA hydrogel encapsulation. (h) Blastoid are encapsulated within a hydrogel to form blastoid capsules. (i) The morphology of the organoids and organoid capsules after 12 hours of cryopreservation.

Reviewer #2 (Remarks to the Author):

This work demonstrates the establishment of monkey blastoids from monkey iPSCs and nuclear transfer ESCs, and attempts to use a hydrogel-based microfluidics platform for blastoid generation. However, in its current form, I believe the article lacks focus on its theme, and each section lacks systematic demonstration. More experiments need to be added for it to be suitable for publication in Nature Communications. Here are my comments:

We thank the reviewer for their thoughtful comments on our manuscript and address their comments below.

(2-1) From the bright field observation, the outer layer cells are dense, with a high nucleus-to-cytoplasm ratio, and co-express GATA4 and GATA6. This suggests to me that these might be endoderm cells (endoderm cells also form cavity structures), rather than trophectoderm cells. More validation is needed to confirm that these cells are indeed TE cells.

Our reply: Thank you for your insightful comment. We acknowledge your concern that the outer layer cells may resemble endoderm cells based on their morphology and co-expression of GATA4 and GATA6. This may be due to the shortened induction time for trophectoderm specification, as our previous optimization for generating aged blastocysts identified that HDM treatment for 4 days followed by TSG treatment for 1 day yielded the best results. However, we have conducted several additional experiments to validate that these cells are indeed TLCs:

Initial Identification: The outer layer cells were initially identified as TLCs based on their morphology and position in the blastoid structure, which is consistent with the expected location of TE cells in natural blastocysts.

Additional Marker Validation: In addition to GATA4 and GATA6, we performed immunofluorescence staining for trophectoderm-specific markers, including CCR7, GATA2, and CDX2. These markers are strongly expressed in trophectoderm cells and are not typically found in endodermal cells. The co-expression of these markers with GATA4 and GATA6 supports the identification

of these cells as trophectoderm. Although TFAP2C was not detected in the outer layer cells of the blastoids, it was expressed in blastoid-derived trophoblast-like cells (b-TLCs). Additionally, we observed syncytiotrophoblast-like cells (SDC1+ and GATA3+) and extravillous cytotrophoblast-like cells (HLA-G+ and GATA3+) differentiating from blastoid-derived cells, further indicating the presence of TLC lineage. (NEW Supplementary Fig. 2n-r and Supplementary Fig. 6d)

Transcriptomic Validation: To further confirm the identity of the TE cells, we performed single-cell RNA sequencing (scRNA-seq) and compared the transcriptomic profiles of these cells to those of natural trophectoderm and primitive endoderm from blastocysts. The results showed a closer similarity to the trophectoderm lineage. (NEW Fig. 3e and Supplementary Fig. 3a, i)

Functional Validation: We extended the *in vitro* culture of the blastoids, and by day 5, mCG secretion was detected in the supernatant, which is a known marker of trophectoderm-derived cells. This functional evidence further supports the identity of these cells as TLC. (NEW Supplementary Fig. 6g)

Morphological Consideration: While a high nucleus-to-cytoplasm ratio is commonly observed in endodermal cells, we also observed this feature in early-stage trophectoderm cells of natural blastocysts in our parallel control experiments. The combination of trophectoderm-specific markers, along with their location in the blastoid structure, strongly supports our conclusion that these are TLC cells, not endoderm.

These results indicate that the TLCs in the induced monkey blastoids share significant similarities with the TE lineage in natural blastocysts.

Response to Reviewers Figure 2-1: The characterization of blastoid.

(a) Representative immunofluorescent staining image of CCR7 in monkey blastoids. Scale bars, 100 μ m. (b) Representative immunofluorescent staining images of CDX2/OCT4/GATA2 in monkey blastoids. Scale bars, 100 μ m. (c) Representative immunofluorescent staining of markers of trophoblast (TFAP2C/GATA3) in ESC derived from monkey blastoids. Scale bars, 100 μ m. (d) Representative immunofluorescent staining images of marker genes detection in STB-like cells (SDC1). Scale bars, 100 μ m. (e) Representative immunofluorescent staining images of marker genes detection in EVT-like cells (HLA-G). Scale bars, 100 μ m. (f) Representative phase contrast images of

monkey blastocyst-E9 (left) and monkey blastoid (right). Scale bars, 200 μ m.
 (g) mCG in Blastoid IVC Day 5 compared with IVC medium as control.

Response to Reviewers Figure 2-1: scRNA-seq analysis shows the presence of TLCs within the monkey blastoids.

(a) UMAP of single-cell transcriptomes of cells from monkey blastoids, published monkey blastoids datasets. (b) UMAP of single-cell transcriptomes of cells from monkey blastoids, and monkey preimplantation embryos. (c) 0, 1,

2 subclusters were identified as ELCs, PLCs and TLCs respectively based on lineage markers. Individual cells are colored by origin: TLC (red), HLC (green), ELC (blue). (d) Projection of TE markers on the UMAP.

(2-2) The characterization of blastoid in the article is severely lacking. At the very least, immunofluorescence staining should be used to identify different lineage cells. Specifically, OCT4, SOX2, NANOG, etc. for Epiblast, GATA6, SOX17, PDGFR, etc. for Hypoblast, CDX2, GATA3, TFAP2C, TEAD4, KRT18, etc. for trophoblast, and CCR7 for polar trophoblast. Additionally, monkey blastocysts should be included as a parallel comparison.

Our reply: Thank you for your detailed comment. We have conducted immunofluorescence staining to compare the lineage marker expression between blastoids and natural blastocysts. Both blastoids and natural blastocysts show OCT4 and NANOG expression in inner cells, while SOX17+, CDX2+, and GATA3+ cells are present in the outer layer. Moreover, CCR7, which shows differential expression in polar and mural TE in human blastocysts¹¹, exhibits a similar expression pattern in our blastoids. (NEW Figs. 2n-r)

In addition, we assessed other key markers reported in natural blastocysts, such as SOX2 in the inner cells of blastoids. Unfortunately, due to the unavailability of TEAD4 antibodies, we were unable to validate TEAD4 via immunostaining; however, its expression was confirmed in our scRNA-seq data. (NEW Supplementary Fig. 3l)

To further characterize the blastoids, we derived ELCs, HLCs, and TLCs from the blastoids. These included OCT4+/SOX2+ blastoid-derived ELCs (b-ELCs), GATA4+/SOX17+ blastoid-derived HLCs (b-HLCs), and GATA3+/TFAP2C+ blastoid-derived TLCs (b-TLCs). Notably, the blastoid-derived cells were able to differentiate into SDC1+/GATA3+ multinucleated cells and HLA-G+/GATA3+ cells, characteristic of syncytiotrophoblast (STB) and extravillous trophoblast (EVT), respectively. This indicates strong similarities between the TLCs in the blastoids and the natural trophoblast lineage. (NEW Supplementary Figs. 6b-f)

Response to Reviewers Figure 2-2-1: The characterization of blastoid compared with monkey blastocyst-E9.

(a) Representative immunofluorescent staining images of GATA3/SOX17 in monkey blastoid (top) and blastocyst (bottom).. Scale bars, 100 μm. (b) Representative immunofluorescent staining images of CDX2 in monkey blastoid (top) and blastocyst (bottom).. Scale bars, 100 μm. (c) Representative immunofluorescent staining images of NANOG in monkey blastoid (top) and blastocyst (bottom). Scale bars, 100 μm. (d) Representative immunofluorescent staining images of OCT4 in monkey blastoid (top) and blastocyst (bottom). Scale bars, 100 μm. (e) Representative immunofluorescent staining images of CCR7 in monkey blastoid (top) and blastocyst (bottom). Scale bars, 100 μm.

Response to Reviewers Figure 2-2-2: UMAP shows the characterization of monkey blastoids.

(a) 0, 1, 2 subclusters were identified as ELCs, PLCs and TLCs respectively based on lineage markers. Individual cells are colored by origin: TLC (red), HLC (green), ELC (blue). (b) Projection of TEAD4 on the UMAP.

a

b

c

d

e

f

Response to Reviewers Figure 2-2-3: The potential of monkey blastoids to differentiate into different lineage cells.

(a) Schematic diagram of monkey blastoid differentiation into multiple cell types. (b) Representative immunofluorescent staining of markers of EPI (OCT4/SOX2) in ESC derived from monkey blastoids. Scale bars, 100 μm . (c) Representative immunofluorescent staining of markers of hypoblast (GATA4/SOX17) in ESC derived from monkey blastoids. Scale bars, 100 μm . (d) Representative immunofluorescent staining of markers of trophectoderm (TFAP2C/GATA3) in ESC derived from monkey blastoids. Scale bars, 100 μm . (e) Representative immunofluorescent staining images of marker genes detection in STB-like cells (SDC1). Scale bars, 100 μm . (f) Representative immunofluorescent staining images of marker genes detection in EVT-like cells (HLA-G). Scale bars, 100 μm .

(2-3) Quantitative analysis of the proportions of different cell types within this structure is needed.

Our reply: Thank you for your suggestion. We have performed quantitative analyses of the proportions of different cell types within the blastoids using two approaches:

Flow Cytometry: We used flow cytometry to quantify the percentage of various cell types in blastoids: the percentage of high-expressing OCT4 cells was 19.3%, GATA6+ cells was 75.9%, and GATA3+ cells was 90.9%.

Single-Cell RNA Sequencing Analysis: Additionally, we performed bioinformatic analysis of single-cell RNA sequencing data. The results revealed a cell ratio of ELC: HLC : TLC= 2:2:6, which aligns with the expected proportions based on the cell lineages present in natural blastocysts.

Response to Reviewers Figure 2-3: Cell ratio in monkey blastoid by scRNA-seq and FACS.

(a) UMAP showed monkey blastoids' scRNA-seq included three subclusters: TLC, HLC and TLC. (b) Pie chart showing the frequencies of ELC, TLC, and HLC in monkey blastoids indicated by lineage markers. (c) FACS shows the percentage of OCT4-, GATA6- and GATA3-positive cells in monkey blastoids.

(2-4) The single-cell analysis is overly simplistic. It needs to include clustering information and the expression profiles of key genes in each lineage (EPI, TE, HYPO). Ideally, this should be compared with the blastocyst (from monkeys and/or humans) dataset.

Based on expression patterns of known marker genes^{9, 12, 13}, EPI-like cells (ELCs, expressing POU5F1, SOX2, NANOG and FOXH1), hypoblast-like cells (HLCs, expressing GATA4 and TTR), and TLCs (expressing GATA3 and TEAD4) could be identified from blastoid cells.

We next wished to investigate whether pluripotent stem cell-derived blastoid could mimic the natural cynomolgus monkey blastocyst at the transcriptome level. Previous data on natural cynomolgus monkey embryos^{10, 12, 14}, monkey blastoid⁸, and human blastoid¹⁵ were referenced for integration analysis^{8, 10}. Integrated clustering analysis revealed that the three main lineages of blastoids clustered with the EPI, HYP, and TE of physiological peri-implantation embryos *in vivo*, respectively. These results indicated that newly induced monkey blastoids contained comparable major lineages and corresponding transcriptomic features with natural blastocysts. (NEW Fig. 3, and Supplementary Figs. 3a-g and I)

Response to Reviewers Figure 2-4: Single-cell transcriptomes analysis and comparison of monkey blastoids, human blastoids and monkey blastocysts.

(a) UMAP show single-cell transcriptomes in monkey blastoids and natural preimplantation monkey embryo. (b) UMAP show single-cell transcriptomes in monkey blastoids. (c) Cell type proportion in each cell clusters in monkey blastoids. (d) Point out TLC, HLC and ELC in cell cluster in monkey blastoid on UMAP. (e) UMAP show cell cluster 0, 1, 2 in monkey blastoid without undefined cells. (f) Genes expression of cluster 0, 1, 2 on ELC, TLC and HLC specific genes. (g) UMAP show TLC, HLC and ELC in monkey blastoid. (h) Joint uniform manifold approximation and projection (UMAP) embedding of single-

cell transcriptomes of monkey blastocysts, blastoids and related datasets. (i) Correlation analysis of ELC, HLC and TLC in monkey blastoid with natural preimplantation monkey and human embryo and published monkey blastoid.

(2-5) Whether these blastoids can continue to develop in vitro is crucial. It is necessary to conduct in vitro culture and characterization experiments to confirm their developmental capacity.

Our reply: Thank you for emphasizing the importance of assessing the *in vitro* developmental potential of the blastoids. In response to this, we conducted an *in vitro* attachment assay following established protocols. The blastoids were cultured in a 2D IVC medium, and their morphological changes were closely monitored. Upon attachment, the blastoids flattened and expanded, forming outgrowth structures similar to those seen in human blastocysts and blastoids generated via other methods. (NEW Supplementary Figs. 6g, h)

By day 2 and day 4 of extended culture, the ELCs in the blastoids expressed OCT4 or SOX2. By day 5, we detected the expression of T (also known as Brachyury or TBXT, a specific marker of primitive streak), along with the secretion of mCG in the culture supernatant, indicative of trophectoderm lineage activity. Notably, after prolonged culture, the segregation between GATA3⁺ TLCs and GATA6⁺ HLCs became more pronounced, further demonstrating the differentiation potential of the blastoids. (NEW Supplementary Figs. 6i-l)

Response to Reviewers Figure 2-5: The developmental capacity of the blastoids *in vitro*.

(a) Schematic showing the process of blastoid formation from Monkey iPSC. (b) mCG in Blastoid IVC Day 5 compared with IVC medium as control. (c) Representative images of monkey blastoid IVC on day 1-4. Scale bars, 200 μ m. (d) Representative immunofluorescent staining images of GATA3/OCT4/GATA6 in monkey blastoid IVC on Day 2. Scale bars, 100 μ m. (e) Representative immunofluorescent staining images of GATA3/SOX2/GATA6 in monkey blastoid IVC on Day 4. Scale bars, 100 μ m. (f) Representative immunofluorescent staining images of T/GATA6 in monkey blastoid IVC on Day 5. Scale bars, 100 μ m.

(2-6) What is the key message the authors want to send that to generate the blastoids from young and old monkeys? It should be clarified whether a

comparative analysis is needed or if their potential applications should be demonstrated.

Our reply: Thank you for raising this question. The key message we aim to convey by generating blastoids from both young and old monkeys is to explore potential age-related differences in blastoid formation and developmental capacity. This work extends primate blastoid research to aged autologous stem cells, providing a unique perspective for studying tissue regeneration in older individuals.

While a detailed comparative analysis of age-related differences is not the primary focus of this study, the ability to derive blastoids from monkeys of different ages offers important insights into how aging impacts cellular reprogramming and blastoid development. Additionally, this study sets the foundation for future investigations into the potential applications of age-specific blastoids, such as in regenerative medicine and age-related disease modeling.

(2-7) The work on hydrogels seems unrelated to the previous sections. The advantages and applications of adding a hydrogel shell have not been demonstrated, which makes the article's theme appear somewhat confusing. Therefore, I suggest removing this part from the article, and this part of the work is more suitable for systematic presentation in a specialized journal.

Our reply: Thank you for your insightful feedback. One of the central innovations of our work is the high-throughput fabrication of organoid capsules using microfluidic technology in combination with biomaterials. This approach not only provides a protective hydrogel shell for the organoids but also significantly enhances the efficiency of their delivery. Additionally, the hydrogel encapsulation improves cell survival after cryopreservation and subsequent thawing of blastoids, as demonstrated by our data.

By integrating microfluidics and biomaterials, our study highlights a scalable method that offers broader potential applications for organoid technology. The blastoid capsules prepared by our hydrogel encapsulation possess well biocompatibility, allowing blastoids to complete culture and growth within them and to hatch out smoothly after the degradation of the hydrogel shell. This is beneficial for the storage, protection, and delivery of blastoids within tissues and under extreme conditions. After careful consideration, we believe that the hydrogel-based encapsulation and its benefits are essential components of our study and should remain in the manuscript to provide a comprehensive understanding of its implications. (NEW Supplementary Fig. 8d)

a

Response to Reviewers Figure 2-7: The hydrogel encapsulation improves cell survival after cryopreservation and subsequent thawing of blastoids. (a) The morphology of the organoids and organoid capsules after 12 hours of cryopreservation. Scale bar 100 μm .

(2-8) The degradability of these hydrogels and whether blastoids can hatch from the hydrogel need to be systematically demonstrated. If the blastoids cannot hatch from the gel, then this approach lacks significant value.

Our reply: We thank the reviewer for this thoughtful comment. First, the hydrogel material we used is synthesized from fish skin gelatin. Gelatin is a product derived from the hydrolysis of collagen, and thus it can be degraded by collagenase. To address the concern regarding whether organoids encapsulated by the material can be released *in vivo*, we conducted the following experiments to demonstrate the rapid degradability of the capsules. During embryo implantation and pregnancy, the remodeling of the endometrium and the formation of the placenta involve increased collagenase activity. We simulated the collagenase concentration in organs during this period to perform an *in vitro* degradation experiment on the biomaterial. After 48 hours of degradation, we observed that the hydrogel material was almost completely degraded. Furthermore, in our earlier experiments, we also conducted *in vitro*

degradation of organoid capsules. After 48 hours of degradation, the organoids were observed to be released from the hydrogel. (NEW Supplementary Figs. 8a, b)

Response to Reviewers Figure 2-8-1: Degradation of hydrogel *in vitro*.

(a) Biodegradation of round GelMA hydrogel with diameter of 5 mm for 48 h. (b) Biodegradation of microspheres with a diameter of 300 μ m prepared using high-throughput microfluidic devices after 42 hours. (c) GelMA-FITC hydrogel grafted on GelMA using fluorescent group FITC was excited to emit green fluorescence under 494nm wavelength irradiation. GelMA-FITC hydrogel is used to observe the biodegradability of hydrogel more intuitively.

Response to Reviewer Figure 2-8-2: The delivery and growth of blastoid capsules.

(a) Blastoid and blastoid capsules pass through the fallopian tubes of monkeys *in vitro*. (b) The frequency of blastoid and blastoid capsules passing through the fallopian tubes of monkeys *in vitro*. (c) Immunofluorescence staining of DAPI, GATA3, OCT4, and GATA4 at 3 hours and 6 hours. (d) The blastoid-like structures gradually grow as the hydrogel degrades *in vitro*. (e) Blastoids were encapsulated within a hydrogel to form blastoid capsules.

(2-9) The *in vivo* transplantation experiments are too simplistic; merely reporting

the capture rate is insufficient. From my understanding, the hydrogel shell actually hinders the blastoids from implanting in the uterus, leaving them in a free-floating state where they can be captured.

Our reply: We agree with the reviewer. *In vitro* transplantation experiment was conducted to address the issue of low implantation efficiency during blastoid transplantation. If blastoids are directly transplanted, they tend to easily adhere to non-target areas rather than the uterus, which hampers successful implantation. We carried out transplantation of blastoids and blastoid capsules in the fallopian tubes of rats and monkeys. It was evident that the passage rate of blastoid capsules was significantly higher than that of blastoids alone, which greatly increased the success rate of transplantation. (NEW Fig. 8b) Additionally, the blastoid capsules that successfully passed through the fallopian tubes and reached the uterus were able to undergo surface shell degradation within the uterus, allowing the blastoids to be released from the capsules within a certain timeframe and continue with their subsequent biological processes. (NEW Supplementary Fig. 8c)

(2-10) The differences and advantages of this work compared to previously published study (Jie Li et al, 2023, Cell Stem Cell) need to be further discussed.

Our reply: Thank you for your comment. While the study by Jie Li et al. (2023, *Cell Stem Cell*) was groundbreaking in constructing monkey blastoids and demonstrating their potential for extended development, our work focuses on applying the blastoid technology with several advancements:

Establishing a High-Efficiency Blastoid Formation Method: Unlike Jie Li et al.'s approach, we developed a more efficient method for constructing monkey blastoids using compound-induced monkey embryonic stem cells. This method significantly improves blastoid formation efficiency. Moreover, we have demonstrated, through multiple analyses, that the morphology and lineage characteristics of our blastoids closely resemble those of natural embryos.

Generation of Blastoids from Autologous Pluripotent Stem Cells in Young and Old Monkeys: In our study, we extended the technology to generate blastoids from fibroblast cell lines of both young and old monkeys. Using induced pluripotent stem cell (iPSC) technology and somatic cell nuclear transfer (SCNT), we reprogrammed these fibroblasts into iPSCs and nuclear-transferred embryonic stem cells (nt-ESCs). This allowed us to explore the structural and lineage features of blastoids derived from both young and aged individuals, which was not explored in Jie Li et al.'s study.

Engineering Blastoids for Scalable Preparation: Our work also advances the engineering aspect of blastoids. We developed a microfluidic system using photo-crosslinkable bioactive hydrogels to create stable and scalable blastoid capsules with consistent size. These engineered blastoids were further tested for their *in vitro* developmental potential and *in vivo* functionality, adding a practical and scalable dimension to blastoid technology that was not a focus of Jie Li et al.'s research.

Reviewer #3 (Remarks to the Author):

In this study by Lei Zhang, Yuyu Niu and colleagues, the authors report on the generation of cavities from induced pluripotent stem cells (iPSCs) and somatic cell nuclear transfer ESCs (nt-ESCs) obtained from young and aged rhesus monkeys, that they have termed blastocyst like structures or “blastoids”. These cells were cultured in their previously established naïve-like XF-PSC medium, these cells survived aggregation in the presence of a high-affinity rock inhibitor (Pro-S) and a commercial lipid-enriched albumin supplement (CloneR). Aggregates after exposure to hypoblast differentiation medium (HDM) and then followed by trophoblast differentiation medium (TSG) form cavities with up to 80% efficiency. In addition, the authors have developed a hydrogel microfluidic platform for the encapsulation of the blastoids in a scalable manner that protects the cells upon transfer to the fallopian tubes.

While I found their results interesting and data is promising, I believe the data does not fully support their claims. Therefore, I do not support the publication of this manuscript in Nature Communications unless my following concerns are addressed:

We thank the reviewer for their thoughtful comments on our manuscript and address their comments below.

Major points:

(3-1) 1. Lack of evidence of bona fide trophoblast formation:

My main concern of their findings is that the cavities the authors observe are not trophoblast but amnion. This concern arises from the nature of the XF-PSC culture conditions, high levels of FGF (high ERK signaling) a key characteristic of primed culture conditions. Naïve culture conditions (5i/LA, PXGL, PXGGY, etc) are normally cultured with a combination of one or more inhibitors of the FGF-RAS-RAF-MEK-ERK pathway and this key signaling difference allows for the differentiation to extra-embryonic tissues (i.e Trophoblast), which the authors have included in the TSG medium.

Our reply: Thank you for your insightful comments. We have taken several steps to address your concern regarding the identity of the cavities observed in our blastoids:

In previous study, we confirmed that XF-PSC cultured cynomolgus monkey embryonic stem cells labelling GFP-Akaluc were chimeric into homologous embryos. After long-term culture, chimeric cells were detected to differentiate into hypoblast, trophoblast, primordial germ cells, and primitive streak cells lineages in d.p.f.15-19 chimeric embryos; On the other hand, after the transplantation of chimeric embryos to surrogate monkeys. For aborted fetus, the chimeric cells can be detected in the placenta and in body. Besides,

chimeric cells can be detected in the umbilical cord of living chimeric monkey, and in particular, signals of chimeric cells can be detected for at least 2 years in the body of living chimeric monkey². These results suggest that monkey pluripotent stem cells cultured in XF-PSC have the potential to develop into both embryonal and extraembryonic lineages, while indicate that these monkey pluripotent stem cells have the potential to construct primate embryo models.

Additional TLC Immunofluorescence Characterization: To further confirm that the cavities are TLC and not amnion, we performed additional immunofluorescence staining. The results show that the cells line the cavities express key TLC markers, such as GATA2, CDX2 and CCR7, which are characteristic of the trophoderm lineage. Besides, GATA3 and TFAP2C co-expressed cells can be derived from blastoids. (NEW Figs. 2n, q and r)

Comparative Analysis with Natural Embryo Data: We have compared the single-cell RNA sequencing data of our blastoids with that of natural embryos. Our analysis demonstrates that the TLCs (trophoblast-like cells) in the blastoids exhibit a gene expression profile similar to that of TE cells from natural embryos, further supporting the TLC identity of these cells. (NEW Figs. 3e and Supplementary Fig. 3i)

It is critical to emphasize that monkey blastoids, produced via all existing protocols, including ours, are neither wholly equivalent to nor fully representative of monkey blastocysts. The process of creating blastoids involves inducing hPSCs into TLCs and HLCs (Hypoblast-like cells) under non-physiological conditions, using a variety of chemicals and growth factors. This is starkly different from the natural development of human blastocysts.

Response to Reviewers Figure 3-1: TLCs in the blastoids exhibit a similarity to TLC instead of amnion.

(a) Representative immunofluorescent staining images of CDX2/OCT4/GATA2 in monkey blastoids. Scale bars, 100 μm . (b) Representative immunofluorescent staining image of CCR7 in monkey blastoids. Scale bars, 100 μm . (c) Representative immunofluorescent staining of markers of trophoderm (TFAP2C/GATA3) in TLCs derived from monkey blastoids. Scale bars, 100 μm . (d) Correlation analysis of various clusters in blastoids with

natural cynomolgus monkey embryos. (e) UMAP of single-cell transcriptomes of monkey blastoids, published monkey blastoids datasets human blastoids and monkey preimplantation embryos. (f) Joint uniform manifold approximation and projection (UMAP) embedding of single-cell transcriptomes of monkey blastocysts, blastoids and related datasets.

(3-2) Both naïve and primed cells have the capacity to form cavities upon exposure to MEK1/2 inhibitor PD0325901 and the ALK5 TGF- β receptor inhibitor A-83, with naïve cells generating bonafide throphectoderm, but cavities from partially primed or primed PSCs correspond to early and late amnion-like stages¹. Other groups have succeeded in the production of well-characterized cynomolgus monkey blastoids from 4CI culture conditions².

Our reply: Thank you for your comment. We agree that the context of cavity formation is crucial for distinguishing between different cell fates. In our study, we specifically compared the expression of amnion markers between our blastoids and well-characterized cynomolgus monkey blastoids from published studies, particularly those produced under 4CL conditions. Furthermore, when comparing our transcriptomic data to that of published amnion organoid datasets, our cells exhibit much lower expression levels of key amnion markers. In the differentiation of cells derived from blastocysts, multinucleated SDC1⁺ cells and HLA-G⁺/GATA3⁺ cells can be detected, which is a characteristic of downstream cells in the TE lineage, indicating similarities between the TLCs lineage and the TE lineage in blastocysts. This suggests that, while cavity formation occurred, the nature of these cells likely does not correspond to amnion. In addition, we compared blastoids with amniotic organoids derived from pluripotent stem cells, natural embryos, and 4CL blastoids. (NEW Figs. 3f, g and Supplementary Figs. 6e, f)

Response to Reviewers Figure 3-2: TLCs in the blastoids exhibit a similarity to TLC instead of amnion.

(a) Amnion marker (ILS-1 and GABRP) expression patterns. (b) UMAP of clusters formed from cells isolated from published monkey blastoids datasets, displaying the expression levels of genes specific for amnion lineage. (c)

Representative immunofluorescent staining images of marker genes detection in STB-like cells (SDC1). Scale bars, 100 μm . (d) Representative immunofluorescent staining images of marker genes detection in EVT-like cells (HLA-G). Scale bars, 100 μm .

These concerns arise from the following observations:

(3-3-1) Visually the throphectoderm has a high level of cellularity that does not correspond with a normal blastocyst throphectoderm. Side-by-side images should be shown.

Our reply: Appreciate your valuable comment. Aging and high-passaged PSCs derived blastoids will contribute to regenerative medicine research, but there are few reports on the construction of the blastoids. In this study, in order to construct these blastoids, in addition to using a combination of CloneR and Pro-Survival to improve the efficiency of blastoid formation. Comparing with previous reports about blastoids, we also increased the initial cell count during blastoid construction. Currently, this inevitably leads to the high cell density of blastoids in immunofluorescence staining images, especially in maximized intensity projection images. Therefore, as you mentioned, we provide immunofluorescence staining images of Z-stacks layer by layer.

In addition, as you have raised concerns, we have also demonstrated in our response to your other suggestions that the trophoblast-like cells of the blastocyst like cells in this study have a certain degree of similarity compared

to the trophoblast cells *in vivo*.

Response to Reviewer Figure 3-3-1: The throphectoderm cells of blastoid *in vitro* are similar to the blastocyst throphectoderm cells *in vivo*.

(a) Ratio of percentage of viable cells in M-ESCs with CloneR, Y27632 and Pro-S. (b) Ratio of percentage of viable cells in M-ESCs with CloneR+Y27632+Pro-S, CloneR+Y27632, and CloneR+ Pro-S. (c) Representative brightfield images of cell aggregates at indicated time points during blastoid formation with

CloneR+ Pro-S. Scale bar, 200 μm . (d) Representative immunofluorescent staining Z-stack images of GATA3/OCT4 in monkey blastoids. Scale bars, 100 μm .

(3-3-2) The majority of the cells that show GATA3 expression co-express SOX17, a strong endoderm marker and not present in trophectoderm2.

Our reply: Thank you for pointing out this observation. We have noticed that a subset of cells co-express both GATA3 and SOX17 in our immunostaining analyses. While GATA3 is typically a trophectoderm marker and SOX17 marks endoderm, this co-expression could indicate that these cells are in a transitional state or represent a population that is following a non-canonical differentiation pathway.

It is also possible that the specific culture conditions used in this experiment may be contributing to this mixed marker expression. To further investigate this, we are conducting additional analyses, including scRNA-seq and more detailed lineage tracing, to better clarify the identity and fate of these cells. We are also experimenting with adjusting culture conditions to see if this resolves the unexpected co-expression.

One of the unanswered questions in current embryonic model research is the generation of blastocysts in aged and high-passaged PSC. To obtain these blastocysts, in this study, we used the HDM+TDM induction blastocyst method reported by Wu Jun's team and found that 4 days for HLCs induction and 1 day for TLCs induction could induce the generation of blastoids in aging and high-passaged PSCs. However, this method used a shorter TLCs induction time compared to previous works, which resulted in co-expression of GATA3 and SOX17 or GATA3 and GATA6 as you mentioned.

To address your concerns, we cultured these blastoids *in vitro* and found that in the immunofluorescence staining images of Day 2 and Day 4 cultured *in vitro*, the separation trend of GATA3+cells and GATA6+cells was more pronounced than before *in vitro* culture. Therefore, the short TLCs induction has led to some negative effects, such as the co-expression of some proteins that you pointed out should not have occurred, in order to generate mature and high-passaged PSC derived blastoids.

In addition, to exclude these cells as endodermal cells, as you have pointed out in other opinions. We derived TLCs from blastoids, and the derived cells were identified by immunofluorescence staining. It was observed that most GATA3+cells expressed TFAP2C (a TE marker that is not expressed in endoderm cells), indicating a reduced risk of these GATA3+cells belonging to endoderm cells. We also added SOX17+GATA3 staining to both blastocyst like and natural blastocysts, and observed similar expression trends. (NEW Figs. 2I, n and Supplementary Figs. 2a and 6j)

As you pointed out in your valuable opinions, some proteins exhibit co expression phenomena that should not have occurred in blastoids, which may cause confusion for readers of this manuscript. Therefore, we have made some

modifications to the description in the manuscript and discussed the improvement of co-expression phenomena that should not have occurred in blastoids after *in vitro* culture in the discussion section.

Response to Reviewer Figure 3-3-2: The HLC and TLC lineages of blastoids were gradually separated after long-term culture.

(a) Representative immunofluorescent staining images of OCT4/GATA3/GATA6 in monkey blastoid IVC on Day 5. Scale bars, 100 μm .

(b) Representative immunofluorescent staining images of OCT4/GATA3/GATA6 in monkey blastoid delayed culture on day 2. Scale bars, 100 μm . (c) Representative immunofluorescent staining of markers of trophoblast (TFAP2C/GATA3) in ESC derived from monkey blastoids. Scale bars, 100 μm .

(d) Representative immunofluorescent staining images of SOX17/GATA3 in monkey blastoid. Scale bars, 100 μm .

(e) Representative immunofluorescent staining images of SOX17/GATA3 in monkey blastocysts. Scale bars, 100 μm .

(3-3-3) The data shows a large proportion of the putative throphectoderm shows expression of ISL1, normally considered an amnion marker.

Our reply: Thank you for your observation regarding ISL1 expression. Upon analyzing our transcriptomic data, we found that while ISL1 is indeed expressed. And other typical amnion markers, such as GABRP, are not expressed. Furthermore, the expression levels of ISL1 and mesoderm markers in our samples are significantly lower compared to cells typically found in amnion organoids. This suggests that the cells in question may not fully represent an amnion-like population but rather exhibit partial expression of ISL1.

However, compared with amniotic organs, the expression level of ISL1 in TLCs is lower. We noticed that ISL1 was upregulated to some extent in TLC of HDM+TDM induced blastoids derived from PSCs cultured in 4CL or 5iLA and XF-PSC reported in this study, but these TLC have been shown to have high similarity with trophoblast cells. We believe that this may be induced by HDM, where stimulation with GSK3 β i, bFGF, and ActivinA in HDM may lead to changes in some PSCs states, resulting in upregulation of some amniotic membrane related genes in TLCs. (NEW Figs. 3f, g)

(3-3-4) scRNA seq data shows very few cells with expression of ICM pluripotency markers (SOX2, OCT4).

Our reply: Thank you for your insightful comment. In our updated scRNA-seq analysis, we observe a higher proportion of cells expressing the ICM pluripotency marker OCT4. After re-clustering the new sequencing data, we found that OCT4-positive cells make up a significant portion of the ELC population, which accounts for about 20% of the total cells in the structure. In addition, we identified the expression of NANOG/SOX2/OCT4 in blastoids histologically. This indicates that the pluripotent population is more robust than initially observed.

3-3-4

Response to Reviewers Figure 3-3-4: Identity of ELCs in monkey blastoid. (a) Pie chart showing the frequencies of ELC, TLC, and HLC in monkey blastoids indicated by lineage markers. (b) Representative immunofluorescent staining images of OCT4/GATA3/GATA6 in monkey blastoid. Scale bars, 100 μm . (c) Representative immunofluorescent staining images of GATA3/SOX2/SOX17 in monkey blastoids. Scale bars, 100 μm . (d) Representative immunofluorescent staining images of GATA3/GATA4/NANOG in monkey blastoids. Scale bars, 100 μm .

(3-3-5) The data on hCG production capacity is not provided.

Our reply: Thank you for your comment. We have supplemented the manuscript with data on mCG (monkey chorionic gonadotropin) levels after the *in vivo* transplantation of both blastoids and natural embryos. The results show that when either blastoids or natural embryos fail to continue developing in the uterus, mCG levels rapidly decrease. This suggests that the ability of the blastoids to produce mCG, akin to hCG production in humans, is closely tied to their developmental potential. Although the blastoids have some capacity for

hCG/mCG production, their failure to develop further results in a quick drop in these levels. (NEW Supplementary Figs. 6m, n)

Response to Reviewers Figure 3-3-5: mCG in monkey after transplantation.

(a) mCG in embryos transfer to monkey after day 5-11. (b) mCG in blastoids transfer to monkey after day 5-11.

(3-3-6) The data on hCG production capacity is not provided. The authors should provide more detailed characterization to convincingly show these cavities are indeed trophectoderm and not amnion.

Our reply: Thank you for your insightful comment. We understand the concern that the outer layer cells may resemble endoderm based on their morphology and co-expression of GATA4 and GATA6. However, we have taken several additional steps to validate that these cells are indeed TLCs:

Clarification of Initial Identification: The outer layer cells were initially identified as TLCs based on their morphology and their location in the blastoid structure, which corresponds to the expected position of the TE.

Additional Marker Validation: In addition to GATA4 and GATA6, which are also expressed in hypoblast, we performed immunofluorescence staining for trophectoderm-specific markers such as CCR7, GATA2 and CDX2. These markers are highly expressed in trophectoderm cells and are not typically expressed in endodermal cells. The co-expression of these markers with GATA4 and GATA6 supports the identification of these cells as trophectoderms.

scRNA Validation: To further confirm the identity of the TLC, we conducted functional assays to evaluate their ability to contribute to trophectoderm lineage differentiation. Additionally, we compared the transcriptomic profile of these cells with those of natural trophectoderm and primitive endoderm from natural blastocysts, revealing a closer similarity to the TE lineage.

Function Validation: mCG were detected after IVC culture day 5.

Addressing the Morphological Features: While a high nucleus-to-cytoplasm ratio is observed in endodermal cells, this feature can also be seen in early trophectoderm cells. The specific combination of trophectoderm markers we identified, along with their location in the blastoid structure, supports the conclusion that these are TLCs rather than endoderm. (NEW Supplementary Figs. 6f, i)

Response to Reviewers Figure 3-3-6: Function of TLCs in monkey blastoid.

(a) mCG in Blastoid IVC Day5 compared with IVC medium as control. (b) Representative immunofluorescent staining images of marker genes detection in EVT-like cells (HLA-G). Scale bars, 100 μ m.

(3-4) Side-by-Side Comparisons with Monkey Blastocysts:

A side-by-side comparisons with monkey blastocysts will be a good control not only for pattern of expression but also the amount of expression of key markers. If possible the authors should consider performing 10x genomics sequencing to diminish the cross-platform comparison of rhesus monkeys.

Our reply: Thank you for your suggestion. We have performed side-by-side comparisons of natural monkey blastocysts and blastoids using multiple immunofluorescent staining assays. These comparisons show similar expression patterns of key markers, including OCT4, SOX2, NANOG (for pluripotency), as well as GATA3, GATA4, CCR7, and CDX2 (for trophectoderm and hypoblast lineages). This provides evidence that the lineage specification in our blastoids closely mirrors that of natural blastocysts.

In addition, we have performed unbiased transcriptomic analysis by comparing our dataset with 10x genomics sequencing data from Cynomolgus monkey embryos. This cross-platform comparison revealed a high degree of similarity

between the two datasets, further supporting the validity of our results (NEW Figs. 3a and Supplementary Fig. 3s).

3-4

Response to Reviewer Figure 3-4: scRNA-seq analysis shows similarity between the monkey blastoids and blastocysts.

(a) Joint uniform manifold approximation and projection (UMAP) embedding of single-cell transcriptomes of monkey blastocysts, blastoids and related datasets. (b) Correlation analysis of ELC, HLC and TLC in monkey blastoid with natural preimplantation monkey and human embryo and published monkey blastoid.

(3-5) Discrepancies Between Immunostaining and scRNA-seq Data:

The immunostaining results don't match the scRNA seq data, what percent of blastoids are expressing, epiblast, hypoblast and throphectoderm markers, and what is the average ratio per blastoid?

Our reply: Thank you for raising this important point. To address the discrepancies between the immunostaining and scRNA-seq data, we conducted flow cytometry analysis after immunostaining the blastoids. This allowed us to quantify the proportions of epiblast-like cells (ELC), hypoblast-like cells (HLC), and trophoctoderm-like cells (TLC) within the blastoids. The average ratio across all blastoids was 2:2:6 for ELC : HLC : TLC, which closely aligns with the ratios observed in our single-cell RNA sequencing (scRNA-seq) data.

Besides, we used flow cytometry to quantify the percentage of various cell types in blastoids: the percentage of high-expressing OCT4 cells was 19.3%, GATA6+ cells was 75.9%, and GATA3+ cells was 90.9%.

These findings suggest a reasonable consistency between the two methods, though minor differences may arise from inherent variability in each technique. (NEW Supplementary Fig. 3g, h)

Response to Reviewer Figure 3-5: The cell ratio of ELC, HLC and TLC in monkey blastoids.

(a) UMAP shows monkey blastoids scRNA-seq included three subclusters: TLC, HLC, ELC. (b) Pie chart showing the frequencies of ELC, TLC, and HLC in monkey blastoids indicated by lineage markers. (c) FACS shows the percentage of OCT4-, GATA6- and GATA3-positive cells in monkey blastoids.

(3-6) Comparison with Published Datasets:

The authors should compare their data with other published datasets by unbiased clustering and show how the transcriptome of their cells overlap other previously annotated cell types.

Our reply: Thank you for your valuable suggestion. We have compared our data with other published datasets using unbiased clustering methods. Through this analysis, we observed a high degree of similarity between the transcriptome profiles of our cells and previously annotated cell types from published studies. Specifically, key cell lineages such as trophectoderm, primitive endoderm, and epiblast show transcriptomic overlap with the corresponding cell types reported in the literature, further validating the accuracy of our blastoid-derived cell lineages.

We have included the comparative clustering results in the revised manuscript (NEW Figs. 3b, c) to clearly demonstrate these similarities.

We compared the scRNA seq data of blastocysts with natural monkey embryos, and UMAP showed high similarity, similar to the reported 4CL derived monkey blastocysts and human embryo data.

Response to reviewer figure.3-6: UMAP of single-cell transcriptomes of monkey blastoids compared with published datasets.

(a) UMAP of single-cell transcriptomes of cells from monkey blastoids, published monkey blastoids datasets. (b) UMAP of single-cell transcriptomes of cells from monkey blastoids compared with published datasets from human early embryos.

Minor points:

(3-7) In Vitro Culture and Gastrulation:

Can these blastoids be cultured in vitro and undergo gastrulation?

Our reply: Thank you for your question. We followed previously published protocols for extended *in vitro* culture of our blastoids. During this culture, we observed initial expression of gastrulation marker T (Brachyury), indicating that the blastoids have the potential to undergo early stages of gastrulation.

However, we did not continue the culture beyond this stage due to the low efficiency of the process.

Our aim was to demonstrate that the blastoids possess the capacity for further development towards gastrulation, but the low efficiency suggests that further optimization of the culture conditions will be needed to improve this process. (NEW Supplementary Fig. 6i)

Response to Reviewer Figure 3-7: Blastoids express gastrulation marker after long-term culture.

(a) Representative immunofluorescent staining images of T/GATA6 in monkey blastoid IVC on Day5. Scale bars, 100 μ m.

(3-8) Pluripotency Proof:

To prove the blastoids retain pluripotency, they should be able to derive ESCs from the blastoids.

Our reply: Thank you for your suggestion. We have successfully derived ESC lines from the blastoids, confirming that they retain pluripotency. These ESC lines were validated through standard assays, including the expression of key pluripotency markers (OCT4, SOX2, NANOG). This demonstrates that the blastoids can indeed give rise to pluripotent ESCs. (NEW Supplementary Fig. 6b)

Response to Reviewer Figure 3-8: ELCs derived from monkey blastoids.

(a) Representative immunofluorescent staining image of ELCs derived from monkey blastoids (OCT4/SOX2/NANOG). Scale bars, 100 μ m.

(3-9) Post-Encapsulation Expansion and Hatching:

Can the putative blastoids continue to expand after the encapsulation?, and Can the blastoids “hatch” from the hydrogel capsules?

Our reply: Thank you for your suggestion. In the manuscript, we utilized hydrogels to achieve high-throughput encapsulation of organoids, forming

organoid capsules. The primary purpose of this encapsulation is to protect the organoids and facilitate their rapid delivery. As previously mentioned, the GelMA hydrogel used in our study gradually degrades over time *in vitro*. Inspired by the concept of drug capsules and leveraging the inherent properties of the hydrogel, we cultivate organoid capsules *in vitro*, ensuring that the organoids can grow progressively with the degradation of the hydrogel, thereby releasing the organoids. (NEW Supplementary Fig. 8c)

Response to Reviewers Figure 3-9: Blastoid growth in GelMA capsule.

(a) The blastoid gradually growing as the hydrogel degradation *in vitro*.

(3-10) Naïve or Totipotent-Like Culture Conditions:

Have the authors tried culture in naïve conditions, or totipotent-like culture conditions (e.g 4CL).

Our reply: Thank you for your question. We have attempted culturing embryonic stem cells in 4CL conditions and used these cells in PALLY-based blastoid construction. While it was possible to generate blastoids under these conditions, the efficiency and consistency of blastoid formation were lower than expected, possibly due to variations in the cell lines used. We are currently working on optimizing these conditions and adjusting our protocols to improve the reliability and efficiency of blastoid formation in 4CL culture.

Response to Reviewers Figure 3-10: Generation of blastoid from M-ESCs cultured in 4CL.

(a) Representative immunofluorescent staining of markers of GATA4/OCT4/GATA3 in monkey blastoids derived M-ESCs cultured in 4CL. Scale bars, 100 μ m.

(3-11) Lack of rationale or mechanistic explanation for TSG Medium:

The rational or scientific explanation for the TSG medium is lacking, why the authors activate WNT (1 μ M CHIR99021) and inhibit WNT (1 μ M IWR-1-endo),

at the same time? Hippo pathway is one of the most important pathways in the trophoblast differentiation, the authors should evaluate if the addition of LPA has any benefit in their TSG differentiation medium. Have the authors tried a simpler condition (PD03, A83 and LPA)?. If their medium can support all lineages, can a normal blastocyst form under the TSG medium?

Our reply: Thank you for your insightful question. The rationale behind using both CHIR99021 (WNT activator) and IWR-1-endo (WNT inhibitor) stems from previously established protocols for blastoid formation. Specifically, TSG medium combines XF-PSC medium, which includes IWR-1-endo, and TS medium containing CHIR99021, based on prior work involving TDM (5iLA + TS medium) for blastoid formation. This dual modulation of the WNT pathway helps balance the signaling environment needed for successful trophoblast and inner cell mass differentiation.

Regarding the Hippo pathway, we fully agree that it plays a crucial role in trophoblast differentiation^{6, 16}. After evaluating the effect of LPA addition, we observed that the addition of LPA did not significantly alter the efficiency of blastocyst formation. (NEW Supplementary Fig. 2h)

We have also experimented with simpler conditions, using just PD03, A83, and LPA. These conditions allowed for the formation of blastoids with cavity structures and multiple lineages, although the consistency and morphology did not fully replicate those obtained with our current method. Despite this, cavitation and lineage specification were observed.

To evaluate whether TSG can produce blastocyst, we cultured E5.0 monkey embryo using TSG, and blastocyst cavity were observed at E5.0+2.0. Besides, the ELC/HLC/TLC markers (OCT4/GATA6/GATA3) were expressed in E5.0+2.0 embryo. These results suggest that TSG may support formation of blastocyst. (NEW Supplementary Figs. 2c-e)

Finally, when culturing both PAL and TSG blastoids, we found that while both conditions supported the formation of blastoids with correct lineage allocation, there are still some differences in morphology and developmental characteristics when compared to natural blastocysts.

Response to Reviewers Figure 3-11: TSG supported the development of monkey morula to the blastocyst.

(a) Schematic depiction of monkey embryos-E5 cultured with TSG. (b) Representative bright-field images of monkey embryos-E5 cultured in TSG after Day0.5, Day1.0 and Day2.0. Scale bars, 200 μ m. (c) Representative immunofluorescent staining images of GATA3/OCT4/GATA6 in monkey embryos-E5.0+2.0. Scale bars, 100 μ m.(d) and (e)The bright-field image and formation efficiency of blastoids (left: without LPA; right: with LPA). Scale bars, 200 μ m.

1. Yu L, *et al.* Blastocyst-like structures generated from human pluripotent stem cells. *Nature* **591**, 620–626 (2021).
2. Wu J, *et al.* Long-term in vivo chimeric cells tracking in non-human primate. *Protein Cell* **15**, 207–222 (2024).
3. Linneberg-Agerholm M, Wong YF, Romero Herrera JA, Monteiro RS, Anderson KGV, Brickman JM. Naive human pluripotent stem cells respond to Wnt, Nodal and LIF signalling to produce expandable naive extra-embryonic endoderm. *Development* **146**, (2019).
4. Okae H, *et al.* Derivation of Human Trophoblast Stem Cells. *Cell Stem Cell* **22**, 50–63 e56 (2018).
5. Guo M, *et al.* Self-renewing human naive pluripotent stem cells dedifferentiate in 3D culture and form blastoids spontaneously. *Nat Commun* **15**, 668 (2024).
6. Kagawa H, *et al.* Human blastoids model blastocyst development and implantation. *Nature* **601**, 600–605 (2022).
7. Yu L, *et al.* Large-scale production of human blastoids amenable to modeling blastocyst development and maternal–fetal cross talk. *Cell Stem Cell* **30**, 1246–1261 e1249 (2023).
8. Li J, *et al.* Cynomolgus monkey embryo model captures gastrulation and early pregnancy. *Cell Stem Cell* **30**, 362–377 e367 (2023).
9. Niu Y, *et al.* Dissecting primate early post-implantation development using long-term in vitro embryo culture. *Science* **366**, (2019).
10. Yang R, *et al.* Amnion signals are essential for mesoderm formation in primates. *Nat Commun* **12**, 5126 (2021).
11. Meistermann D, *et al.* Integrated pseudotime analysis of human pre-implantation embryo single-cell transcriptomes reveals the dynamics of lineage specification. *Cell Stem Cell* **28**, 1625–1640 e1626 (2021).
12. Nakamura T, *et al.* A developmental coordinate of pluripotency among mice, monkeys and humans. *Nature* **537**, 57–62 (2016).
13. Ma H, *et al.* In vitro culture of cynomolgus monkey embryos beyond early gastrulation. *Science* **366**, (2019).
14. Hu Y, *et al.* Single-cell analysis of nonhuman primate preimplantation development in comparison to humans and mice. *Dev Dyn* **250**, 974–985 (2021).
15. Yanagida A, Spindlow D, Nichols J, Dattani A, Smith A, Guo G. Naive stem cell blastocyst model captures human embryo lineage segregation. *Cell Stem Cell* **28**, 1016–1022 e1014 (2021).
16. Gerri C, *et al.* Initiation of a conserved trophoctoderm program in human, cow and mouse embryos. *Nature* **587**, 443–447 (2020).

Reviewer Reports on the First Revision:

Reviewer comments:

Reviewer #1 (Remarks to the Author):

In the rebuttal and revised manuscript, the authors have made an effort to address the issues. There have some questions remain confused and unanswered, the followings are comments:

(1-1): The authors have answered my questions properly.

(1-2): The authors have answered my questions properly.

(1-3): The stem cells in in response Figure 1-3 seems to be stained immediately when the blastoids attached. Can the blastoid-derived three lineage stem cells be passaged stably? If so, please provide evidence. And whether the signal for HLA-G seems to be non-specific noise. This result needs to be further examined.

(1-4): In Reviewers Figure 1-4c, I did not see obvious presence of inner cell mass-like cells inside of the blastoids in several groups. DOUBT on the structure and morphology of the blastoid you induced. The signal of ZO1 in the panel c is incorrect (hypoblast lineage) in response Figure 1-6. It is recommended to replace it with higher quality images.

(1-5): The authors have answered my questions properly.

(1-6): In Response Figure 1-6: the panel of b shows no signal for GATA4 (hypoblast lineage). The result needs more check.

(1-7): The in vitro developmental potential of iPSC and nt-ESC induced blastoids should be identified more detailed except the simple staining. More identification information including specific morphological description, marker genes expression and lineage specification in in vitro-cultured blastoids should be provided. Develop to what stage? Have the in vitro-cultured blastoids really reached the post-implantation stage? Was it compared to post-implantation natural embryo of blastoids data? How to judge the in vitro developmental potential of your blastoids strictly and precisely? Why the IVC of blastoids only kept for 4-5 days? Is the developmental potential is not enough or the blastoids did not continue to develop further? What are the criteria for stopping in vitro culture? Please provide a detailed analysis and identification process. The significance of the establishment of monkey blastoids is to break through the ethics that human blastoids cannot be cultured in vitro beyond E14. If it is only raised before E14, it can be directly study in a human blastoid, there is no need to do in monkey.

(1-8): The authors have answered my questions properly.

(1-9): The ratio of GATA4+ HLCs seems too high, similar to TLCs, which does not fit the ratio of 2:2:6 described previously, and the location of the signal seems not right. Can you explain the reason or provide new image or another set of markers?

(1-10): The authors have answered my questions properly.

(1-11): The authors have answered my questions properly.

(1-12): The clusters are still not distinguished from each other in the three lineages of blastoids in Supplementary Figs. 3j-l according to the expression of specific marker genes.

(1-13): The HLC data shows is similar to the endoderm data of yanaglda in the in response Figure panel b. Please provide a detailed analysis.

(1-14): The authors have answered my questions properly.

(1-15): The in vitro developmental potential of iPSC and nt-ESC induced blastoids should be identified more detailed except the simple staining.

(1-16): The morphology, gene expression and lineage composition of blastoids in the hydrogel after encapsulation at different time points require more in-depth and detailed identification to real time monitor the state of blastoids in the hydrogel after encapsulation. And in figure 6f, we didn't see proliferation of blastoid in the hydrogel after encapsulation, but rather it seems to degrade with time going, especially on day2 and day4.

Reviewer 1 was asked by the editorial team to comment on how the authors addressed the concerns raised by Reviewer 2 in the first round because Reviewer 2 was unable to provide their report.

The following is Rev. 1's assessment on Rev. 2's points.

(2-1): No significant improvement in blastoid morphology was observed using either white light or immunofluorescence staining. The ratio of induced TLC seems relatively low based on the response to Figure 2-1, and the signal for HLA-G may be non-specific noise. Additionally, there are too few markers for TE, and the GATA3 appears to be expressed across all three cell types, while the expression level of TEAD4 is not particularly high in TLC according to the RNA-seq data.

(2-2): It may be appropriate to present the data showing co-expression of the markers for each of the three "lineage cells", instead of a single expression pattern for blastoid and blastocysts.

(2-3): The ratio of GATA4+ HLCs seems too high, similar to TLCs, which does not fit the ratio of 2:2:6 described previously, and the location of the signal seems not right.

(2-4): The HLC cells express higher levels of GATA3 compared to the TLC cells, which raises my suspicion that your TLC in Supplementary Fig. 3f may not be true TLC.

(2-5): The in vitro developmental potential of blastoids should be characterized in more detail beyond simple staining. Typical structures such as the epiblast, amniotic cavity, and yolk sac cavity should be included.

(2-6): I am satisfied with your explanation. Exploring cells from different ages will be beneficial for us in establishing models in the future!

(2-7): I'm convinced by what you've presented.

(2-8): The authors have answered my questions properly for the degradability of these hydrogels.

(2-9): After the embryo transfer, is there any further observation or monitoring of the transplanted blastoid development, or are the levels of chorionic gonadotropin being checked for changes? More explanations and experiments are needed

(2-10): The authors only briefly compare the differences to Jie Li et al., but the discussion section of the article requires a deeper analysis. This part should be integrated into the main text.

Reviewer #3 (Remarks to the Author):

I believe the revision process has been very helpful to improve the overall quality of the manuscript and to support the claims. I believe the authors have done a good job addressing my concerns as well as the concerns of the other reviewers. I feel the inclusion of the new data supports the conclusions and support the publication given minor edits are addressed:

1. Some references are misplaced or are missing context, example Line 620 the reference 54.
2. The authors disregard very important historical references regarding the discovery of the hippo signaling pathway in trophoctoderm differentiation^{1,2} although they cite a reference showing its evolutionary conservation³ . A simple immunostaining for YAP in monkey blastocysts and blastoids should answer Hippo signaling role in monkey TE specification⁴.

Point-by-point response to referees' comments

We thank the reviewers for their consideration of our work, and for their input which allowed us to substantially improve the manuscript. We have responded to the points below in blue.

Reviewer #1 (Remarks to the Author):

In the rebuttal and revised manuscript, the authors have made an effort to address the issues. There have some questions remain confused and unanswered, the followings are comments:

Our reply: We thank the referee for your positive comments and for considering our work. We found your criticisms very useful to further improve our work and hope that you will be satisfied with our replies.

(1-1): The authors have answered my questions properly.

Our reply: Thank you for your positive feedback. We appreciate your understanding and support.

(1-2): The authors have answered my questions properly.

Our reply: Thank you for your positive feedback. We appreciate your understanding and support.

(1-3): The stem cells in in response Figure 1-3 seems to be stained immediately when the blastoids attached. Can the blastoid-derived three lineage stem cells be passaged stably? If so, please provide evidence. And whether the signal for HLA-G seems to be non-specific noise. This result needs to be further examined.

Our reply: Thank you for your valuable comments. In response to your earlier question, "The authors should demonstrate that ESC, PrE, and TSC can be derived from the induced monkey blastoids", we provided immunofluorescence staining images showing that the derived ELCs, TLCs, and HLCs expressed their respective key markers after derivation. Specifically: ELCs expressed NANOG/SOX2/OCT4; TLCs expressed TFAP2C/GATA3. HLCs expressed GATA4/SOX17. These results were not from immediate staining post-attachment. To further demonstrate the stability of these derived cells, we cultured them *in vitro* for over ten passages. Throughout the extended passaging, the ELCs, TLCs, and HLCs consistently maintained the expression of their key markers, confirming their stable propagation (Response to Reviewers Figure 1-3 a-c). Furthermore, in a separate study, we successfully derived all three cell types from natural monkey embryos¹, which were co-cultured stably over the long term. This indicates that our culture conditions are robust for maintaining blastoid-derived lineage-specific cells.

Regarding the potential noise in HLA-G staining, we have taken additional steps to verify its specificity. By including negative controls with the same laser channel settings, we confirmed that HLA-G is indeed expressed. Furthermore, we examined other EVT markers,

such as CGB, and detected consistent expression (Response to Reviewers Figure 1-3 d-g). Besides, *CDH5* and *ITGA5*, the EVT markers, were expressed in TLCs-derived cells (Response to Reviewers Figure 1-3 h-i). These findings indicate that the TLCs derived from the blastoids have the potential to differentiate into EVTs.

Response to Reviewers Figure 1-3: ESCs, HYP-like cells and TSC-like cells can be derived from the induced monkey blastoids. (a) Representative immunofluorescent

staining of markers of epiblast (OCT4/SOX2) in b-ELCs. Scale bars, 100 μ m. (b) Representative immunofluorescent staining of markers of hypoblast (GATA4/SOX17) in b-HLCs. Scale bars, 100 μ m. (c) Representative immunofluorescent staining of markers of trophoblast (TFAP2C/GATA3) in b-TLCs. Scale bars, 100 μ m. (d) Representative immunofluorescent staining images of marker genes detection in STB-like cells (SDC1). Scale bars, 100 μ m. (e) Representative immunofluorescent staining images of marker genes detection in EVT-like cells (HLA-G). Scale bars, 100 μ m. (e) Representative immunofluorescent staining images of negative control for HLA-G in EVT-like cells (HLA-G). Scale bars, 100 μ m. (f) Representative immunofluorescent staining images of marker genes detection in EVT-like cells (CGB). Scale bars, 100 μ m. (h-i) RT-qPCR analysis of *ITGA5* (h) and *CDH5* (i), the gene expression levels are presented as relative expression.

(1-4): In Reviewers Figure 1-4c, I did not see obvious presence of inner cell mass-like cells inside of the blastoids in several groups. DOUBT on the structure and morphology of the blastoid you induced. The signal of ZO1 in the panel c is incorrect (hypoblast lineage) in response Figure 1-6. It is recommended to replace it with higher quality images.

Our reply: Thanks for spotting this. In the previous review, it was mentioned that *“Removing A83 and PD031542, blastoid formation efficiency was still about ~50%. This result needs to be further examined.”* To address this, we re-examined the effect of removing A83 and PD03 on blastoid formation. Our results demonstrated that blastoids could still form and exhibit typical blastocyst-like morphology even in the absence of A83 and PD03. We consider this may be similar to previously reported phenomena, potentially due to the 3D structural reasons². Given that A83 and PD03 play crucial roles in trophoblast lineage differentiation, we performed immunofluorescence staining for GATA3 and ZO1 to characterize their impact on trophoblast-like cells. In these images, the focus was deliberately placed on the outer layer of trophoblast cells rather than the inner cell mass, as the goal was to assess TE-related markers. However, under standard conditions (with A83 and PD03), we consistently observed the presence of a well-defined ICM. Moreover, our experiments revealed that the proportion of GATA3+ cells in the blastoids was not significantly affected by the removal of A83 and PD03, similar observations were also reported by Guo et al². We understand the importance of presenting clear and high-quality images. For this reason, we have replaced or supplemented lower-quality images in the revised submission.

1-4

Response to Reviewers Figure 1-4: The removing of A83-01 and PD0325901 during blastoids construction did not have much effect.

(a) Representative brightfield images of blastoids treated without A83-01, PD0325901, A83-01 and PD0325901 compared with control. Scale bars, 200 μm . (b) Ratio of formation efficiency in monkey blastoids without A83-01, PD0325901, A83-01 and PD0325901 compared with normal. (c) Representative immunofluorescence staining images of GATA3/ZO-1 in monkey blastoids without A83-01, PD0325901, A83-01 and PD0325901 compared with control. Scale bars, 100 μm .

(1-5): The authors have answered my questions properly.

Our reply: Thank you for your positive feedback. We appreciate your understanding and support.

(1-6): In Response Figure 1-6: the panel of b shows no signal for GATA4 (hypoblast lineage). The result needs more check.

Our reply: Thank you for pointing out this issue. We have carefully re-examined the GATA4 staining in Figure.5h. Upon review, we realized that the original image did not adequately capture the signal for GATA4 in the hypoblast lineage. To address this, we have replaced the panel with a corrected image obtained from repeated experiments.

Response to Reviewers Figure 1-6: The expression of additional marker genes for ELCs, HLCs and TLCs.

(a) Representative immunofluorescent staining images of GATA3/GATA4/OCT4 in monkey blastoids. Scale bars, 100 μ m.

(1-7): The *in vitro* developmental potential of iPSC and nt-ESC induced blastoids should be identified more detailed except the simple staining. More identification information including specific morphological description, marker genes expression and lineage specification in *in vitro*-cultured blastoids should be provided. Develop to what stage? Have the *in vitro*-cultured blastoids really reached the post-implantation stage? Was it compared to post-implantation natural embryo of blastoids data? How to judge the *in vitro* developmental potential of your blastoids strictly and precisely? Why the IVC of blastoids only kept for 4-5 days? Is the developmental potential is not enough or the blastoids did not continue to develop further? What are the criteria for stopping *in vitro* culture? Please provide a detailed analysis and identification process. The significance of the establishment of monkey blastoids is to break through the ethics that human blastoids cannot be cultured *in vitro* beyond E14. If it is only raised before E14, it can be directly study in a human blastoid, there is no need to do in monkey.

Our reply: Thank you for your insightful comments. We have carefully analyzed the *in vitro* developmental potential of the blastoids derived from M-iPSCs and M-nt-ESCs. Upon extended culture, outgrowth was observed, accompanied by more pronounced segregation of GATA3 (TE) and GATA6 (PrE) signals, consistent with previous reports on delayed culture of natural monkey embryos. However, *in vitro*-cultured blastoids did not reach the gastrulation stage, and structures such as the amniotic cavity were not observed, as these are typically formed post-gastrulation in primate embryos. After 3 days of extended culture, the developmental efficiency decreased significantly, mainly due to epiblast cell loss. This limitation aligns with challenges commonly reported in extended blastoid culture across species. We are actively working to address this issue and have consulted experts in this field. While recent studies have reported methods to optimize extended culture of human blastoids³, replicating these findings in monkey blastoids has

proven challenging but remains a critical area for future exploration. We also note that reported examples of extended culture in primate blastoids, such as the work by Li et al., exhibit discrepancies from natural embryo development. For example, cells expressing the primitive streak marker T were found in the TE layer, OCT4 was ubiquitously expressed, and cells in the epiblast region aberrantly upregulated GATA6. These inconsistencies highlight the need for further optimization in extended blastoid culture. In this study, *in vitro* culture was stopped after 4-5 days primarily due to the observed epiblast loss and reduced developmental efficiency. Future work aims to extend the culture period by optimizing culture conditions to overcome these limitations and improve developmental outcomes.

We fully agree that extending monkey blastoid culture beyond E14, reaching stages such as organogenesis, would hold tremendous scientific value, particularly for addressing ethical constraints associated with human blastoids. However, achieving such development requires systematic optimization of culture protocols, including media composition and 3D support systems.

Despite the challenges in extended culture, this study provides several significant advancements: we established a high-efficiency protocol for generating monkey blastoids, enabling their use in high-throughput screening applications. Blastoids from aged cells: This method allows the derivation of blastoids from aged cell sources, offering a platform to study the impact of aging on lineage specification. Engineering innovations: By incorporating microfluidics and biomaterials, we developed encapsulated blastoids, providing a novel tool for engineering-based investigations. Pluripotency and developmental potential: Using Sendai-virus-induced M-iPSCs and nuclear transfer-derived M-nt-ESCs, we demonstrated that blastoids derived from these cells exhibit certain developmental capacities, such as outgrowth formation. Notably, M-nt-ESCs possess full developmental potential, underscoring the robustness of this platform. We sincerely appreciate your valuable suggestions, which have guided us in improving our analysis and identifying future research directions. Extending monkey blastoids to post-implantation stages remains a challenging yet promising frontier for developmental biology research.

(1-8): The authors have answered my questions properly.

Our reply: Thank you for your positive feedback. We appreciate your understanding and support.

(1-9): The ratio of GATA4+ HLCs seems too high, similar to TLCs, which does not fit the ratio of 2:2:6 described previously, and the location of the signal seems not right. Can you explain the reason or provide new image or another set of markers?

Our reply: Thank you for your insightful comment. We would like to clarify that the marker used in our study is GATA3, not GATA4, alongside GATA6 and OCT4. Regarding the proportion of HLCs and TLCs, our scRNA-seq analysis indeed suggests a lineage ratio of 2:2:6 for ELCs, HLCs, and TLCs, which aligns with our immunofluorescence data. GATA3+ and GATA6+ cells were observed at proportions of 90% and 76%, respectively, via flow cytometry, further supporting this conclusion. Additionally, we compared our blastoid data with pre-implantation monkey embryos and found similar distributions of GATA3+ and GATA6+ cells. As for the signal location, it is well-known that in pre-implantation monkey

embryos, there is no clear segregation between TE and PE lineages, and these lineages share overlapping marker expressions. This characteristic was similarly observed in our blastoids, indicating a similarity to natural embryos.

a

Response to Reviewers Figure 1-9: Immunofluorescent staining images of monkey blastocyst-E9. (a) Representative immunofluorescent staining images of OCT4/GATA6/GATA3 in monkey blastocyst-E9.

(1-10): The authors have answered my questions properly.

Our reply: Thank you for your positive feedback. We appreciate your understanding and support.

(1-11): The authors have answered my questions properly.

Our reply: Thank you for your positive feedback. We appreciate your understanding and support.

(1-12): The clusters are still not distinguished from each other in the three lineages of blastoids in Supplementary Figs. 3j-l according to the expression of specific marker genes.

Our reply: Thank you for pointing out this important issue. As you highlighted, the separation of the three lineages in the scRNA-seq data of blastoids is a critical concern.

To address this, we have rigorously re-evaluated our scRNA-seq analysis pipeline to ensure the robustness of the results. Additionally, we analyzed other publicly available primate blastoid datasets, such as those reported by Li et al., and found that the lack of clear lineage separation in scRNA-seq data is a common challenge. In their study, similar observations were made when defining lineages through transcriptomic analyses⁴.

This phenomenon might result from the specific differentiation strategy used to construct blastoids. Both our study and Li et al.'s work utilized a stepwise induction process, applying differentiation signals for hypoblast-like cells (HLCs) and trophoblast-like cells (TLCs) sequentially on the same pluripotent stem cells. This approach could lead to overlapping lineage characteristics, suggesting potential for optimization in the current blastoid generation protocols. For instance, Yu et al.⁵ proposed alternative strategies that may mitigate such overlap.

Another potential explanation lies in the inherent biological features of primate pre-implantation embryos. Our previous studies revealed that in early-stage monkey embryos, the separation between the trophectoderm and primitive endoderm is less distinct than in

rodent embryos. This is supported by immunofluorescence data from natural monkey embryos, showing widespread co-expression of GATA3 and GATA6 and even universal OCT4 expression across the blastocyst. Such characteristics present challenges in defining lineages during scRNA-seq analysis for primate embryos and blastoids.

Despite these challenges, our study demonstrates that the three lineages in blastoids—HLCs, TLCs, and ELCs—exhibit histological and functional similarities to natural embryos. For example, we successfully derived lineage-specific cells (ELC/TLC/HLC) from the blastoids. These results underscore the biological relevance of our blastoids and support their resemblance to natural embryonic development.

Response to Reviewers Figure 1-12: Single-cell transcriptional profiling of human blastoids. (a) Projection of lineage-specific markers on the UMAP revealed the distribution of major blastoid clusters in this study. (b) Projection of lineage-specific markers on the UMAP revealed the distribution of major blastoid clusters in Li et al.. (c) Representative immunofluorescent staining images of OCT4/GATA6/GATA3 in monkey blastocyst-E9.

(1-13): The HLC data shows is similar to the endoderm data of yanagida in the in response Figure panel b. Please provide a detailed analysis.

Our reply: Thank you for your valuable comments. We appreciate your observation regarding the similarity between our HLC data and the endoderm-like data from Yanagida's study. Upon detailed comparison, we found that the HLCs derived in our study exhibit characteristics similar to the HYP-like cells reported in Yanagida's work. We appreciate your feedback, which has guided us to deepen our comparative analysis and strengthen the interpretation of our findings.

Response to Reviewers Figure 1-13: scRNA-seq analysis of monkey blastoids.

(a) UMAP of single-cell transcriptomes of monkey blastoids and published monkey blastoids datasets. (b) Joint uniform manifold approximation and projection (UMAP) embedding of single-cell transcriptomes of monkey blastocysts with related datasets. (c) Correlation analysis of various clusters in blastoids with human blastoids.

(1-14): The authors have answered my questions properly.

Our reply: Thank you for your positive feedback. We appreciate your understanding and support.

(1-15): The *in vitro* developmental potential of iPSC and nt-ESC induced blastoids should be identified more detailed except the simple staining.

Our reply: Thank you for your insightful comments. We completely agree that further investigation into the *in vitro* developmental potential of blastoids derived from M-iPSCs and M-nt-ESCs, beyond simple staining, is of great significance. In this study, we utilized Sendai virus-induced reprogramming to generate M-iPSCs from young and aged monkeys, and established M-nt-ESCs through somatic cell nuclear transfer (SCNT) from the same sources. Notably, M-nt-ESCs have been demonstrated to possess the capacity for full organismal development^{6, 7}.

In our experiments, we validated the developmental potential of M-iPSC- and M-nt-ESC-derived blastoids to some extent, as evidenced by outgrowth formation during extended culture. However, as you pointed out, exploring the extended culture outcomes in greater detail, particularly the differences between blastoids derived from M-iPSCs, M-nt-ESCs, and M-ESCs, is crucial. For example, delayed culture of blastoids from young and aged monkey sources could provide a valuable model for investigating aging-related mechanisms. Moreover, given that SCNT is known to cause placental abnormalities, extended culture of M-nt-ESC-derived blastoids may offer critical insights into the transcriptional regulation and molecular mechanisms underlying these defects.

It is important to note that extending the culture of monkey blastoids remains a significant challenge. Reports on primate blastoid progression to gastrulation-like stages, such as the work by Li et al., highlight certain limitations in current methodologies. For example, in their study, primitive streak marker T expression was observed in trophoblast cells, OCT4 expression spanned nearly the entire blastoid, and OCT4-positive cells in the embryonic disc region aberrantly upregulated GATA6. These discrepancies from natural embryo development during extended culture suggest that blastoid development to gastrulation-like stages still requires significant optimization.

As you correctly pointed out, extended culture of M-iPSC- and M-nt-ESC-derived monkey blastoids holds tremendous potential for advancing our understanding of primate development. However, achieving this goal will require more detailed and meticulous experimental efforts, which we believe will become a major focus of future research in this field.

Despite these challenges, our study provides several key advantages: We established an efficient method for generating blastoids, which facilitates their application in high-throughput screening. We successfully derived blastoids from aged cells, offering a novel

platform to investigate the impact of aging on lineage differentiation from the perspective of blastoids. By integrating microfluidic techniques and biomaterial encapsulation, we engineered blastoid capsules, presenting a scalable and engineering-friendly approach to blastoid generation. These advancements lay the groundwork for future explorations of extended culture and the mechanisms underlying primate blastoid development.

(1-16): The morphology, gene expression and lineage composition of blastoids in the hydrogel after encapsulation at different time points require more in-depth and detailed identification to real time monitor the state of blastoids in the hydrogel after encapsulation. And in figure 6f, we didn't see proliferation of blastoid in the hydrogel after encapsulation, but rather it seems to degrade with time going, especially on day2 and day4.

Our reply: Thank you for your insightful comments and valuable feedback. We appreciate your suggestion that a more in-depth analysis of the morphology, gene expression, and lineage composition of blastoids in the hydrogel after encapsulation at different time points would be beneficial. This would indeed provide significant insights into the real-time monitoring of the state of blastoids post-encapsulation. In response to your comments, we acknowledge the limitations imposed by our current experimental setup, which does not allow for real-time monitoring of blastoid development within the hydrogel encapsulation *in vitro*. However, we have made efforts to address this concern by conducting a comprehensive analysis at multiple time points post-encapsulation. This additional data helps to elucidate the developmental trajectory of the blastoids within the hydrogel and provides a more complete picture of their state over time (Figure 8c). We found that while there is an initial period of adaptation post-encapsulation, we expanded our discussion on the biocompatibility and biodegradability of the blastoid capsules, emphasizing the significance of these properties for *in vitro* applications. With regard to Figure 6f, there may be a description in our image and text that leads to misunderstandings for the reviewer. We are here to explain the image in Figure 6f. In Figure 6f, the scale bars for Day 1, Day 2, Day 3, and Day 4 are all represented by 200µm, but the lengths of the scale bars in the images are actually not consistent. If measured according to the scale bars, it can be observed that the blastoids gradually expand on the hydrogel. In Supplement Figure 8c, we observed that blastoids encapsulated by the hydrogel proliferate and hatch out within it, further demonstrating the biocompatibility of the hydrogel with blastoids. Furthermore, we have included data that demonstrates the enhanced tubal-delivery and cryopreservation capabilities of blastoids encapsulated by hydrogel capsules compared to those without this 3D protection. This underscores the potential advantages of our approach for *in vivo* applications. We would like to re-emphasize the efficiency of generating primates blastoids from different cell sources and the utility of our high-throughput blastoids-capsules preparation strategy via microfluidic technology. We believe this approach could be extended to other globular organoids, expanding the potential applications. We are grateful for the opportunity to address your concerns and believe that the additional data and revisions have significantly strengthened our manuscript. Inspired by your suggestions, we plan to explore the real-time development of blastoids in a 3D hydrogel environment in our future research to further advance the field. At the same time, we are collecting primate resources and incorporating the transplantation of blastoids into

our future work plans.

Reviewer 1 was asked by the editorial team to comment on how the authors addressed the concerns raised by Reviewer 2 in the first round because Reviewer 2 was unable to provide their report.

The following is Rev. 1's assessment on Rev. 2's points.

(2-1): No significant improvement in blastoid morphology was observed using either white light or immunofluorescence staining. The ratio of induced TLC seems relatively low based on the response to Figure 2-1, and the signal for HLA-G may be non-specific noise. Additionally, there are too few markers for TE, and the GATA3 appears to be expressed across all three cell types, while the expression level of TEAD4 is not particularly high in TLC according to the RNA-seq data.

Our reply: Thank you for your valuable comments. We acknowledge your concern about the relatively low ratio of induced TLCs. This could be due to intrinsic differences between blastoids and natural blastocysts, as well as the adaptation of methods originally developed for human blastocysts and human blastoid-derived TLCs. Despite the lower efficiency, the derived TFAP2C+/GATA3+ TLCs demonstrated the ability to be stably passaged *in vitro* for over 10 passages. Meanwhile, we reported that ESC, TS, XEN cells could be co-cultured and stably passaged multiple times¹, this approach may provide insights for improving the derivation and maintenance of monkey blastoid-derived cell lines.

Regarding the potential non-specific noise for HLA-G staining, we confirm the presence of HLA-G expression in our TLCs. To further strengthen our conclusion, we also assessed the expression of other EVT markers, such as CGB, which corroborated the differentiation potential of these cells toward EVT. Besides, *CDH5* and *ITGA5*, the EVT markers, were expressed in TLCs-derived cells (Response to Reviewers Figure 2-1 e-f).

We appreciate your observation regarding the expression of GATA3 and TEAD4 in TLCs. In mice, CDX2 and GATA3 are regulated downstream of TEAD4⁸. However, in humans, GATA3 expression is TEAD4-independent, with GATA3 and TEAD4/CDX2 functioning in parallel during trophoblast differentiation⁹.

In our monkey blastoid data, GATA3 was ubiquitously expressed in TE cells, while TEAD4 was not consistently upregulated in TE. This discrepancy could partially result from suboptimal conditions for trophoblast differentiation. We agree that optimizing conditions for trophoblast lineage development is crucial, and we plan to address this in our ongoing experiments.

Response to Reviewers Figure 2-1: Derivation and identification of TLC from blastoid. (a) Representative immunofluorescent staining of markers of trophoctoderm (TFAP2C/GATA3) in b-TLCs. Scale bars, 100 μ m. (b) Representative immunofluorescent staining images of marker genes detection in STB-like cells (SDC1). Scale bars, 100 μ m. (c) Representative immunofluorescent staining images of marker genes detection in EVT-like cells (HLA-G). Scale bars, 100 μ m. (d) Representative immunofluorescent staining images of marker genes detection in EVT-like cells (CG β). Scale bars, 100 μ m. (e-f) RT-qPCR analysis of *ITGA5* (e) and *CDH5* (f), the gene expression levels are presented as relative expression.

(2-2): It may be appropriate to present the data showing co-expression of the markers for each of the three “lineage cells”, instead of a single expression pattern for blastoid and blastocysts.

Our reply: Thank you for your valuable suggestion. In response to the previous review comment regarding the parallel staining of blastoids and natural embryos, we have now incorporated additional parallel immunofluorescence staining data to further illustrate the similarities between these structures. The new staining results demonstrate the following: NANOG expression is specifically restricted to the inner cell mass (ICM) in both blastoids and natural blastocysts. SOX2 is widely expressed throughout both blastoids and blastocysts but is restricted to NANOG+ epiblast cells within the ICM, where it does not co-express with GATA3 or SOX17. The outer layer of both blastoids and natural blastocysts shows clear expression of GATA2 and GATA3, marking the trophoblast lineage. SOX17+ cells are diffusely distributed within both blastoids and natural blastocysts, indicating the presence of hypoblast lineage cells. By presenting co-expression patterns of lineage-specific markers and their spatial distribution in both blastoids and natural blastocysts, we aim to provide a more detailed characterization. This approach highlights the structural and molecular similarities between these two systems and demonstrates the potential of our blastoids to mimic natural embryonic development.

Response to Reviewers Figure 2-2: The characterization of blastoid compared with monkey blastocyst-E9. (a) Representative immunofluorescent staining images of GATA2/GATA3/NANOG in monkey blastoid (top) and blastocyst (bottom). Scale bars, 100

µm. (b) Representative immunofluorescent staining images of GATA3/SOX2/SOX17 in monkey blastoid (top) and blastocyst (bottom). Scale bars, 100 µm.

(2-3): The ratio of GATA4+ HLCs seems too high, similar to TLCs, which does not fit the ratio of 2:2:6 described previously, and the location of the signal seems not right.

Our reply: Thank you for your valuable comments. Upon reviewing our scRNA-seq analysis, we observed that GATA4+ cells are predominantly clustered within the HLC group, which could explain the higher ratio observed in this lineage. While GATA4 is also expressed at lower levels in other lineages, this might reflect the current limitations in our monkey blastoid construction method, which still requires further optimization. Additionally, the challenge of achieving successful *in vivo* development after blastoid transfer in primates has not been fully addressed yet. Nonetheless, our study has significantly improved the efficiency of monkey blastoid construction and explored the potential of both young and aged M-iPSCs and M-nt-ESCs in generating blastoids, with further advancements in engineering blastoids using biomaterials.

In addition to the scRNA-seq data, we also performed flow cytometry analysis, which showed that GATA3+ and GATA6+ cells constituted 90% and 76% of the blastoid population, respectively, indicating a high proportion of GATA3+/GATA6+ cells in the blastoids. Furthermore, we collected pre-implantation natural monkey embryos and performed immunofluorescence staining, which revealed a similar high presence of GATA3+/GATA6+ cells. The unclear separation between TE and PE in natural monkey embryos is a well-established observation, and given the similarity between our monkey blastoids and natural embryos, we also observed a similar pattern in our blastoids.

a

Response to Reviewers Figure 2-3: Immunofluorescent staining images of monkey blastocyst-E9. (a) Representative immunofluorescent staining images of OCT4/GATA6/GATA3 in monkey blastocyst-E9.

(2-4): The HLC cells express higher levels of GATA3 compared to the TLC cells, which raises my suspicion that your TLC in Supplementary Fig. 3f may not be true TLC.

Our reply: Thank you for pointing out this concern. In our study, the cell clustering was defined based on cellular proportions (Supplementary Fig. 3c), with Cluster 2 identified as HLC, Clusters 4 and 7 as ELC, and Clusters 1, 3, and 6 as TLC. We acknowledge the observation that trophoblast lineage (TLC) markers are upregulated in the HLC lineage. This phenomenon has also been reported in similar studies on blastoids by previous work^{4, 10}. All of these studies, including ours, utilized an induction strategy involving sequential

exposure to HDM and TDM signals, which may have contributed to this overlap in marker expression.

Additionally, in early primate embryonic development, as shown in our previous research, the distinction between the TE and PE lineages is not well-defined before implantation, with TE and PE sharing several markers. To further investigate this, we analyzed monkey embryos at day 9 post-fertilization. Immunofluorescence staining revealed that most GATA6⁺ cells also express GATA3, consistent with our scRNA-seq data.

In this study, we validated the identity of TLC cells by confirming their expression of trophoblast markers, such as GATA3, GATA2, TFAP2C, and CDX2. Moreover, after extended *in vitro* culture, we detected mCG expression, which indicates that our TLCs share significant similarity with the *in vivo* TE.

Response to Reviewers Figure 2-4: The characterization of monkey blastocyst, blastoid and blastoid-derived TLCs. (a) Representative immunofluorescent staining images of OCT4/GATA6/GATA3 in monkey blastocyst-E9. Scale bars, 100 μ m. (b) Representative immunofluorescent staining images of GATA2/GATA3/NANOG in monkey blastoid. Scale bars, 100 μ m. (c) Representative immunofluorescent staining images of TFAP2C/GATA3 in monkey blastoid-derived TLCs. Scale bars, 100 μ m. (d) Representative immunofluorescent staining image of CDX2 in monkey blastoid. Scale bars, 100 μ m.

(2-5): The in vitro developmental potential of blastoids should be characterized in more detail beyond simple staining. Typical structures such as the epiblast, amniotic cavity, and yolk sac cavity should be included.

Our reply: Thank you for your valuable comment. After delayed culture, we observed outgrowths in the blastoids, with more pronounced separation of GATA3 and GATA6, which aligns with previously reported findings on the delayed culture of natural blastoids. It is typically during the gastrulation stage of primate embryos that the amniotic cavity becomes visible, and since our delayed culture did not reach this stage, we did not observe a distinct amniotic cavity. After 3 days of delayed culture, the development efficiency of our blastoids was lower, mainly due to the loss of epiblasts. We also consulted with the first author of a study on blastoids delayed culture, who confirmed that the efficiency of delayed culture for blastoids was similarly low.

As for the delayed culture of primate blastoids to the gastrulation stage, particularly in the work of Li et al., there is still potential for optimization. For example, cells expressing the primitive streak marker T appeared in the trophoblast layer, OCT4 was expressed throughout the blastoid, and OCT4-positive cells in the epiblast region showed incorrect upregulation of GATA6. These findings were inconsistent with previously reported delayed culture results for natural embryos, suggesting that further improvements are needed in the delayed culture of blastoids to reach the gastrulation stage. Thus, as you pointed out, extending the culture of monkey blastoids to E14 (e.g., for organogenesis and clear fetal morphology) would be of great scientific value. However, achieving this will require more detailed and in-depth research, which will be a focus for future studies in this field.

Despite these challenges, our study provides significant advancements: we have established an efficient method for generating blastoids, which facilitates their use in high-throughput screening. Additionally, this method enables the generation of blastoids from aged cells, offering a platform to study how aging influences cell lineage differentiation from a blastoid perspective. Based on this, we have combined microfluidic technology with biomaterial encapsulation to generate engineered blastoids, offering a new approach for blastoid production.

(2-6): I am satisfied with your explanation. Exploring cells from different ages will be beneficial for us in establishing models in the future!

Our reply: Thank you for your kind feedback. We are glad that you found our explanation satisfactory.

(2-7): I'm convinced by what you've presented.

Our reply: Thank you for your kind feedback. We are glad that you found our explanation satisfactory.

(2-8): The authors have answered my questions properly for the degradability of these hydrogels.

Our reply: Thank you for your kind feedback. We are glad that you found our explanation satisfactory.

(2-9): After the embryo transfer, is there any further observation or monitoring of the transplanted blastoid development, or are the levels of chorionic gonadotropin being checked for changes? More explanations and experiments are needed

Our reply: Thank you for raising this important point. After the blastoid transfer to the monkey, we conducted continuous monitoring. One week after the blastoid transfer, we observed an initial rise in mCG levels, which subsequently decreased. Three weeks post-transfer, ultrasound scans revealed that the recipient monkey did not achieve pregnancy. This outcome is similar to natural embryo transfers, where a transient increase in mCG levels is often observed, followed by a decline, even when pregnancy does not progress further. In the work by Li et al., no significant difference was observed in the mCG levels between the blastoid-transferred monkeys and negative control monkeys. Additionally, they did not perform lineage-specific identification of the embryos after transfer.

Overall, while we observed some initial hormonal responses (such as mCG rise) following the blastoid transfer, the development did not proceed to pregnancy. This suggests that further studies are needed to better understand the development and potential of blastoid transfer in primates. These findings also highlight the need for more comprehensive research to optimize the *in vivo* development of monkey blastoids.

(2-10): The authors only briefly compare the differences to Jie Li et al., but the discussion section of the article requires a deeper analysis. This part should be integrated into the main text.

Our reply: Thank you for pointing this out. We have incorporated the relevant content into the discussion section of the manuscript.

Reviewer #3 (Remarks to the Author):

I believe the revision process has been very helpful to improve the overall quality of the manuscript and to support the claims. I believe the authors have done a good job addressing my concerns as well as the concerns of the other reviewers. I feel the inclusion of the new data supports the conclusions and support the publication given minor edits are addressed:

Our reply: Thank you very much for your positive feedback and constructive comments. We are pleased that the revisions have improved the overall quality of the manuscript and we appreciate your recognition of our efforts to address the concerns you raised.

1. Some references are misplaced or are missing context, example Line 620 the reference 54.

Our reply: Thank you for pointing out these issues. We have reviewed and corrected the references.

2. The authors disregard very important historical references regarding the discovery of the hippo signaling pathway in trophoctoderm differentiation^{1,2} although they cite a reference showing its evolutionary conservation³. A simple immunostaining for YAP in monkey blastocysts and blastoids should answer Hippo signaling role in monkey TE

specification4.

Our reply: Thank you for your valuable comments. We have added relevant historical references, including the important findings by Nishioka et al. (2008, 2009) on the role of the Hippo signaling pathway in trophoblast differentiation. Therefore, in future optimization and transplantation experiments with blastoids, we will investigate the impact of the Hippo signaling pathway on TE differentiation and consider its potential role in improving the *in vitro* culture and *in vivo* implantation efficiency of blastoids. Additionally, we plan to conduct YAP immunofluorescence experiments to validate the role of the Hippo signaling pathway in monkey blastoids. These experiments will help us gain a more comprehensive understanding of the specific functions of the Hippo signaling pathway in blastoids development.

References

1. Wei Y, *et al.* Dissecting embryonic and extraembryonic lineage crosstalk with stem cell co-culture. *Cell* **186**, 5859-5875 e5824 (2023).
2. Guo M, *et al.* Self-renewing human naive pluripotent stem cells dedifferentiate in 3D culture and form blastoids spontaneously. *Nat Commun* **15**, 668 (2024).
3. Karvas RM, *et al.* 3D-cultured blastoids model human embryogenesis from pre-implantation to early gastrulation stages. *Cell Stem Cell* **30**, 1148-1165 e1147 (2023).
4. Li J, *et al.* Cynomolgus monkey embryo model captures gastrulation and early pregnancy. *Cell Stem Cell* **30**, 362-377 e367 (2023).
5. Yu L, *et al.* Large-scale production of human blastoids amenable to modeling blastocyst development and maternal-fetal cross talk. *Cell Stem Cell* **30**, 1246-1261 e1249 (2023).
6. Kang Y, *et al.* Cloning and base editing of GFP transgenic rhesus monkey and off-target analysis. *Sci Adv* **8**, eabo3123 (2022).
7. Liu Z, *et al.* Cloning of Macaque Monkeys by Somatic Cell Nuclear Transfer. *Cell* **172**, 881-887 e887 (2018).
8. Ralston A, *et al.* Gata3 regulates trophoblast development downstream of Tead4 and in parallel to Cdx2. *Development* **137**, 395-403 (2010).
9. Stamatiadis P, *et al.* TEAD4 regulates trophectoderm differentiation upstream of CDX2 in a GATA3-independent manner in the human preimplantation embryo. *Hum Reprod* **37**, 1760-1773 (2022).
10. Yu L, *et al.* Blastocyst-like structures generated from human pluripotent stem cells. *Nature* **591**, 620-626 (2021).